# TRIB3 promotes MYC-associated lymphoma development through suppression of UBE3B-mediated MYC degradation

Ke Li[1,2,5], Feng Wang[3,5], Zhao-na Yang[3], Ting-ting Zhang[3], Yu-fen Yuan[4], Chen-xi Zhao[3], Zaiwuli Yeerjiang[3], Bing Cui [3], Fang Hua [3], Xiao-xi Lv[3], Xiao-wei Zhang [3], Jiao-jiao Yu[3], Shan-shan Liu[3], Jin-mei Yu [3], Shuang Shang[3], Yang Xiao[1] & Zhuo-wei Hu [1,3✉]

The transcription factor MYC is deregulated in almost all human cancers, especially in aggressive lymphomas, through chromosomal translocation, amplification, and transcription hyperactivation. Here, we report that high expression of tribbles homologue 3 (TRIB3) positively correlates with elevated MYC expression in lymphoma specimens; *TRIB3* deletion attenuates the initiation and progression of MYC-driven lymphoma by reducing MYC expression. Mechanistically, TRIB3 interacts with MYC to suppress E3 ubiquitin ligase UBE3B-mediated MYC ubiquitination and degradation, which enhances MYC transcriptional activity, causing high proliferation and self-renewal of lymphoma cells. Use of a peptide to disturb the TRIB3-MYC interaction together with doxorubicin reduces the tumor burden in $Myc^{E\mu}$ mice and patient-derived xenografts. The pathophysiological relevance of UBE3B, TRIB3 and MYC is further demonstrated in human lymphoma. Our study highlights a key mechanism for controlling MYC expression and a potential therapeutic option for treating lymphomas with high TRIB3-MYC expression.

[1] National Clinical Research Center for Metabolic Disease, Department of Metabolism and Endocrinology, the Second Xiangya Hospital, Central South University, 410011 Changsha, Hunan, China. [2] NHC Key Laboratory of Biotechnology of Antibiotics, Institute of Medicinal Biotechnology, Chinese Academy of Medical Sciences & Peking Union Medical College, 100050 Beijing, China. [3] Immunology and Cancer Pharmacology Group, State Key Laboratory of Bioactive Substance and Function of Natural Medicines, Institute of Materia Medica, Chinese Academy of Medical Sciences & Peking Union Medical College, 100050 Beijing, China. [4] Anyang Tumor Hospital, Henan University of Science and Technology, 300020 Anyang, China. [5]These authors contributed equally: Ke Li, Feng Wang. ✉email: huzhuowei@csu.edu.cn

Non-Hodgkin lymphoma represents a type of aggressive malignancy that originates in lymphocytes or lymph node (LN) tissues[1,2]. Although advances in immunochemotherapy and supportive care have improved patient outcomes, a significant proportion of patients experience early treatment failure or relapse after initial therapy[3,4]. The oncogene *MYC* is a transcription factor that drives cancer cell growth by controlling universal transcription programs, including the cell survival, cell cycle, and metabolism[5–7]. MYC is deregulated in almost all human cancers, especially Burkitt lymphoma (BL), other aggressive B cell lymphomas (BCLs) and T cell lymphomas (TCLs). Although chromosomal translocation or amplification of MYC partially explains the altered MYC protein[8–10], a large proportion of lymphomas with high MYC protein expression rarely exhibit these *MYC* rearrangements, suggesting that mechanisms other than gene rearrangements are responsible for the elevated MYC expression in a considerable proportion of lymphoma cases. Moreover, high MYC expression is correlated with poor prognoses and drug resistance of lymphomas and other hematological malignancies[11,12]. Targeting MYC, especially in combination with traditional therapies, is considered an attractive therapeutic strategy for lymphomas and other MYC-driven cancers.

Tribbles homologue 3 (TRIB3), a member of the pseudokinase family, acts as a stress sensor that responds to a diverse range of stresses, including inflammation, insulin, insulin-like growth factor 1, and ER stress[13–15]. TRIB3 is also well known as a crucial "stress adjusting switch" that links homeostasis, metabolic disease, and cancer through its interactions with intracellular signaling and functional proteins[16–19]. TRIB3 is emerging as a potential therapeutic target for cancer because abrogating its expression dramatically reduces tumorigenesis and cancer progression[17–22]. Interestingly, the expression of TRIB2, another member of the pseudokinase family, is elevated in T-cell acute lymphoblastic leukemia (T-ALL)[23], and TRIB2 has emerged as a regulator of thymocyte cellular proliferation[24]. TRIB1, the third member of this family, has a negative regulatory effect on immunoglobulin production in murine B cells[25]. However, the role of TRIB3 in lymphomagenesis remains uncharacterized.

Despite its attractiveness as a cancer target, MYC has been considered "undruggable" and remains outside reach of pharmacological regulation, mainly due to its nuclear localization, lack of a defined ligand-binding site, and large protein-protein interaction (PPI) surface[26,27]. Because targeting MYC itself is so challenging, efforts have focused on indirect targeting strategies[26–30]. One evolving approach is the selective degradation of MYC by hijacking the degradation machinery or targeting specific E3 ligases of MYC[31–33]. Utilizing peptides to overcome the limitations of small-molecule compounds, which can be inefficient in interfering with large PPI surfaces, is a promising strategy for MYC inhibition[34]. We recently reported that TRIB3 enhances the stability of the oncoproteins PML-RARα and β-catenin/TCF4 to promote advanced precancerous lesions (APL) and colorectal cancer progression[17,18].

In this work, we hypothesize that TRIB3 contributes to lymphoma pathogenesis by promoting MYC-deregulated lymphomagenesis. We examined the expression and roles of TRIB3 in primary lymphoma cells from patients and patient-derived xenograft (PDX) mice. We found that TRIB3 interacts with MYC to suppress E3 ubiquitin ligase UBE3B-mediated MYC ubiquitination and degradation, which causes high proliferation and self-renewal of lymphoma cells. This study reveals several functional implications for MYC-associated lymphoma therapy.

## Results

**Deletion of TRIB3 suppresses lymphomagenesis.** To examine the role of TRIB3 in lymphomagenesis, we searched the Oncomine database and found that *TRIB3* expression was elevated in peripheral T-cell lymphoma (PTCL) and diffuse large B-cell lymphoma (DLBCL) compared to normal lymphocytes (Fig. 1a, b). Additionally, the copy number of the *TRIB3* gene was increased in marginal zone BCL compared to lymphoid tissue (Fig. 1c). We sequenced the coding region of *TRIB3* in 89 human lymphoma samples to identify nonsynonymous variants. The allele frequencies of nonsynonymous (Q84R) and synonymous (Y111Y and A323A) mutations in lymphoma patients were 38.2% and 47.2%, respectively (Fig. 1d and Supplementary Tables 1 and 2). Because of the unlikely biological significance of the two synonymous mutations (Y111Y and A323A), only the prevalent missense Q84R variant, in which a polar uncharged amino acid is substituted with a charged one, was further examined for association with lymphoma phenotypes. Interestingly, the TRIB3 Q84R mutant was shown to be more stable than wild-type (WT) TRIB3 (Supplementary Fig. 1a), which may result in high TRIB3 expression in human lymphomas. These data suggest the physiological significance of *TRIB3* in human lymphoma. We crossed floxed *Trib3* (*Trib3*^F/F) mice with *Cre*^Lysm mice expressing Cre in myeloid cells[18], *Cre*^Lck mice expressing Cre in thymocytes or *Cre*^CD19 mice expressing Cre in B lymphocytes (Fig. 1e and Supplementary Fig. 1b). Reduction of TRIB3 expression was found in these cell lineages, and the heterozygous deletion of *Trib3* reduced TRIB3 expression to a level similar to that resulting from the homozygous deletion of *Trib3* (Supplementary Fig. 1c, d). The thymus and spleen sizes of 5-week-old *Cre*^Lysm*Trib3*^F/F, *Cre*^Lck*Trib3*^F/F, and *Cre*^CD19*Trib3*^F/F mice did not differ substantially from those of age-matched WT mice (Supplementary Fig. 1e). Notably, *Trib3* ablation in myeloid/thymocyte/B cells did not affect the total number of thymocytes, splenic B cell, T cell, and myeloid cell populations (Supplementary Fig. 1f–i), indicating that specific ablation of *Trib3* in these cells does not affect lymphocyte and myeloid cell development.

We separately crossed *Cre*^Lysm*Trib3*^F/F, *Cre*^Lck*Trib3*^F/F, or *Cre*^CD19*Trib3*^F/F mice with congenic *Myc*^Eμ mice, the most widely studied model used to understand the genetic mechanisms of lymphomagenesis (Supplementary Fig. 1b). As expected, *Myc*^Eμ mice developed BCL with an average latency of 140 days with enlarged spleens and LNs (Figs. 1e, f). However, the sizes and weights of spleens and LNs in *Myc*^Eμ*Cre*^CD19*Trib3*^F/+ and *Myc*^Eμ*Cre*^CD19*Trib3*^F/F mice were significantly reduced compared with those in *Myc*^Eμ *Cre*^Lysm*Trib3*^F/+ mice, *Myc*^Eμ *Cre*^Lck*Trib3*^F/+ mice and *Myc*^Eμ mice (Fig. 1f–h). Moreover, B lymphocyte-specific ablation of *Trib3* resulted in the extended survival of mice with lymphoma (Fig. 1i). Thus, the loss of *TRIB3* in B lymphocytes, but not in myeloid cells or thymocytes, reduces MYC-driven BCL development. To determine whether TRIB3 was required for not only lymphoma initiation but also the malignancy of established lymphomas, we crossed *Trib3*^F/F mice with *Cre*^ERT2 mice to generate *Cre*^ERT2*Trib3*^F/F mice (Supplementary Fig. 1j), enabling temporally controlled Cre activation by tamoxifen treatment. These mice were further crossed with *Myc*^Eμ mice to analyze BCL development. Upon the emergence of lymphoma, *Myc*^Eμ *Cre*^ERT2*Trib3*^F/+ mice were treated with vehicle or tamoxifen to monitor BCL progression (Fig. 1j). Although *Myc*^Eμ *Cre*^ERT2*Trib3*^F/+ mice became moribund as early as 53 days after vehicle dosing and 65% died or became moribund by day 120, none of the tamoxifen-treated *Myc*^Eμ *Cre*^ERT2*Trib3*^F/+ mice died within 180 days post treatment (Fig. 1k). Histological analysis revealed reduced tumor foci in the spleens and LNs of tamoxifen-treated *Myc*^Eμ *Cre*^ERT2*Trib3*^F/+ mice compared to lymphoma-matched vehicle recipients (Fig. 1l), which was caused by the tamoxifen-induced loss of TRIB3 in lymphoma tissues (Supplementary Fig. 1j). These data suggest that *Trib3* deletion suppresses the initiation and progression

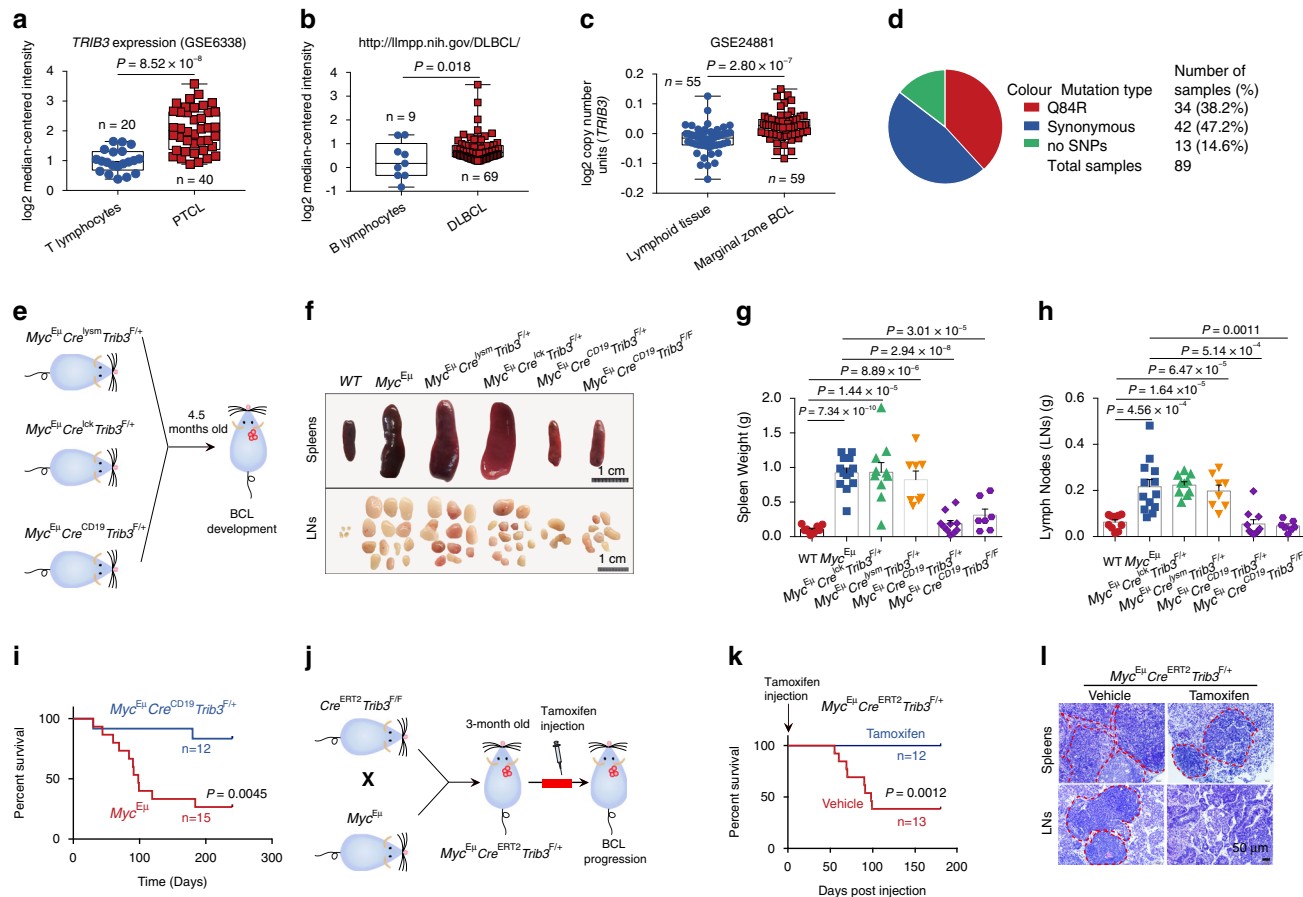

**Fig. 1 TRIB3 promotes B cell lymphomagenesis. a, b** The mRNA level of *TRIB3* in lymphoma patients and the corresponding control groups. The center line indicates the median, bounds of box = 25th and 75th percentiles; bars = 10th and 90th percentiles; whiskers = min to max, show all points. Statistical significance was determined by two-tailed Student's *t* test. *P* values: $8.52 \times 10^{-8}$ **a**; 0.018 **b**. **c** *TRIB3* gene copy numbers in the lymphoma set (GSE24881) were analyzed at the Oncomine website. The center line indicates the median, bounds of box = 25th and 75th percentiles; bars = 10th and 90th percentiles; whiskers = min to max, show all points. Statistical significance was determined by two-tailed Student's *t* test. *P* value: $2.80 \times 10^{-7}$. **d** Pie chart showing all the types and statistics of *TRIB3* gene mutation in 89 lymphoma patients. **e** Schematic strategy for studying the development of B cell lymphoma (BCL) in *Myc*$^{E\mu}$ mice with myeloid cell-specific deletion of *Trib3* (*Myc*$^{E\mu}$*Cre*$^{Lysm}$*Trib3*$^{F/+}$), thymocyte-specific deletion of *Trib3* (*Myc*$^{E\mu}$*Cre*$^{Lck}$*Trib3*$^{F/+}$), or B lymphocyte-specific deletion of *Trib3* (*Myc*$^{E\mu}$*Cre*$^{CD19}$*Trib3*$^{F/+}$). **f** Representative spleens and lymph nodes (LNs) obtained from WT, *Myc*$^{E\mu}$, *Myc*$^{E\mu}$*Cre*$^{Lysm}$*Trib3*$^{F/+}$, *Myc*$^{E\mu}$*Cre*$^{Lck}$*Trib3*$^{F/+}$, *Myc*$^{E\mu}$*Cre*$^{CD19}$*Trib3*$^{F/+}$, or *Myc*$^{E\mu}$*Cre*$^{CD19}$*Trib3*$^{F/F}$ mice (4.5 months old). Scale bar, 1 cm. **g, h** The data indicate the spleen and LN weights of WT (*n* = 10), *Myc*$^{E\mu}$ (*n* = 13), *Myc*$^{E\mu}$*Cre*$^{Lysm}$*Trib3*$^{F/+}$ (*n* = 8), *Myc*$^{E\mu}$*Cre*$^{Lck}$*Trib3*$^{F/+}$(*n* = 10), *Myc*$^{E\mu}$*Cre*$^{CD19}$*Trib3*$^{F/+}$ (*n* = 10), or *Myc*$^{E\mu}$*Cre*$^{CD19}$*Trib3*$^{F/F}$ mice (*n* = 7) (4.5 months old). Data are represented as means ± SEM. Statistical significance was determined by two-tailed Student's *t* test. *P* values: $7.34 \times 10^{-10}$ (WT vs. *Myc*$^{E\mu}$), $1.44 \times 10^{-5}$ (WT vs. *Myc*$^{E\mu}$*Cre*$^{Lck}$*Trib3*$^{F/+}$), $8.89 \times 10^{-6}$ (WT vs. *Myc*$^{E\mu}$*Cre*$^{Lysm}$*Trib3*$^{F/+}$), $2.94 \times 10^{-8}$ (*Myc*$^{E\mu}$ vs. *Myc*$^{E\mu}$*Cre*$^{CD19}$*Trib3*$^{F/+}$), $3.01 \times 10^{-5}$ (*Myc*$^{E\mu}$ vs. *Myc*$^{E\mu}$*Cre*$^{CD19}$*Trib3*$^{F/F}$) **g**; $4.56 \times 10^{-4}$ (WT vs. *Myc*$^{E\mu}$), $1.64 \times 10^{-5}$ (WT vs. *Myc*$^{E\mu}$*Cre*$^{Lck}$*Trib3*$^{F/+}$), $6.47 \times 10^{-5}$ (WT vs. *Myc*$^{E\mu}$*Cre*$^{Lysm}$*Trib3*$^{F/+}$), $5.14 \times 10^{-4}$ (*Myc*$^{E\mu}$ vs. *Myc*$^{E\mu}$*Cre*$^{CD19}$*Trib3*$^{F/+}$), and 0.0011 (*Myc*$^{E\mu}$ vs. *Myc*$^{E\mu}$*Cre*$^{CD19}$*Trib3*$^{F/F}$) **h**. **i** Kaplan–Meier survival curves for *Myc*$^{E\mu}$ (*n* = 15) and *Myc*$^{E\mu}$*Cre*$^{CD19}$*Trib3*$^{F/+}$ mice (*n* = 12). Statistical difference was determined by two-sided log-rank test. *P* value: 0.0045. **j** Schematic representation of generating the inducible deletion of *Trib3* in *Myc*$^{E\mu}$ mice (*Myc*$^{E\mu}$*Cre*$^{ERT2}$*Trib3*$^{F/+}$). **k** Kaplan–Meier survival curves for *Myc*$^{E\mu}$*Cre*$^{ERT2}$*Trib3*$^{F/+}$ mice treated with vehicle (*n* = 13) or tamoxifen (*n* = 12 per group, 250 mg/kg). Statistical difference was determined by two-sided log-rank test. *P* value: 0.0012. **l** Representative H&E staining of the spleens and LNs of 5-month-old *Myc*$^{E\mu}$*Cre*$^{ERT2}$*Trib3*$^{F/+}$ mice treated with or without tamoxifen. Scale bar, 50 μm. The data are presented as presentative from three independent experiments. The *Myc*$^{E\mu}$*Cre*$^{Lysm}$*Trib3*$^{F/+}$, *Myc*$^{E\mu}$*Cre*$^{Lck}$*Trib3*$^{F/+}$, *Myc*$^{E\mu}$*Cre*$^{CD19}$*Trib3*$^{F/+}$, *Myc*$^{E\mu}$*Cre*$^{CD19}$*Trib3*$^{F/F}$, and *Myc*$^{E\mu}$*Cre*$^{ERT2}$ mice are heterozygous for the Myc$^{E\mu}$ allele. Source data are provided as a Source Data file.

of MYC-driven BCL and that *Trib3* is essential for B cell lymphomagenesis.

**TRIB3 promotes lymphoma by supporting MYC activity.** To investigate the mechanism by which TRIB3 promotes B cell lymphomagenesis, BCL cells were isolated from the spleens of *Myc*$^{E\mu}$ and *Myc*$^{E\mu}$*Cre*$^{CD19}$*Trib3*$^{F/-}$ mice and evaluated (Fig. 2a). *Trib3* deletion decreased the number of Ki-67-positive cells (Fig. 2b). BCL cells from *Myc*$^{E\mu}$ *Cre*$^{CD19}$*Trib3*$^{F/F}$ mice yielded substantially fewer colonies than those from *Myc*$^{E\mu}$ mice (Fig. 2c).

Especially, there was significant difference between the survival rates of recipient mice transplanted from *Myc*$^{E\mu}$ *Cre*$^{CD19}$*Trib3*$^{F/+}$ and *Myc*$^{E\mu}$ mice for the transplantation cell dose of $5 \times 10^{4}$, suggesting that BCL cells from *Myc*$^{E\mu}$ *Cre*$^{CD19}$*Trib3*$^{F/+}$ mice had a lower capacity for killing the recipient mice (Fig. 2d). Knocking out *TRIB3* in primary human DLBCL (T69), BL (T4 and T5), and PTCL (T144) cells freshly isolated from patient tumors or lymphoma cell lines (Fig. 2e and Supplementary Table 1) reduced the colony-formation units (CFUs) and proliferation (Fig. 2f, g). We next detected the cell death and differentiation in Raji cells with

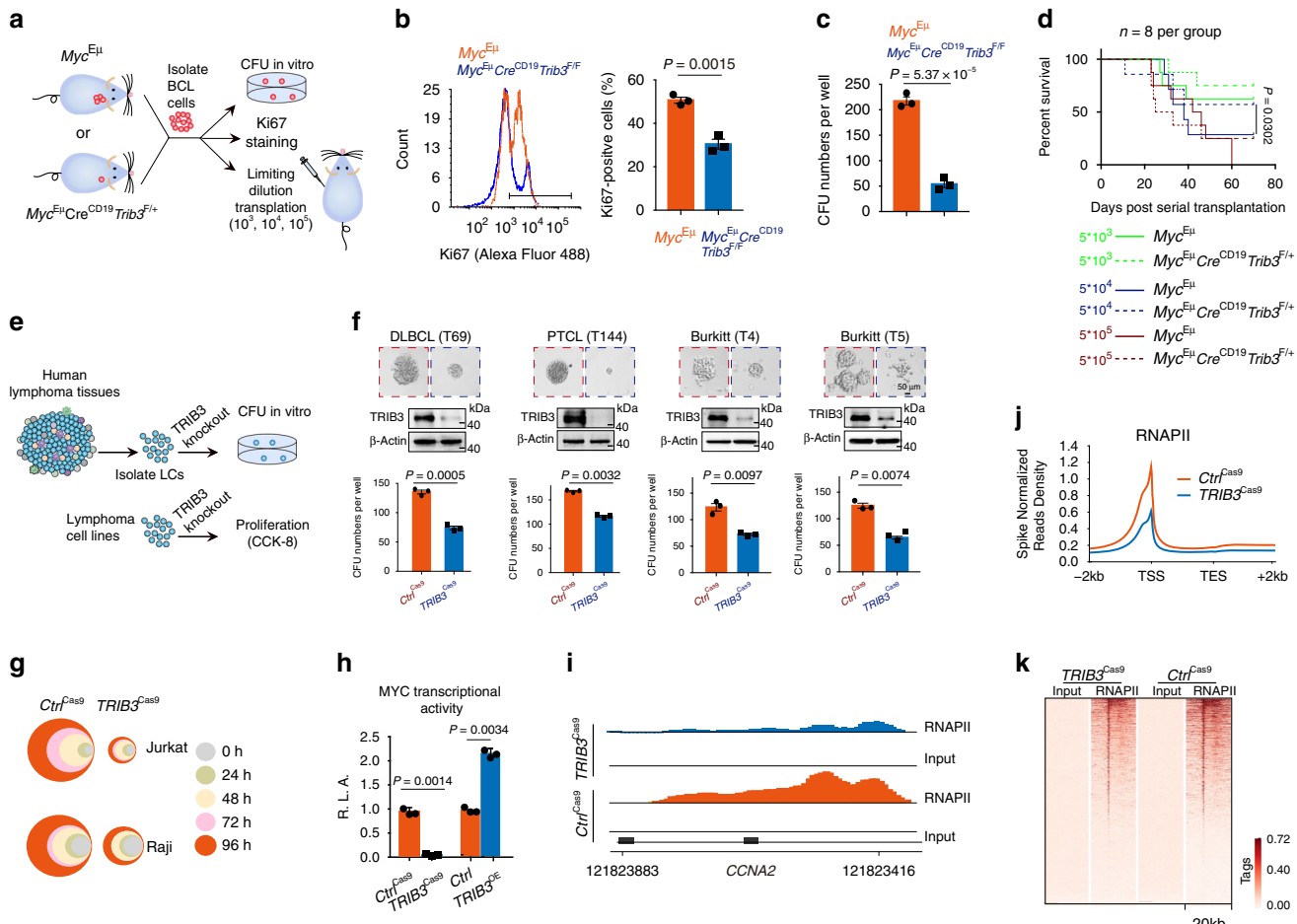

**Fig. 2 TRIB3 enhances the self-renewal and proliferation of LCs. a** Approaches to defining self-renewal ability and proliferation in mouse BCL cells. **b** Cell proliferation was determined by Ki67 staining in BCL cells isolated from $Myc^{E\mu}$ or $Myc^{E\mu}Cre^{CD19}Trib3^{F/F}$ mice (5 months old). Data are representative flow cytometry plots of Ki67 staining (left) and means ± S.E.M of three independent experiments. Statistical significance was determined by two-tailed Student's $t$ test. $P$ value: 0.0015. **c** Colony-forming ability of BCL cells from $Myc^{E\mu}$ or $Myc^{E\mu}Cre^{CD19}Trib3^{F/F}$ mice (5 months old). Shown is means ± S.E.M of three independent experiments. Statistical significance was determined by two-tailed Student's $t$ test. $P$ value: $5.37 \times 10^{-5}$. **d** The tumorigenicity of BCL cells from the indicated mice was compared by limiting dilution transplantation into NSG mice; $n = 8$ mice per group. $P = 0.0302$ by Kaplan–Meier analysis (two-sided log-rank test) for comparison between the groups of the transplantation cell dose of $5 \times 10^{4}$. **e** Approaches to defining the effects of $TRIB3$ deletion on the self-renewal ability of primary human LCs and on the proliferation of lymphoma cell lines. **f** Representative images and colony numbers of LCs from the indicated groups with or without $TRIB3$ deletion Data are presented as the means ± S.E.M from three independent experiments. Statistical significance was determined by two-tailed Student's $t$ test. $P$ value: 0.0005, 0.0032. 0.0097, and 0.0074. Scale bar, 50 μm. **g** Relative cell viabilities of Jurkat and Raji cells with or without $TRIB3$ deletion were detected by CCK-8 for the indicated times of three independent experiments. The colors represent different time points; the diameter indicates the relative cell viability. **h** Effects of $TRIB3$ deletion or overexpression on the transcriptional activity of MYC. $Ctrl^{Cas9}$, $TRIB3^{Cas9}$, $Ctrl$, and $TRIB3^{OE}$ Raji cells were transfected with E-box-dependent luciferase reporter genes of MYC transcription activity, and pTK-Renilla was used as an internal control. After 24 h of transfection, relative luciferase activities (R.L.A) were measured. Shown is means ± S.E.M of three independent experiments. Statistical significance was determined by two-tailed Student's $t$ test. $P$ value: 0.0014, 0.0034. **i** ChIP-sequencing tracks for $CCNA2$ from $Ctrl^{Cas9}$ and $TRIB3^{Cas9}$ Raji cells normalized to spike-in controls. **j** Metagene plots of global RNAPII occupancy at gene bodies in $Ctrl^{Cas9}$ and $TRIB3^{Cas9}$ Raji cells. **k** Heatmap showing occupancy of genome-wide RNAPII peaks in $Ctrl^{Cas9}$ and $TRIB3^{Cas9}$ Raji cells in a window of ±10 kb surrounding the transcription start site (TSS). $Myc^{E\mu}Cre^{CD19}Trib3^{F/+}$ and $Myc^{E\mu}Cre^{CD19}Trib3^{F/F}$ are heterozygous for the $Myc^{E\mu}$ allele. Source data are provided as a Source Data file.

or without $TRIB3$ deletion. $TRIB3$ deletion did not induce a sub-G1 fraction of the cellular DNA content (Supplementary Fig. 2a), nor did it increase CD11b positive cells ratio in FACS analysis (Supplementary Fig. 2b), suggesting that $TRIB3$ depletion does not induce cell death or differentiation in lymphoma cells. Thus, TRIB3 promotes lymphoma by supporting lymphoma cell self-renewal ability and proliferation.

Through RNA sequencing (RNA-seq), we analyzed the gene expression profiles of lymphoma cells with or without $Trib3$ deletion. Gene set enrichment analysis (GSEA) showed highly enriched $MYC$ target genes in BCL cells from

$Myc^{E\mu}Cre^{CD19}Trib3^{F/F}$ mice (Supplementary Fig. 2c). qRT-PCR assays showed that $TRIB3$ deletion suppressed the transcription of MYC target genes (Supplementary Fig. 2d). Consistently, $TRIB3$ depletion decreased MYC transcriptional activity, while $TRIB3$ overexpression had the opposite effect (Fig. 2h). Cluster analysis of the heatmap revealed that the $TRIB3$-depleted and $MYC$-depleted groups had similar $MYC$ target gene expression tendencies (Supplementary Fig. 2e). Because MYC activation has global effects on both promoter binding and elongation by RNA polymerase II (RNAPII) and, in some situations, globally enhances RNAPII function[35], we performed ChIP-sequencing

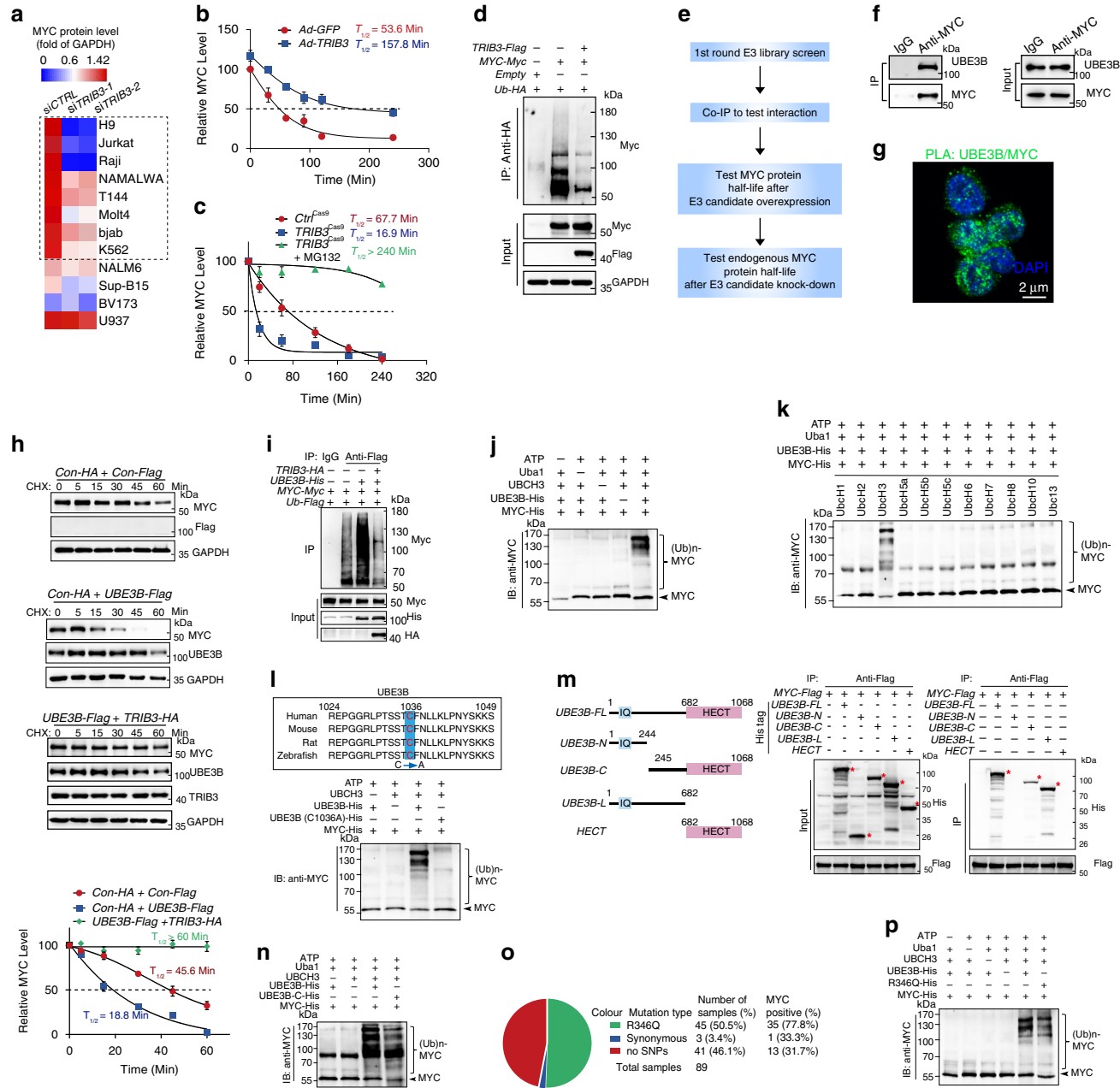

with an antibody against total RNAPII. *TRIB3* deletion globally reduced the association of RNAPII with promoters (Fig. 2i–k). Thus, TRIB3 enhances BCL self-renewal and proliferation by enhancing MYC transcriptional activity and increasing the expression of its target genes.

**TRIB3 disrupts UBE3B-mediated MYC ubiquitination and degradation.** Although similar trends of *MYC* mRNA expression were observed in *Ctrl*[Cas9] and *TRIB3*[Cas9] Raji cells (Supplementary Fig. 2f), reduced MYC protein expression was detected in *Myc*[Eμ]*Cre*[CD19]*Trib3*[F/+] mice compared to *Myc*[Eμ] mice (Supplementary Fig. 2g). Notably, *TRIB3* knockdown reduced MYC expression in most lymphoma lines and in some leukemia lines (Fig. 3a and Supplementary Fig. 2h). *TRIB3* overexpression prolonged the degradation half-life of the MYC protein, while *TRIB3* knockdown had the opposite effect, and the shortened MYC half-life induced by *TRIB3* knockdown was reversed by the proteasome inhibitor MG132 (Fig. 3b, c and Supplementary Fig. 2i, j).

Serine 62 (S62) and threonine 58 (T58) phosphorylation of MYC is important for MYC-mediated gene regulation and its protein stability[36–40]. We used phospho-specific antibodies to accurately quantify the level of Phospho-T58 (P-T58) and Phospho-S62 (P-S62) relative to total MYC in *TRIB3* depleted cells. Upon TRIB3 knockdown in Jurkat and Raji cells, we observed a decrease in the P-S62 signal, but no change in the P-T58 signal (Supplementary Fig. 2k, l). Quantifying the ratio of P-T58 to total MYC revealed a substantial increase in the amount of T58 phosphorylated MYC with *TRIB3* knockdown (Supplementary Fig. 2k, l). In addition, the signal for P-S62 relative to total MYC decreased in TRIB3 depleted cells compared with control group (Supplementary Fig. 2k, l). Similarly, TRIB3 overexpression increased S62 and decreased T58 phosphorylation relative to total MYC level in Raji cells (Supplementary Fig. 2m). Furthermore, the analysis of half-life of endogenous MYC protein revealed that *TRIB3* deletion reduced the stability of MYC protein, especially accelerated P-T58 MYC protein degradation after cycloheximide treatment, as

**Fig. 3 TRIB3 supports MYC stability by disturbing UBE3B-mediated MYC ubiquitination. a** Heatmap presenting TRIB3 expression in 11 blood cancer cell lines and 1 primary human lymphoma cell line (T144) with or without *TRIB3* depletion. **b** Effect of *TRIB3* overexpression on MYC degradation in vivo. Raji cells were infected with *TRIB3*- or *GFP*- adenovirus, and 12 h later, incubated with cycloheximide (CHX; 10 μg/mL) for the indicated times. The data are presented as the means ± S.E.M from three independent experiments. **c** Effect of *TRIB3* deletion on MYC degradation in vivo. *Control* or *TRIB3*-deleted Raji cells were incubated with CHX (10 μg/mL) or CHX plus MG132 (5 μm) for the indicated times. The MYC protein was detected by western blot; GAPDH was used as a loading control. The data are presented as the means ± S.E.M from three independent experiments. **d** Effect of *TRIB3* overexpression on MYC ubiquitination in vivo. HEK 293T cells were transfected with the indicated plasmids for 24 h. Extracts of cells were IP with an anti-HA Ab. Ubiquitinated MYC was detected by immunoblotting. The data are presented as presentative from three independent experiments. **e** Strategy for screening the potential E3 ligases of MYC. **f** The interaction of UBE3B and MYC was evaluated by Co-IP assays. Raji cell extracts were IP with rabbit immunoglobulin G (IgG) or an anti-MYC Ab and blotted with an anti-UBE3B Ab. The data are presented as presentative from three independent experiments. **g** Colocalization of MYC and UBE3B was detected in primary human DLBCL cells (T69) by the Duolink PLA assay. Scale bar, 2 μm. The data are presented as presentative from three independent experiments. **h** Effect of TRIB3 on MYC degradation mediated by UBE3B. HEK 293T cells were transfected with the indicated plasmids, and 12 h later, incubated with CHX (10 μg/mL) for the indicated times. The data are presented as presentative and the means ± S.E.M from three independent experiments. **i** Effect of TRIB3 on MYC ubiquitination induced by UBE3B in vivo. HEK 293T cells were transfected with the indicated plasmids for 24 h. Cell extracts were IP with an anti-Flag Ab. Ubiquitinated MYC was detected by immunoblotting. The data are presented as presentative from three independent experiments. **j** In vitro ubiquitination assays show that purified UBE3B induces polyubiquitination of purified MYC in the presence of UBCH3 (E2), ATP, Uba1 (E1), and biotinylated ubiquitin. The data are presented as presentative from three independent experiments. **k** In vitro ubiquitination assays show that UBE3B mildly induces polyubiquitination of MYC in the presence of UBCH3 (E2) but not in that of other E2s. The data are presented as presentative from three independent experiments. **l** Alignment of partial UBE3B sequences (1024–1049 amino acids in human UBE3B) from the indicated species. The cysteine at position 1036 (C1036) of human UBE3B was mutated to alanine (C1036A). In vitro ubiquitination assays showed that the UBE3B C1036A mutant did not induce polyubiquitination of MYC. The data are presented as presentative from three independent experiments. **m** Mapping UBE3B regions binding to MYC. Left: deletion mutants of UBE3B. Right: HEK 293T cells were cotransfected with the indicated constructs of UBE3B (His tag) and MYC (Flag tag). Cell extracts were IP with an anti-Flag Ab. The data are presented as presentative from three independent experiments. **n** In vitro assays show that the C terminus of UBE3B (UBE3B-C) mildly induces polyubiquitination of MYC. The data are presented as presentative from three independent experiments. **o** Pie chart shows all the types and statistics of *UBE3B* mutations in 89 lymphoma patients. **p** In vitro ubiquitination assays showed that the UBE3B R346Q mutation decreased the polyubiquitination of MYC compared with the wild-type MYC. The data are presented as presentative from three independent experiments. Source data are provided as a Source Data file.

compared to control gRNA (Supplementary Fig. 2n), suggesting mainly the stability of T58 MYC protein is affected in TRIB3 deleted cell. These data indicated that the altered T58 and S62 phosphorylation of MYC contributed to the MYC transcriptional output in TRIB3-depleted cells. Thus, the lymphoma-promoting roles of TRIB3 are associated with increased MYC stabilization in lymphoma cells.

Degradation of the MYC protein depends on continuous ubiquitination via several ubiquitin ligases, including SKP2, TRIM6, TRIM32, FBXW7, and HUWE1[41]. We found that TRIB3 decreased MYC ubiquitination (Fig. 3d) but did not affect the interactions of MYC with these ligases or ligase-mediated MYC ubiquitination (Supplementary Fig. 3a–d). We thus screened an E3 ligase library with 71 classical members by Co-IP assays (Fig. 3e and Supplementary Table 3). Four new E3 ligases were identified as potential E3 ligases of MYC (Supplementary Fig. 3e), in which TRIM21, HACE1, and COP1 neither induced MYC degradation nor increased its ubiquitination (Supplementary Fig. 3f–h). However, UBE3B interacted and colocalized with MYC in human DLBCL cells (Fig. 3f, g). *UBE3B* overexpression decreased the stability and half-life of the MYC protein (Fig. 3h) but increased MYC ubiquitination (Fig. 3i). Interestingly, TRIB3 inhibited UBE3B-enhanced MYC ubiquitination and proteolysis (Fig. 3h, i). Knockdown of *UBE3B*, but not the analogous family members *UBE3A* or *UBE3C*, increased MYC expression (Supplementary Fig. 3i) in Raji cells. We detected the levels of P-S62 and P-T58 MYC in UBE3B-overexpressed cells. Although UBE3B overexpression obviously reduced the levels of P-S62, P-T58 MYC, and total MYC, UBE3B overexpression did not affect P-S62 MYC, nor P-T58 MYC level relative to total MYC in Raji cells (Supplementary Fig. 3j). Furthermore, the prolonged MYC degradation induced by *UBE3B* depletion could be reversed by proteasome inhibition (Supplementary Fig. 4a).

By screening 11 ubiquitin-conjugating (E2) enzymes, UBCH3 was identified to interact with UBE3B (Supplementary Fig. 4b). The in vitro ubiquitylation assay showed that UBE3B-catalyzed MYC ubiquitylation via UBCH3, while the other 10 E2 enzymes did not (Fig. 3j, k). HECT E3 ligases have a cysteine active site that is required for their catalytic activity. The cysteine at position 1036 (C1036) of human UBE3B is evolutionarily conserved from zebrafish to humans (Fig. 3l). Compared with UBE3B, UBE3B (C1036A) did not induce MYC ubiquitination in the presence of UBCH3 (Fig. 3l). The domain covering aa 245–682 of UBE3B was shown to be required for UBE3B-mediated MYC binding (Fig. 3m). The C terminus of UBC3B covering the binding domain of MYC and its catalytic site (C1036) catalyzes MYC ubiquitination similar to that of full-length UBE3B (Fig. 3n). Interestingly, up to 53.9% of lymphoma patients were shown to have missense mutations (R346Q) in exon 12 (97) and synonymous mutations (S962S) in exon 26 (76) (Fig. 3o). Among them, up to 77.8% of patients with the UBE3B R346Q mutation were MYC positive (Fig. 3o and Supplementary Table 2). We thus constructed the UBE3B R346Q mutant, which showed a reduced ability to induce MYC ubiquitination (Fig. 3p) and degradation (Supplementary Fig. 4c) due to an impeded UBE3B/MYC interaction (Supplementary Fig. 4d). These data suggest that UBE3B is a bona fide E3 ubiquitin ligase for MYC ubiquitination and degradation and that UBCH3 functions as an E2 ubiquitin-conjugated enzyme in this pathway. Additionally, cysteine 1036 and arginine 346 in UBE3B function separately as active or binding sites for this process.

We examined the potential ubiquitination site(s) in MYC, and the lysine (K) residues 332, 427, and 443/445 of MYC were identified as potential targets of MYC ubiquitination (Supplementary Fig. 4e). *UBE3B* overexpression did not enhance the ubiquitination and degradation of the MYC K427R mutant but increased the ubiquitination of the K443/445R mutants (Supplementary Fig. 4f, g). We next mutated threonine 58 (T58) to a nonphosphorylatable amino acid. This mutation not only reduced MYC ubiquitination but also diminished the effect of *UBE3B* overexpression on MYC ubiquitination (Supplementary Fig. 4h). Indeed, *UBE3B* overexpression increased MYC

ubiquitylation, while *UBE3B* depletion enhanced the MYC protein stability regardless of the absence of FBXW7 (Supplementary Fig. 4i, j), indicating that two E3 ligases utilize an independent pathway for MYC degradation. In addition, the T58A mutation reduced binding of UBE3B to MYC (Supplementary Fig. 4k). Given that MYC T58 is phosphorylated by GSK3, a GSK3 inhibitor decreased UBE3B-mediated MYC ubiquitination by reducing T58 phosphorylation (Supplementary Fig. 4l). To test whether expression of T58A alleviates the dependence of MYC function and gene expression on TRIB3, we applied CRISPR/Cas9 technology to establish *MYC*[T58A] mutant Raji cells with or without *TRIB3* depletion (Supplementary Fig. 5a), and further examined the function and gene expression profiling of these cells. The expression of phspho-T58 and total MYC were first examined in *TRIB3* depleted and/or *MYC*[T58A] Raji cells. Similar with previous observation, *TRIB3* depletion decreased total MYC expression and increased P-T58 relative to total MYC level. However, T58A mutation almost impeded *TRIB3*'s ability of reducing MYC expression (Supplementary Fig. 5b), suggesting that T58 phosphorylation is essential for the process of MYC expression regulated by *TRIB3* depletion. Compared with Raji cells with wild-type *MYC* (*MYC*[WT]), the *MYC*[T58A] mutant Raji cells exhibited enhanced proliferation (Supplementary Fig. 5c) as well as more and enlarged colonies (Supplementary Fig. 5d). Importantly, *TRIB3* knockdown in *MYC*[T58A] mutant Raji cells did not significantly influence the abilities of proliferation and colony formation (Supplementary Fig. 5c, d), suggesting that the *MYC*[T58A] mutant Raji cells were much less sensitive to *TRIB3* depletion compared with *MYC*[WT] Raji cells. In addition, we analyzed the gene expression profiles of *MYC*[T58A] cells with or without *TRIB3* deletion through RNA-seq. GSEA data showed enriched MYC pathway-related genes in *MYC*[T58A] lymphoma cells compared with *MYC*[WT] lymphoma cells (Supplementary Fig. 5e). Similar with the observation of functional experiment, GSEA analysis shows no statistically significant difference between the *MYC*[T58A] mutant Raji cells with and without *TRIB3* depletion (Supplementary Fig. 5f), suggesting that T58A mutation alleviated the dependence of MYC pathway-related gene expression on TRIB3. Taken together, these data indicated that K427 of MYC is critical for UBE3B-catalyzed MYC ubiquitination and the functions of TRIB3 in promoting lymphoma cell growth are dependent on T58 phosphorylation of MYC.

**TRIB3 interacts with MYC to impair the E3 ligase function of UBE3B.** To evaluate the biological consequences of UBE3B-mediated MYC ubiquitination and degradation, we examined the effect of the K427 mutation on MYC transactivation. The *K427R* mutation increased the transcriptional activity of MYC compared to WT *MYC* (Supplementary Fig. 6a). We next applied CRISPR/Cas9 technology to establish *MYC*[K427R] mutant Raji cell lines (Supplementary Fig. 6b), which were directly used to study *MYC*[K427R] function and activity without transfection of exogenous plasmids. The experiments of MYC ChIP-seq with spike-in normalization indicated that the K427 mutation increased the MYC occupancy at the promoters of its target genes (Fig. 4a, b and Supplementary Fig. 6c). Furthermore, *UBE3B* ablation increased MYC transcriptional activity, while *UBE3B* overexpression had the opposite effect (Supplementary Fig. 6d). qRT-PCR assays and GSEA showed that *UBE3B* overexpression suppressed the transcription of MYC target genes (Fig. 4c and Supplementary Fig. 6e). Inspection of multiple individual traces and global analyses showed that *UBE3B* overexpression globally reduced the association of RNAPII with promoters (Fig. 4d and Supplementary Fig. 6f). By examining the endogenous

interactions with Co-IP, we found that MYC, UBE3B, and MAX (also known as MYC-associated factor X) formed the heterotrimer in lymphoma cells (Supplementary Fig. 6 g). In the purified component system, MYC bound with UEB3B and MAX (Supplementary Fig. 6h), but no direct binding was detected between MAX and UBE3B (Supplementary Fig. 6i). UBE3B overexpression decreased the MYC/MAX interaction (Supplementary Fig. 6j), while the MYC K427R mutant bound more MAX than the WT MYC (Supplementary Fig. 6k). We examined *MYC*[K427R] biological function in the established *MYC*[K427R] mutant Raji cells with or without *Ad-UBE3B* infection. K427R mutation of MYC increased the proliferation of Raji cells, while *UBE3B* overexpression lost its ability of reduction in *MYC*[K427R] Raji cell number compared to that in *MYC*[WT] Raji cells (Supplementary Fig. 6l). These data indicate that UBE3B-mediated K427 ubiquitination of MYC interferes with the MYC/MAX interaction and plays an essential role in determining the biological output of UBE3B.

Compared with the *Ctrl* cells, *UBE3B*[OE] Raji cells developed fewer colonies (Fig. 4e) and exhibited reduced proliferation (Fig. 4f). *TRIB3* overexpression reversed the antiproliferative effects of *UBE3B* overexpression in Raji cells (Fig. 4f). However, *UBE3B* ablation also impeded Raji cell growth (Supplementary Fig. 6m). We further detected the cell death ratio of Raji cells with *UBE3B* deletion or *UBE3B* overexpression via Annexin V-APC/7-AAD staining. UBE3B overexpression did not induce apoptosis of Raji cells (Supplementary Fig. 6n). These data have indicated that the reduction in Raji cell number with UBE3B overexpression was due to reduced proliferation. We evaluated the effects of UBE3B and TRIB3 on lymphoma development in *Myc*[Eμ] mice (Fig. 4g). *Ube3b* overexpression reduced the enlarged spleens and LNs and reduced the number of Ki-67-positive cells in the spleens of lymphoma mice, which was reversed by TRIB3 overexpression (Fig. 4h–j). At the end of the experiment, BCLs were isolated from these mice and evaluated (Fig. 4k). During methylcellulose culture, BCLs from *Ad-Ube3b*[OE] mice yielded fewer colonies and were less capable of killing the recipient mice than BCLs isolated from *Ad-GFP* mice; however, *Trib3* overexpression diminished the antilymphoma effects of *Ube3b* overexpression (Fig. 4l, m). Overexpression of TRIB3 alone did not increase spleen weight, lymph node size or proliferation (Fig. 4h, i, j, l), which could be caused by endogenous levels of TRIB3 sufficient to suppress endogenous UBE3B activity. Additionally, knockout of *Ube3b* accelerated the death of *Myc*[Eμ] and *Myc*[Eμ]*Cre*[CD19]*Trib3*[F/+] mice (Fig. 4n, o and Supplementary Fig. 6o, p). Hence, the effects of *Ube3b* deletion on *Myc*[Eμ]*Cre*[CD19]*Trib3*[F/+] mice depended on reversing the reduced MYC expression and CFU number induced by *Trib3* knockout (Fig. 4p, q).

An in vitro ubiquitination assay showed that TRIB3 impeded MYC ubiquitination (Fig. 5a) and reduced the association of E3 with the MYC substrate but did not affect the association of UBE3B with UBCH3 (Fig. 5b, c). Indeed, TRIB3 interacted with MYC in Raji cells (Fig. 5d), which was supported by the colocalization of TRIB3/MYC in human lymphoma cells (Fig. 5e). MYC mainly interacted with the kinase domain (KD) of the TRIB3 C terminus (Fig. 5f). Using three main truncations of MYC (N-terminal domain, central region, and C-terminal domain), we found that the C- and N-terminal domains, but not the central region (CT) of MYC mediated the interaction of MYC with either TRIB3 or UBE3B (Fig. 5g, h left). Among them, the C-terminal domain of MYC showed a major, but the N-terminal domain showed a minor contribution to its binding with TRIB3 or UBE3B (Fig. 5g, h, left). In addition, we performed the reverse Co-IP assay with anti-MYC truncation antibodies (anti-GFP) and examined the levels of TRIB3 of UBE3B in the IP samples. Similarly, the C-terminal MYC possessed the strongest

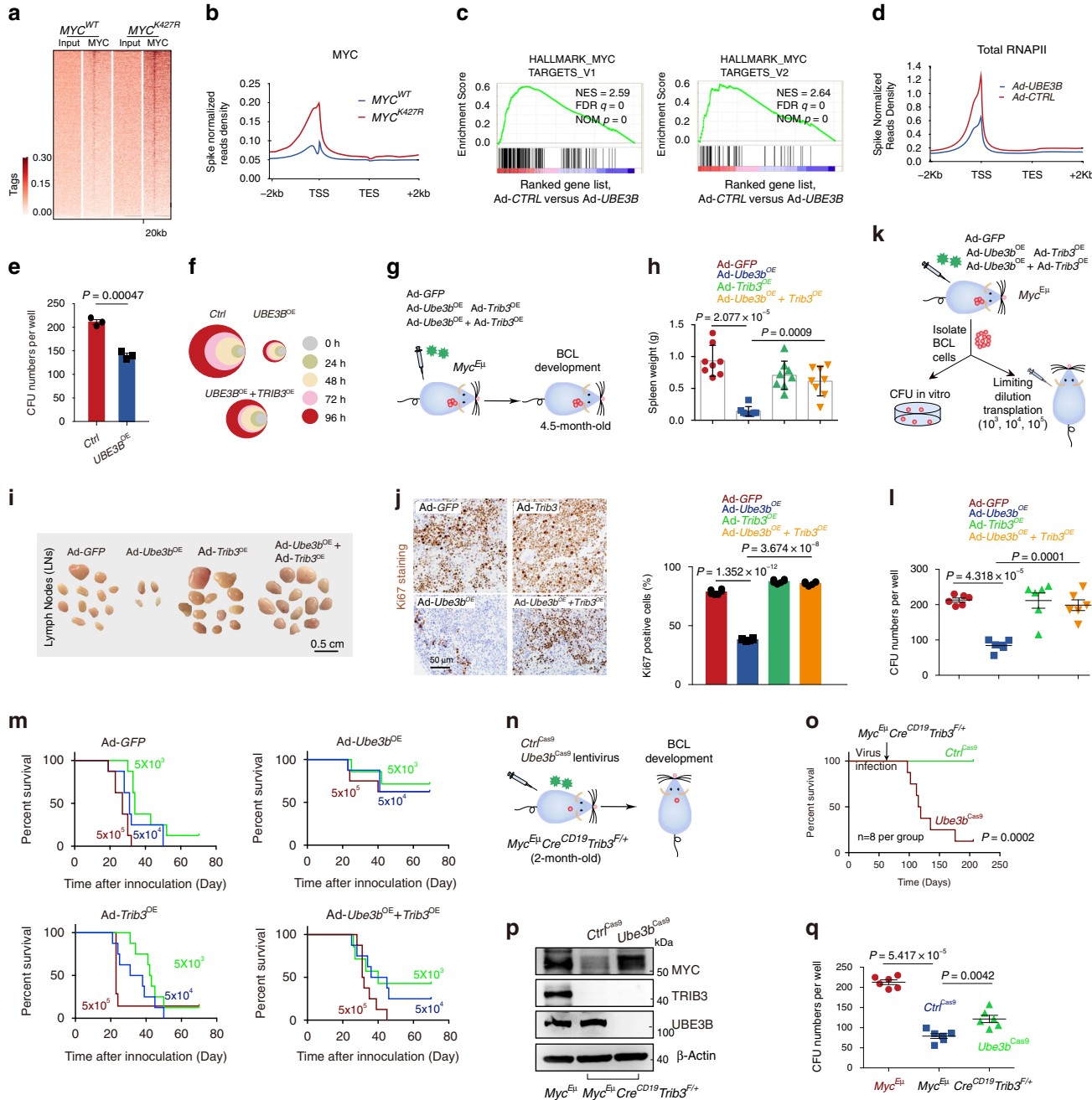

binding affinity to TRIB3 or UBE3B, while the N-terminal MYC weakly bind to TRIB3 or UBE3B, but the CT domain of MYC did not exhibit any detectable level of binding (Fig. 5g, h, right). Thus, the TRIB3-MYC interaction interferes with the binding of MYC and UBE3B (Fig. 5i, j), likely due to TRIB3 occupying these MYC domains.

Given that the C terminal domain of MYC also interacted with MAX, we examined the effect of TRIB3 on the MYC/MAX complex. Overexpression of TRIB3 increased the binding of MYC and MAX, as indicated by the Co-IP and surface plasmon resonance (SPR) assays (Fig. 5k, l). Further, MYC, TRIB3, and MAX formed the heterotrimer in lymphoma cells (Fig. 5m). Moreover, restored expression of full-length TRIB3 or the KDC domain of TRIB3 reversed the repressed proliferation of lymphoma cells induced by *TRIB3* deletion (Fig. 5n), indicating that the KDC domain of TRIB3 interacts with MYC, which stabilizes MYC and enhances lymphoma cell proliferation.

**The TRIB3-MYC interaction confers a therapeutic target against lymphoma.** Four α-helical peptides (CM1, CM2, CM3, and CM4) derived from C terminus of MYC were screened using the predictive I-TASSER server (Supplementary Fig. 7a), and only CM4 displayed a high binding affinity for TRIB3 (KD = 76.4 nM; Fig. 6a and Supplementary Fig. 7b, c). Homology modeling based on the TRIB1 crystal structure[42] predicted a hydrogen bond between Glu344 (E344) of the TRIB3 protein and the CM4 peptide (Supplementary Fig. 7d), suggesting that the C terminus of TRIB3 is the binding domain of CM4 (Fig. 6b). We further performed sequence alignment within the C termini of TRIB family members and found that E344 of TRIB3 is not conserved at the same positions on TRIB1 and TRIB2 (Supplementary Fig. 7e), indicating that CM4 may be specific for TRIB3. Furthermore, CM4 itself, but not the scramble peptide of CM4, decreased the binding affinity of TRIB3 and MYC in vitro (Fig. 6c). Because the LZ and HLH domains of MYC bind to

**Fig. 4 TRIB3 suppresses the functions of the UBE3B:MYC axis to attenuate lymphomagenesis. a** Occupancy heatmap of genome-wide MYC peaks in $MYC^{WT}$ and $MYC^{K427R}$ Raji cells in a window of ±10 kb surrounding the transcription start site (TSS). **b** Distribution of MYC tags in $MYC^{WT}$ and $MYC^{K427A}$ Raji cells. Binding is shown for a window of ±2 kb around MYC tags as reference coordinates. **c** GSEA shows global downregulation of MYC target genes in Ad-UBE3B versus Ad-CTRL Raji cells. **d** Metagene plots of global RNAPII occupancy at gene bodies in Raji cells with or without UBE3B overexpression. **e** Colony-forming capacity of Raji cells with or without UBE3B overexpression. The data are presented as the means ± S.E.M from three independent experiments. Statistical significance was determined by two-tailed Student's t test. P value: 0.00047. **f** Relative cell viabilities of Raji cells with overexpression of the indicated genes for the indicated times. The colors represent different time points; the diameter indicates the relative cell viability. The data are presented as the means ± S.E.M from three independent experiments. **g** Schematic strategy for studying BCL development in $Myc^{Eμ}$ mice with the indicated infection of Ad-GFP, Ad-Ube3b, Ad-Trib3, or Ad-Ube3b plus Ad-Trib3 in vivo. **h** Data are the spleen weights of $Myc^{Eμ}$ mice (n = 8 per group) with overexpression of the indicated genes. Data are represented as means ± SEM. Statistical significance was determined by two-tailed Student's t test. P values: $2.077 \times 10^{-5}$, 0.0009. **i** Data are representative LNs in $Myc^{Eμ}$ mice overexpressing Ad-GFP, Ad-Ube3b, Ad-Trib3, or Ad-Ube3b plus Ad-Trib3. Scale bar, 0.5 cm. **j** Cell proliferation was determined by Ki67 staining in the spleens of the indicated mice (n = 6 per group). Data are representative images of Ki67 staining and statistical analyses of Ki67-positive cells. Data are represented as means ± SEM. Statistical significance was determined by two-tailed Student's t test. P values: $1.352 \times 10^{-12}$, $3.674 \times 10^{-8}$. Scale bar, 50 μm. **k** Approaches to defining the self-renewal ability of mouse LCs from the indicated mice (5 months old). **l** Colony-forming ability of BCL cells isolated from $Myc^{Eμ}$ mice (n = 6 per group) overexpressing Ad-GFP, Ad-Ube3b, Ad-Trib3, or Ad-Ube3b plus Ad-Trib3. Data are represented as means ± SEM. Statistical significance was determined by two-tailed Student's t test. P values: $4.318 \times 10^{-5}$, 0.0001. **m** The tumorigenicity of BCL cells from the indicated mice was compared by limiting dilution transplantation into NSG mice; n = 8 mice per group. **n** Schematic strategy for studying BCL development in $Myc^{Eμ}CD19^{Cre}Trib3^{F/+}$ mice infected with or without lentivirus $Ube3b^{Cas9}$. **o** Kaplan–Meier survival curves for $Myc^{Eμ}CD19^{Cre}Trib3^{F/+}$ mice infected with or without lentivirus $Ube3b^{Cas9}$ (n = 8 per group). Statistical difference was determined by two-sided log-rank test. P value: 0.0002. **p** Expression of MYC in the spleens of $Myc^{Eμ}CD19^{Cre}Trib3^{F/+}$ mice (5 months old) infected with or without lentivirus $Ube3b^{Cas9}$. The data are presented as presentative from three independent experiments. **q** Colony-forming ability of BCL cells isolated from the LNs of the indicated mice (n = 6 per group). The data are representative or the means ± SEM of three assays. Statistical significance was determined by two-tailed Student's t test. P value: $5.417 \times 10^{-5}$, 0.0042. The $Myc^{Eμ}Cre^{CD19}Trib3^{F/+}$ mice are heterozygous for the $Myc^{Eμ}$ allele. Source data are provided as a Source Data file.

MAX, we detected the abilities of these peptides to bind MAX, and CM1, but not CM2, CM3, or CM4, showed a high ability to bind MAX and decreased the binding of MYC and MAX (Supplementary Fig. 7f, g).

To determine whether the CM4 peptide disturbs the TRIB3-MYC interaction in lymphoma cells, a fused peptide, PCM4, was generated by linking CM4 to a cell-penetrating peptide[43]. High-resolution structured illumination microscopy (SIM) showed the TRIB3-MYC interaction in lymphoma cells treated with a penetrated control peptide (PCON), whereas PCM4 inhibited the colocalization and interaction of TRIB3 with MYC (Fig. 6d and Supplementary Fig. 7h) and rescued the interaction of MYC and UBE3B caused by TRIB3 expression (Fig. 6e, f and Supplementary Fig. 7i). Off-target effects of PCM4 were examined because TRIB3 has been shown to interact with AKT, p62, COP1, and PML[16–18]. PCM4 did not disturb the interactions of TRIB3 with these proteins (Supplementary Fig. 7j), indicating that PCM4 specifically interrupts the TRIB3-MYC interaction. PCM4 rescued MYC ubiquitination and degradation (Fig. 6g–i and Supplementary Fig. 7k) and reduced the MYC abundance (Fig. 6j and Supplementary Fig. 7l) in lymphoma cells. Furthermore, we examined the effects of PCM4 on the colocalization and the interaction of MYC and MAX through confocal and CO-IP assays, respectively. The PLA assay showed the reduced foci of MAX-MYC colocalization in PCM4-treated Raji cells (Supplementary Fig. 7m). Interestingly, in the examination of PCM4's effects on the interaction of MYC and MAX by CO-IP assay, if sufficient MAX antibody (5 μg) was added to the lysates to capture the MAX protein, the quantity of the precipitated MYC protein in control cells was much more than that in PCM4-treated cells (Supplementary Fig. 7n, left). However, if less anti-MYC antibody (1 μg) was added to capture the same quantity of MYC protein in the CO-IP assay, and the quantity of the precipitated MAX protein in control cells was identical with that in PCM4-treated cells (Supplementary Fig. 7n, right). Furthermore, we used the NanoBRET™ assay to identify whether CM4 interferes with the MYC:MAX protein interaction in a dose-dependent manner using the respective proteins tagged with NanoLuc® luciferase (NanoLuc®) and HaloTag® ligands.

With 10058-F4 as a PPI inhibitor of MYC-MAX interaction, we found that CM4 showed no effects on the MYC/MAX heterodimer in this assay (Supplementary Fig. 7o). In addition, quantification of the MYC/MAX heterodimer using Proximity ligation assay (PLA) has been performed in the CM4-treated Raji cells with proteasome inhibitor MG132 reversing MYC degradation. CM4 treatment indeed did not affect the colocalization of MYC and MAX in MG132-treated Raji cells (Supplementary Fig. 7p). Overall, our results suggested that the low amount of MYC/MAX complex accounts for the reduced MYC transcriptional output, and CM4 shows specificity to MYC and TRIB3 but not to the MYC and MAX.

Consistent with these findings, PCM4-reduced MYC transcriptional activity (Fig. 6k). qRT-PCR and RNA-seq showed that PCM4 suppressed the transcription of MYC target genes (Fig. 6l and Supplementary Fig. 7q). Inspection of multiple individual traces and global analyses of RNAPII function showed that PCM4 globally reduced the association of RNAPII with promoters of its transcribed genes (Fig. 6m, n). Additionally, PCM4 reduced the intrinsic clonogenicity of human lymphoma cells (Fig. 6o and Supplementary Fig. 7r, s) and primary mouse BCL cells (Supplementary Fig. 7t) and inhibited the proliferation of most lymphoma and leukemia cells in a time-dependent manner (Fig. 6p). Interestingly, PCM4-sensitive cells expressed TRIB3 and MYC at higher levels than PCM4-resistant cells (Fig. 6q). The combination of PCM4 with doxorubicin (DOX), a chemotherapeutic agent for the treatment of lymphoma, synergistically inhibited Raji cell proliferation (Fig. 6r). These data indicate that disturbing the TRIB3-MYC interaction reduces MYC expression and blocks the lymphoma-promoting effects of MYC in vitro.

We examined the anti-BCL effect of PCM4 in $Myc^{Eμ}$ mice (Fig. 7a). On day 90 after treatment, dramatic reductions in the spleen/LN weights and numbers of Ki-67-positive cells in the spleens were observed in mice treated with PCM4 compared to those treated with PCON (Fig. 7a, d). Indeed, the secondary recipients from the PCM4 group had a better survival rate than those from the PCON group (Fig. 7e). The PCM4 group exhibited a decrease in the TRIB3-MYC interaction and MYC expression in the LNs and spleens of $Myc^{Eμ}$ mice (Fig. 7f, g). Additionally,

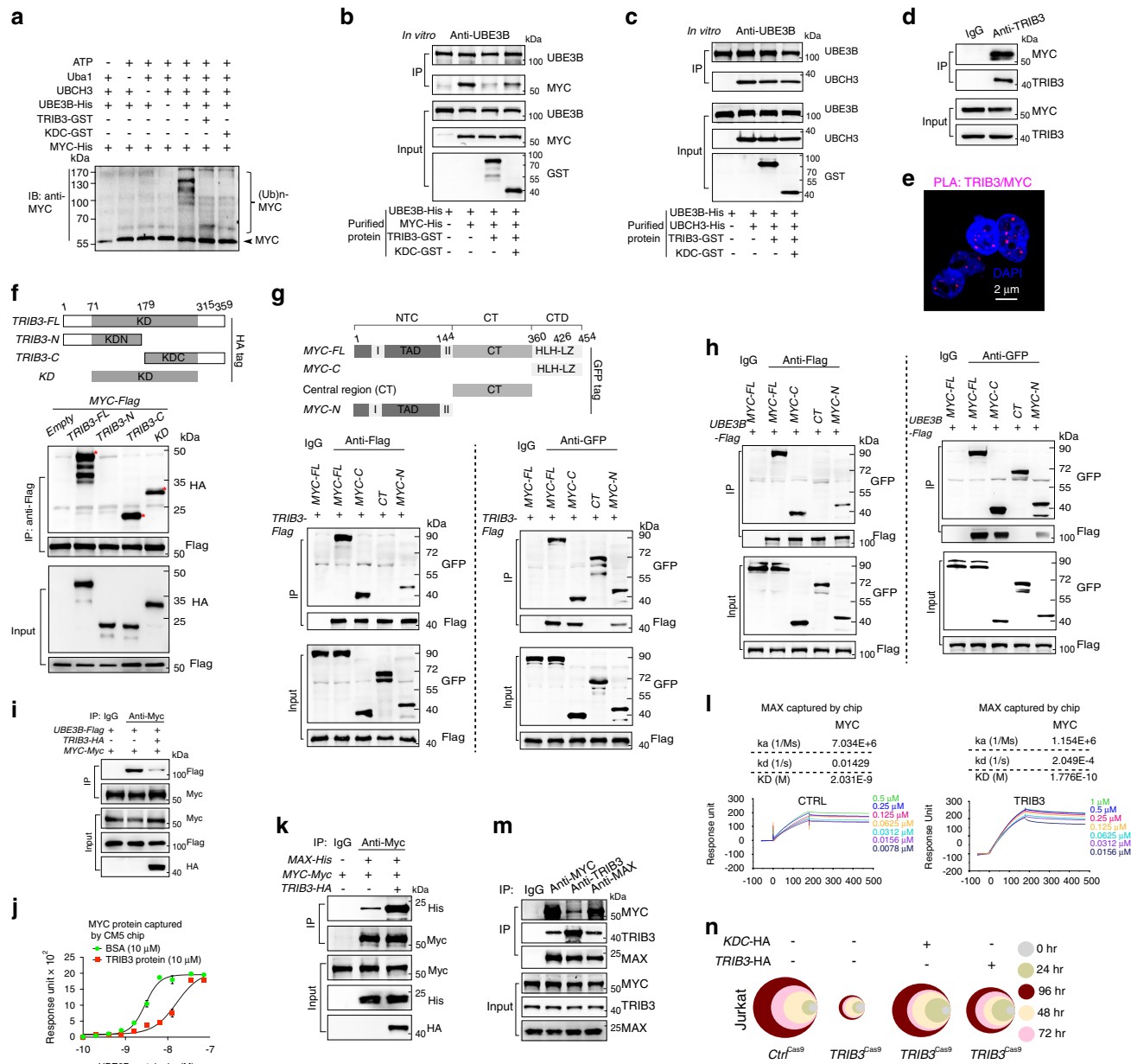

**Fig. 5 TRIB3 interacts with MYC to inhibit the UBE3B:MYC axis. a** In vitro ubiquitination assays show that the TRIB3 or TRIB3-KDC protein decreased polyubiquitination of MYC mediated by UBE3B and UBCH3. **b** The TRIB3 protein or the KDC domain of TRIB3 decreased the interaction of MYC and UBE3B in the purified system. **c** Neither the TRIB3 protein nor the KDC domain of TRIB3 affected the interaction of UHCH3 and UBE3B in the purified system. **d** The interaction of TRIB3 and MYC in Raji cells was evaluated by Co-IP assays. **e** Colocalization of TRIB3 and MYC in Raji cells was detected with immunostaining (PLA assay). Data are representative images from three independent experiments. Scale bar, 2 μm. **f** Mapping TRIB3 regions binding to MYC. Top: deletion mutants of TRIB3. Bottom: HEK 293T cells were cotransfected with the indicated constructs of TRIB3 (HA tag) and MYC (Flag tag). Cell extracts were IP with an anti-Flag Ab. **g**, **h** Mapping MYC regions binding to TRIB3 (**g**) or UBE3B (**h**). Top: deletion mutants of MYC. Bottom: HEK293T cells were cotransfected with the indicated constructs of MYC and TRIB3 or UBE3B. Cell extracts were IP with an anti-Flag Ab. **i** TRIB3 overexpression interfered with the interaction of UBE3B and MYC. HEK 293T cells were cotransfected with the indicated constructs. Cell extracts were IP with an anti-MYC Ab. **j** TRIB3 decreased the binding of UBE3B and MYC. Kinetic interactions of the UBE3B and MYC proteins were determined by SPR analyses with the BSA or TRIB3 proteins. **k** TRIB3 overexpression increased the interaction of MAX and MYC. HEK 293T cells were cotransfected with the indicated constructs. Cell extracts were IP with an anti-MYC Ab. **l** TRIB3 overexpression increased the binding of MAX and MYC. Kinetic interactions of the MAX and MYC proteins were determined by SPR analyses with the CTRL (GST) or TRIB3 (TRIB3-GST) protein. **m** The heterotrimers of MYC, TRIB3, and MAX in Raji cells were detected by Co-IP assays. Cellular extracts were IP with mouse IgG as a negative control or anti-MYC, anti-TRIB3, or anti-MAX antibodies. Western blots were performed with anti-MYC, anti-TRIB3, and anti-MAX antibodies. **n** Recovery of the *TRIB3*-HA wild-type or *KDC*-HA mutant enhanced the decreased cell viabilities of *TRIB3*[Cas9] Jurkat cells for the indicated times. The colors represent different time points; the diameter indicates the relative cell viability. Source data are provided as a Source Data file.

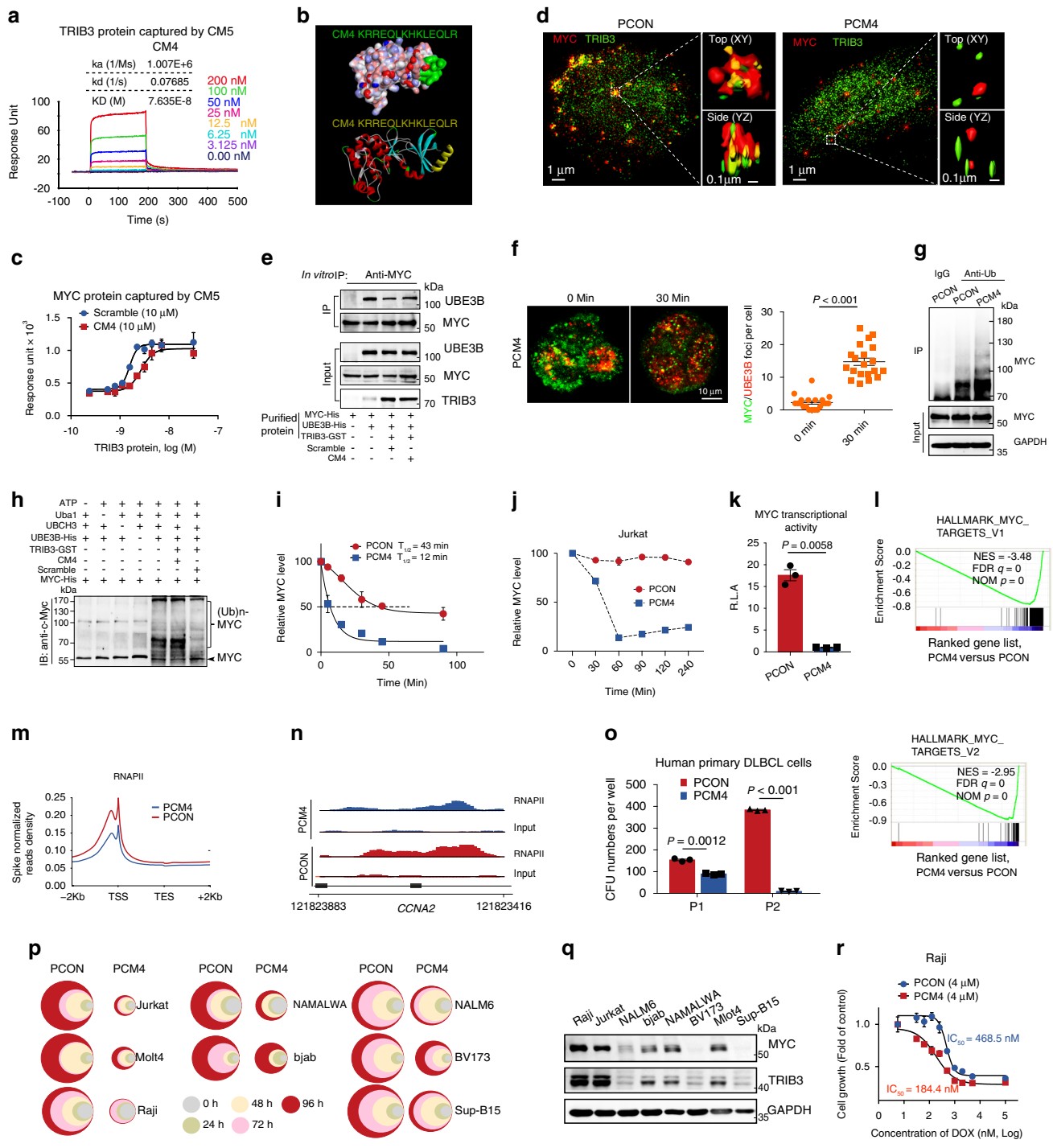

PCM4 did not exert obvious toxic effects in these mice, as indicated by renal/liver function assays (Supplementary Fig. 8a, b). The combination of PCM4 and DOX enhanced the therapeutic efficacy of DOX, as indicated by the enhanced survival rate of $Myc^{E\mu}$ BCL mice treated with PCM4 and DOX (Fig. 7h). These data indicate that suppressing the TRIB3-MYC interaction and MYC expression attenuates the lymphoma-promoting effects of MYC in vivo.

We also examined the effects of PCM4 on established lymphoma PDX models derived from 10 patients with DLBCL (T64, T69, T169, T156, and T1), PTCL (T144), follicular lymphoma (FL; T151 and T174), T cell lymphoblastic lymphoma

(T-LBL; T123) and TCL (T3; Fig. 8a, Supplementary Fig. 8c, and Table 1). PCM4 inhibited the tumor growth of PDXs (T64, T69, T144, T123, T174, T169, and T156) with high TRIB3 and MYC expression compared with PDXs (T151, T1, and T3) with low TRIB3 and MYC expression (Fig. 8b–h). The combination of PCM4 and DOX further decreased the growth and weights of tumors in PDX mice with high TRIB3 and MYC expression (Fig. 8b–i). In comparison with the benign LN tissues, most lymphoma specimens showed high expression of TRIB3 but not UBE3B (Fig. 8j and Supplementary Fig. 8d). Notably, a positive correlation was observed between TRIB3 and MYC expression (Fig. 8k, l). These data indicate that TRIB3 plays a critical role in

**Fig. 6 Disturbing the TRIB3/MYC interaction destabilizes MYC and inhibits lymphoma. a** The kinetic interaction of CM4 and TRIB3 was determined by surface plasmon resonance (SPR) analyses. **b** The highest scoring Dock model of the CM4 and TRIB3 complex is shown. Upper: the surface of CM4 (green) and the TRIB3 complex. Below: the 3D structure of CM4 (yellow) and the TRIB3 complex. **c** The kinetic interaction of the TRIB3 and MYC proteins was determined by SPR analyses with or without CM4. **d** Structured illumination microscopic (SIM) images of PCON- or PCM4-treated Raji cells (30 min) stained for MYC and TRIB3. Dotted line border square: area processed for 3D surface rendering (insets). **e** In vitro interaction assays show that the CM4 peptide increased the binding of MYC and UBE3B. **f** Representative images of MYC and UBE3B foci (left) and quantification of the number of MYC/UBE3B colocalized foci in Raji cells before and after PCM4 treatment. Scale bar, 10 μm. Data are represented as means ± SEM. Statistical significance was determined by two-tailed Student's *t* test. *P* value: $3.74 \times 10^{-13}$. **g** Effect of PCM4 on MYC ubiquitination. Extracts of PCON- and PCM4-treated Raji cells were IP with an anti-Ub Ab. Ubiquitinated MYC was detected by immunoblotting. **h** In vitro ubiquitination assays show that the CM4 peptide restored the polyubiquitination of MYC. **i** Effect of PCM4 on MYC protein degradation. Raji cells were treated with CHX (10 μg/mL) and the PCON or PCM4 peptide (4 μm) for the indicated times. The protein abundance of MYC was detected by immunoblotting. **j** Effect of PCM4 treatment on MYC protein levels. Jurkat cells were treated with PCON or PCM4 for the indicated times (4 μm). The protein expression of MYC was detected by immunoblotting. **k** The transcriptional activity of MYC in Raji cells treated with PCON or PCM4. PCON- or PCM4-treated Raji cells were transfected with the reporter genes with MYC transcriptional activity. After 24 h of transfection, luciferase activities were measured. Data are represented as means ± SEM. Statistical significance was determined by two-tailed Student's *t* test. *P* value: 0.0058. **l** GSEA shows global downregulation of *MYC* target genes in PCM4-treated versus PCON-treated cells. **m** Metagene plots of global RNAPII occupancy at gene bodies in PCON- or PCM4-treated Raji cells. **n** ChIP-sequencing tracks for *CCNA2* from PCM4-treated cells versus PCON-treated cells normalized to spike-in controls. **o** PCM4 decreased the serial colony formation ability (P1 and P2) of primary human DLBCL cells (T69). Data are represented as means ± SEM. Statistical significance was determined by two-tailed Student's *t* test. *P* value: 0.0012, $3.403 \times 10^{-5}$. **p** Effects of PCM4 on the cell viabilities of the indicated lymphoma and leukemia cells for the indicated times. The colors represent different time points; the diameter indicates the relative cell viability. **q** Expression of MYC and TRIB3 in the indicated cells. **r** The synergetic effect of PCM4 and DOX on the cell viability of Raji cells. Source data are provided as a Source Data file.

MYC-driven lymphoma by interacting with MYC and that disturbing this interaction in combination with DOX treatment results in potent antilymphoma efficacy (Fig. 9).

## Discussion

Although MYC rearrangements partially explain the high MYC expression in all BLs and non-Burkitt high-grade BCLs, there is no consensus regarding the causes of high MYC levels in lymphomas without MYC rearrangement. In this study, we observed that TRIB3 expression was elevated in more than 50% of human lymphoma cases. Using transgenic mice with *Trib3* deletion in different cell lineages, we found that TRIB3 participated in MYC-driven lymphoma tumorigenesis and pathogenesis. *TRIB3* knockout suppresses the cell proliferation and self-renewal ability of MYC-associated lymphoma by reducing MYC protein stability and abundance in these cells. Our study thus provides insights into the pathogenesis of *MYC*-associated lymphomagenesis.

*Trib3* deletion in B lymphocytes, but not in myeloid cells or T lymphocytes, inhibited lymphoma development. Several mechanisms may account for *TRIB3* ablation-induced lymphoma inhibition. First, *TRIB3* depletion restores UBE3B-mediated MYC ubiquitination and degradation, which suppresses MYC expression and MYC-related oncogenic activities. Second, *TRIB3* depletion increases the activity of the autophagy and ubiquitin proteasome system (UPS), p53-dependent senescence, PML nuclear body formation and regulation of cancer metabolism, which may collaborate to participate in the inhibition of lymphomagenesis[17,18]. Third, *Trib3* deletion decreases TRIB2 expression in lymphocytes, while TRIB2 functions as an oncogene in acute myeloid lymphoma (AML) by regulating C/EBPα expression and degradation[44-46]. Thus, *Trib3* depletion-reduced TRIB2 expression in lymphoma cells may contribute to lymphoma pathogenesis in $Myc^{E\mu}Cre^{CD19}Trib3^{F/+}$ mice. TRIB2 was reported to act downstream of Wnt/TCF, and Wnt/β-catenin activation was shown to induce TRIB2 expression in cancer cells[47]. Our recent study indicated that *TRIB3* depletion decreased β-catenin/TCF transcriptional activity by reducing the recruitment of TCF and β-catenin to the promoter region of genes regulated by Wnt in colon cancer cells[22]. Hence, we presumed that the decreased protein expression of TRIB2 is caused by *TRIB3* knockout-reduced *TRIB2* transcription through suppressing the activity of Wnt/β-catenin in lymphoma cells. Obviously, it is worth further dissecting the molecular mechanism accounting for the TRIB3 regulation of TRIB2 expression.

The MYC level is tightly controlled by E3 ligase-regulated protein degradation. FBXW7 is the most important E3 ligase for MYC ubiquitination and proteasomal degradation in hematologic malignancies, which is dependent on T58 phosphorylation of MYC[41,48]. However, mutation of T58 to a nonphosphorylatable residue stabilizes MYC more than the ablation of FBXW7[49], suggesting that additional E3 ligases also stimulate the proteolysis of MYC T58 phosphorylation. Indeed, we found that UBE3B, a newly identified E3 ligase of MYC, drives the proteolytic turnover of MYC dependent on T58 phosphorylation. TRIB3 sustained MYC stability specifically via UBE3B but not via other known E3 ligases. Indeed, UBE3B overexpression reduces MYC abundance and suppresses its transcriptional activity, resulting in lymphoma inhibition. Interestingly, our work indicates that UBE3B may not represent a viable drug target because *UBE3B* deletion in lymphoma cells decreases tumor cell proliferation.

Loss of UBE3B function causes Kaufman oculocerebrofacial syndrome, a rare developmental disorder[50]. The relationship between UBE3B deregulation and cancer has not yet been reported or validated. Indeed, we did not find altered UBE3B expression in human lymphoma samples compared to benign LNs. Thus, high MYC expression may be mainly caused by a decreased interaction of UBE3B and MYC, which is induced by elevated TRIB3 expression in lymphoma cells. Moreover, a loss-of-function mutation of UBE3B (R346Q) is found in MYC-associated lymphoma patients and impedes MYC degradation. Notably, *TRIB3* knockdown decreases MYC expression in most blood cancer cells but not in U937 AML cells or bjab BL cells. One possible reason is that the E3 ligase FBXW7 or SKP2, but not UBE3B, mediates MYC ubiquitination and degradation in these cells. Another reason may be that these cells express very low levels of UBE3B. Thus, it is worth examining the expression of MYC-related E3 ligases when MYC expression is evaluated in *TRIB3*-depleted cancer cells.

Deregulation of *MYC* is a pervasive finding in lymphoma that causes tumor progression and a poor prognosis, and MYC is obviously a good therapeutic target in lymphoma. Although the development of small molecules inhibiting MYC activity has been

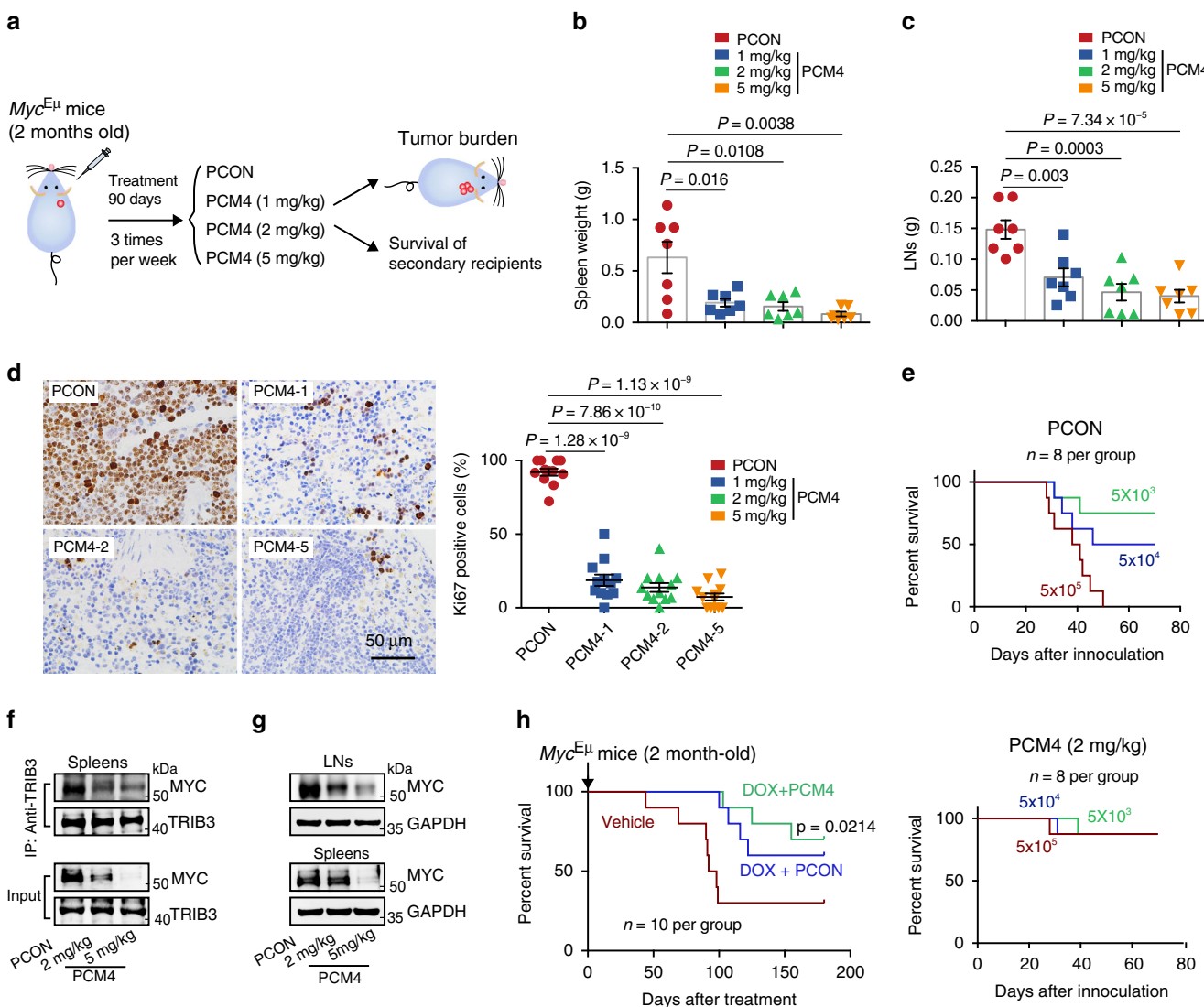

**Fig. 7 Disturbing the TRIB3/MYC interaction attenuates lymphoma in mice. a** The strategy for investigating the anti-BCL effects of PCM4 on $Myc^{E\mu}$ mice in vivo. **b**, **c** The data represent the statistical analyses of spleen and LN weights in $Myc^{E\mu}$ mice (5 months old; $n = 7$ per group) treated with the indicated agents. Data are represented as means ± SEM. Statistical significance was determined by two-tailed Student's $t$ test. $P$ value: 0.016; 0.0108, and 0.0038 (**b**); 0.003, 0.0003, 7.34 × 10$^{-5}$ (**c**). **d** Representative images of Ki67 staining (left) and statistical analyses of Ki67-positive cells (right) in the spleens of $Myc^{E\mu}$ mice (5 months old; $n = 12$ per group) treated with the indicated agents. Scale bar, 50 µm. Data are represented as means ± SEM. Statistical significance was determined by two-tailed Student's $t$ test. $P$ value: 1.28 × 10$^{-9}$, 7.86 × 10$^{-10}$, and 1.13 × 10$^{-9}$. **e** The tumorigenicity of BCL cells from the indicated mice was compared by limiting dilution transplantation into NSG mice; $n = 8$ mice per group. **f** PCM4 disturbed the TRIB3/MYC interaction in vivo. Spleen lysates from the indicated mice (5 months old) were IP with an anti-TRIB3 Ab and blotted with an anti-MYC Ab. The data are representative of three independent assays. **g** PCM4-reduced MYC expression. LN and spleen extracts from the indicated mice (5 months old) were blotted with an anti-MYC Ab. The data are presented as representative from three independent experiments. **h** Kaplan–Meier survival curves for $Myc^{E\mu}$ BCL mice treated with the indicated agents ($n = 10$ per group). Statistical difference was determined by two-sided log-rank test. $P$ value: 0.0214 (DOX + PCM4 vs. DOX + PCON). Source data are provided as a Source Data file.

difficult, several MYC-directed therapies, including bromodomain, extraterminal (BET) inhibitors, and the Omomyc peptide, have been assessed in preclinical settings and in clinical trials (NCT01713582) involving hematological malignancies with durable responses[51,52]. However, acquired resistance will limit responsiveness to BET inhibitor treatment through the rewiring of transcriptional programs or activation of the WNT pathway[53,54]. In this study, we found that lymphoma cells with high TRIB3 and MYC expression were more sensitive to PCM4 than those with low TRIB3 or MYC expression and that PCM4 increased the sensitivity of lymphoma cells to DOX due to decreased MYC expression. Notably, the CM4 peptide

corresponds to aa 436–450 of the LZ region of MYC, which is the MAX-binding region domain. However, CM4 specifically bound TRIB3 and subsequently disturbed the MYC-TRIB3 and MYC-MAX interactions. The PCM4-reduced interaction and colocalization of MYC-MAX in lymphoma cells may largely be caused by a reduction in MYC abundance induced by PCM4. Although the short-term toxicity of this peptide is not obvious in lymphoma mice, the long-term toxicity and target specificity of PCM4 need to be further investigated in normal proliferating tissues. Moreover, PCM4 is not a viable option for human trials due to its intrinsic weaknesses, including poor physicochemical stability and a short circulating plasma half-life. Hence,

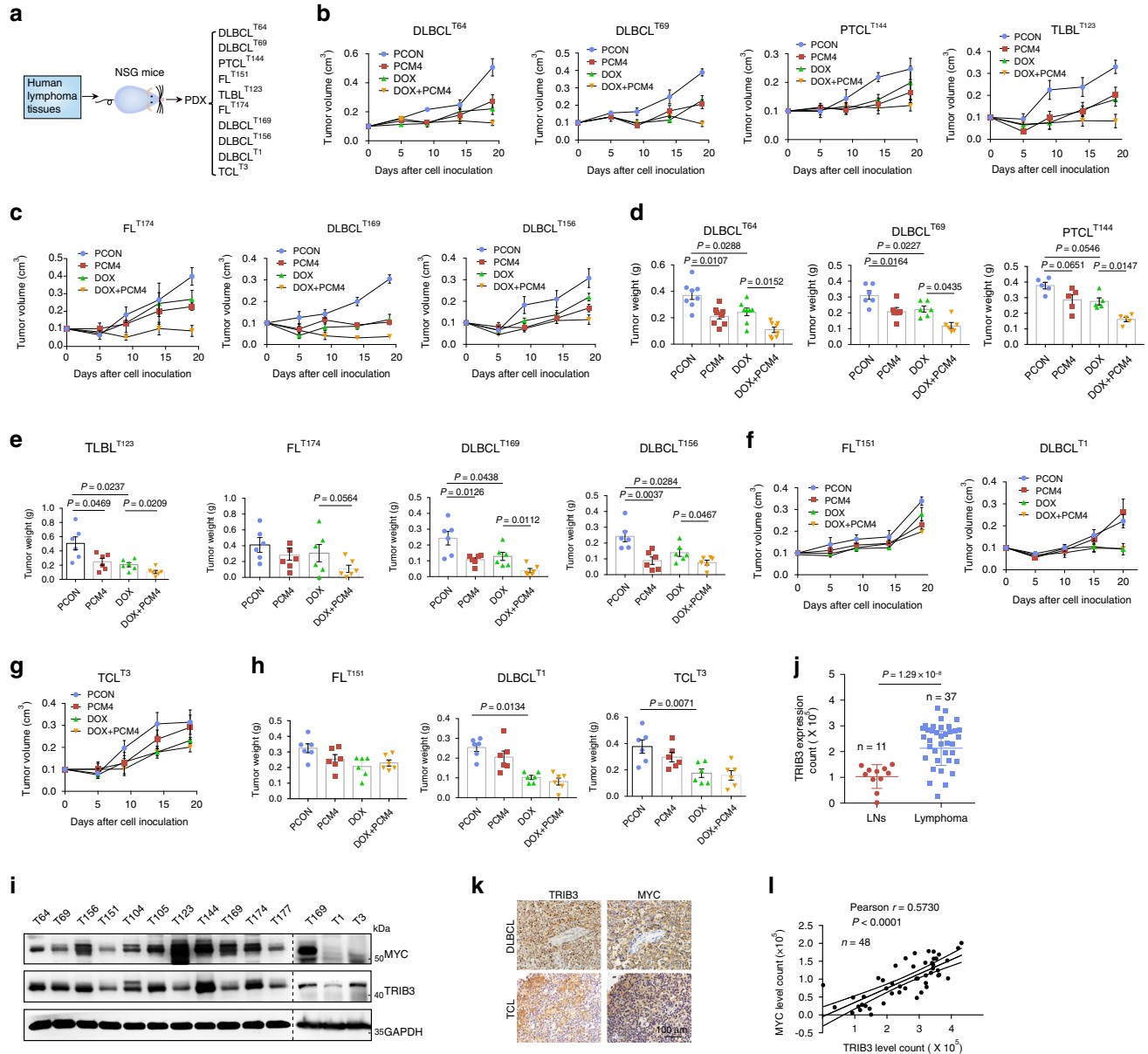

**Fig. 8 Combination of PCM4 with doxorubicin inhibits synergistically lymphoma in PDX models. a** Strategy for constructing patient-derived tumor xenograft (PDX) models from lymphoma patients. **b, c** Effects of the indicated treatments on tumor growth in the indicated PDX models. Tumors were subcutaneously engrafted in NSG mice ($n = 8$ per group). One day later, the mice were treated with PCM4 (5 mg/kg) or/and DOX (2 mg/kg) for 21 days, and the tumors were measured twice a week. Data are represented as means ± SEM. **d, e** Effects of the indicated treatments on the tumor weights in the indicated PDX models. (DLBCL[T64] $n = 8$ per group, DLBCL[T69] $n = 6$ per group, PTCL[T144] $n = 5$ per group, TLBL[T123] $n = 6$ per group, FL[T174] $n = 6$ per group, DLBCL[T169] $n = 6$ per group, DLBCL[T156] $n = 6$ per group). Data are represented as means ± SEM. Statistical significance was determined by two-tailed Student's $t$ test. $P$ values were indicated in the panels **d** and **e**. **f–h** Effects of the indicated treatments on the tumor growth and tumor weights in the indicated PDX models. (FL[T151] $n = 6$ per group, DLBCL[T1] $n = 6$ per group, TCL[T3] $n = 6$ per group). Data are represented as means ± SEM. $P$ values were indicated in the panel **h**. **i** The expression of MYC and TRIB3 in 13 primary lymphoma cells was detected by western blotting. The dashed line showed the two gels which were probed in parallel. The data are representative of three independent assays. **j** Statistical analyses of IHC staining of TRIB3 in human lymphoma ($n = 37$ samples) and benign lymph node (LN) tissues ($n = 11$ samples). TRIB3 expression was assessed as integrated optical density (IOD) scores. The horizontal lines shown in the graphs represent the median of each protein expression level. The data are presented as the means ± SEM. Statistical significance was determined by two-tailed Student's $t$ test. $P$ value: $1.29 \times 10^{-8}$. **k** Representative images of IHC staining of TRIB3 and MYC in human lymphoma tissues. The data are representative of three independent assays. Scale bars, 100 μm. **l** Correlation between TRIB3 and MYC expression in human LNs and lymphoma specimens ($n = 48$). Each data point represents the value from an individual patient. Statistical significance was measured by Pearson's correlation test. Source data are provided as a Source Data file.

stabilization of the α-helix peptide, substitution of amino acids and other modifications to improve the physicochemical properties of PCM4 are further needed to obtain the desired druggable properties.

In summary, our study indicates that TRIB3 is a reasonable molecular target for the treatment of high MYC-expressing lymphoma and that interruption of the MYC-TRIB3 and MYC-MAX interactions has therapeutic potential for this lymphoma.

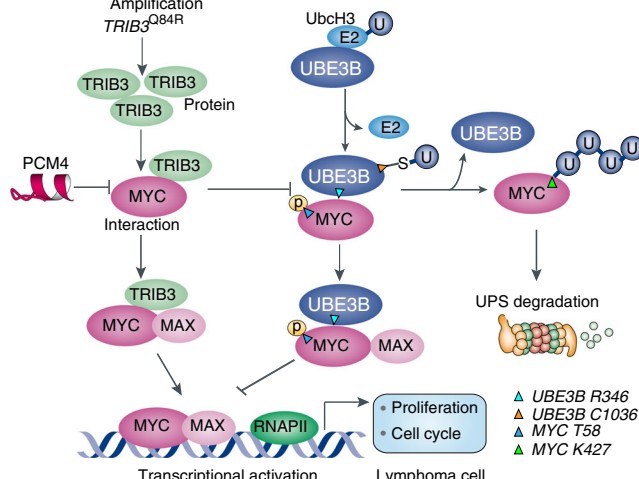

**Fig. 9 Schematic diagram illustrates the TRIB3-UBE3B-MYC axis driving lymphoma development and progression.** *TRIB3* expression is enhanced in human lymphoma cells due to either the increased copy number in response to various stresses and nonsynonymous (Q84R) mutation of *TRIB3*. Whereas, UBE3B, an E3 ubiquitin ligase, and UBCH3, an E2 ubiquitin-conjugated enzyme, have been identified to target MYC for proteasomal degradation. The enhanced TRIB3 interacts with MYC to decrease the UBE3B-mediated ubiquitination of MYC K427 and increase MYC stability; this interaction also supports the formation of MYC and MAX heterodimer. These actions enhance MYC transcriptional activity, causing high proliferation and self-renewal of lymphoma cells. Interfering with the TRIB3/MYC interaction enhances MYC degradation and suppresses the development and progression of high TRIB3/MYC-associated lymphoma.

Furthermore, our results raise the intriguing possibility of targeting this interaction to treat other MYC-associated malignancies by reducing MYC abundance.

## Methods

**Animal studies.** NOD-*scid* IL2Rg$^{null}$ (NSG) mice (4–6 weeks old, male) were purchased from the Nanjing Biomedical Research Institute of Nanjing University (Nanjing, China). *Trib3*-knockout (*Trib3*$^{loxP/loxP}$, *Trib3*$^{F/F}$) mice (5–6 weeks old, male) were generated as described in our previous study (Li et al.[18]). *Cre*$^{ERT2}$ (B6. Cg-Tg(CAG-cre/Esr1*)5Amc/J) mice (5–6 weeks old, 1 male and 2 females) (The Jackson Laboratory, 004682) were obtained from The Jackson Laboratory (CA, USA). When *Cre*$^{ERT2}$ transgenic mice bred with mice containing loxP-flanked sequences, tamoxifen-inducible Cre-mediated recombination results in deletion of the floxed sequences in widespread cells/tissues of the offspring. *Cre*$^{Lck}$ (B6. Cg-Tg (Lck-cre)548Jxm/J) mice (5–6 weeks old, 1 male and 2 females; The Jackson Laboratory, 003802), and *Cre*$^{CD19}$ (B6.129P2(C)-Cd19tm1(cre)Cgn/J) mice (5–6 weeks old, 1 male and 2 females; The Jackson Laboratory, 006785) were obtained from the Shanghai Research Center for Model Organisms (Shanghai, China). B6.129P2-Lyz2tm1(cre)/Nju (LysM-Cre) mice(5–6 weeks old, 1 male and 2 females; N000056) were obtained from Model Animal Resource Information Platform. Myeloid cell-specific, thymocyte-specific, and B lymphocyte-specific *Trib3*-knockout mice were generated by crossing *Trib3*$^{F/F}$ mice with *Cre*$^{Lysm}$ mice, *Cre*$^{Lck}$ mice and *Cre*$^{CD19}$ mice, respectively. The generation of inducible *Trib3*-knockout mice was performed by breeding *Trib3*$^{F/F}$ mice with *Cre*$^{ERT2}$ mice and treating them with tamoxifen to induce *Trib3* gene deletion. Lymphomagenesis was induced by transgenic Eμ-driven Myc (*Myc*$^{Eμ}$). *Myc*$^{Eμ}$ mice were obtained from the Nanjing Biomedical Research Institute of Nanjing University (Nanjing, China, N000116). To determine the role of Trib3 in lymphomagenesis, *Myc*$^{Eμ}$ mice were crossed with *Cre*$^{Lysm}$*Trib3*$^{F/F}$, *Cre*$^{Lck}$*Trib3*$^{F/F}$, *Cre*$^{CD19}$*Trib3*$^{F/F}$, and *Cre*$^{ERT2}$ *Trib3*$^{F/F}$ mice. For the induced deletion of *Trib3* in *Myc*$^{Eμ}$*Cre*$^{ERT2}$*Trib3*$^{F/+}$ mice, 3-month-old *Myc*$^{Eμ}$*Cre*$^{ERT2}$*Trib3*$^{F/+}$ mice were treated with tamoxifen (Sigma Aldrich, 06734) at a dose of 250 mg/kg i.p. for 6 days, followed by tamoxifen injection once per week until killing at 6 months. Deletion of *Trib3* was corroborated by quantitative PCR and immunoblot analysis. Animals were housed in groups of 4–6 mice per individually ventilated cage in a 12 h light/dark cycle (07:30–19:30 light. 19:30–7:30 dark), with controlled room temperature (23 ± 2 °C) and relative humidity (40–50%).

All mice were maintained in the animal facility at the Institute of Materia Medica under specific-pathogen free (SPF) conditions. For animal studies, the mice were earmarked before grouping and then randomly separated into groups by an independent person; however, no particular method of randomization was used. Sample size was predetermined empirically according to previous experience using the same strains and treatments. Generally, we used $n ≥ 6$ mice per genotype and condition. We ensured that the experimental groups were balanced in terms of animal age and weight. All animal procedures were conducted in accordance with the guidelines of the Institutional Committee for the Ethics of Animal Care and Treatment in Biomedical Research of Chinese Academy of Medical Sciences and Peking Union Medical College (PUMC). The animal study was also performed according to the ARRIVE guidelines[55].

**Human lymphoma tissues/samples and cell culture.** Human lymphoma patient specimens were obtained from Anyang Tumor Hospital, Henan University of Science and Technology and the Institute of Hematology and Blood Diseases Hospital of PUMC. Informed consent was obtained from all participants in accordance with the Declaration of Helsinki. The procedure was approved by the institutional review board at Anyang Tumor Hospital and the Ethics Committee of the Institute of Hematology and Blood Diseases Hospital of PUMC (KT2019055-EC-1). All participants provided written informed consent to publish information that identifies individuals. Our study is compliant with the 'Guidance of the Ministry of Science and Technology (MOST) for the Review and Approval of Human Genetic Resources', which requires formal approval for the export of human genetic material or data from China. Patient-related information is provided in Supplementary Table 1.

H9 (T cell lymphoma) cells, NAMALWA (BL) cells, K562 (chronic myelogenous leukemia) cells, MOLT-4 (acute lymphoblastic leukemia) cells, bjab (BL) cells, Jurkat (acute T cell leukemia) cells, Sup-B15 (acute B cell leukemia) cells, U937 (myeloid leukemia) cells, NALM6 (acute B cell leukemia) cells, BV173 (chronic myeloid leukemia) cells, Raji (BL), and HEK 293T cells were purchased from Shanghai Bioleaf Biotech Co., Ltd., where they were recently authenticated by short tandem repeat (STR) profiling and characterized by mycoplasma and cell vitality detection. These cells were cultured and maintained in RPMI 1640 medium supplemented with 10% fetal bovine serum (FBS, Invitrogen, CA, USA) under 5% carbon dioxide. All cell lines were verified negative for mycoplasma contamination by MycoAlert™ Mycoplasma Detection Kit (Lonza, LT07-418). No cells lines used here appear in the database of commonly misidentified cell lines (the International Cell Line Authentication Committee). For generation of cells stably expressing *TRIB3*$^{Cas9}$ or sh*UBE3B*, *TRIB3*$^{Cas9}$ or sh*UBE3B* lentiviral particles were purchased from TransOMIC Technologies Inc. The gRNA sequence targeting *TRIB3* was 5′-CATGGTTCATCGACAGAGCC-3′, and the shRNA sequence targeting *UBE3B* was 5′-GAGAGAGATTGATGACTTT-3′. After 24 h of infection, stable cells were selected in medium containing 1 μg/mL puromycin (Gibco) for 14 days. After 2–3 passages in the presence of puromycin, the cultured cells were used for experiments without cloning. The Amaxa™ 4D-Nucleofector system was used to transiently transfect plasmids into primary lymphoma cells or suspension cells following the standard instructions (Lonza).

**Plasmid construction.** The Human *MYC*-Flag-tagged (HG11346-CF), *SKP2*-Flag-tagged(HG15079-CF), *TRIM6*-untagged (HG18626-UT), *TRIM32*-untagged (HG18626-UT), *TRIM21*-Flag-tagged (HG18010-CF), *FBXW7*-His-tagged (HG29625-CH), *AURORA*-Myc-tagged (HG10669-CM), *MAX*-His-tagged (HG12885-CH), *COP1*-FLAG-tagged (HG19467-CF), *TRIB2*-FLAG-tagged (HG10725-CF), *UBCH1 (UBE2K)*-Myc-tagged (HG16296-CM), *UBCH3 (CDC34)*-Myc-tagged (HG11443-CM), *UBCH5A (UBE2D1)*-Myc-tagged (HG11432-CM), *UBCH5B (UBE2D2)*-Flag-tagged(HG17832-CF), *UBCH5C (UBE2D3)*-Myc-tagged (HG16261-CM), *UBCH6 (UBE2L3)*-Myc-tagged (HG12003-CM), *UBCH7 (UBE2L3)*-Myc-tagged(HG12005-CM), *UBCH8 (Ube2L6)*-Myc-tagged (HG16791-CM), *UBCH10 (UBE2C)*-Myc-tagged(HG16911-CM), *UBCH13*-Myc-tagged (HG16310-CM), and *UB*-HA-tagged (HG16831-NY) plasmids were purchased from Sino Biological Inc. (Beijing, China). The human *UBE3B*-Flag-tagged (OHu09547) and *HACE1*-FLAG-tagged (OHu17253) plasmid was purchased from GenScript Inc. The human *UBCH2(UBE2H)*-Myc-DDK-tagged (RC202516) was purchased from Origene. The truncations of *MYC*, *MYC*-N1 (amino acids 1-360), Δ*LZ* (amino acids 1-426), Δ*TAD* (amino acids 144-454), Δ*TAD*-Δ*LZ* (amino acids 144–426), and central region (amino acids 144–360) were inserted into the pcDNA3.1-myc-his vector (Invitrogen,V85520) by standard subcloning. The truncations of *MYC*, MYC-N (amino acids 1–144), *MYC*-C (amino acids 360–454), and central region (amino acids 144–360) were inserted into the pEGFP-C1 vector by standard subcloning. *TRIB3*-FL (amino acids 1–359) and the truncations of TRIB3, *T3-N* (amino acids 1–179), *T3-C* (amino acids 180–359), and *KD* (amino acids 72–315) plasmids were inserted into the pcDNA3.1-N-HA vector (GenScript, N/A)by standard subcloning. *TRIB3*-HA was cloned into the pcDNA3.1-N-HA vector by standard subcloning. *TRIB3*-Flag-tagged (HG10731-CF) was purchased from Sino Biological Inc. (Beijing, China). MYC T58A based on the isoform of MYC with 439 aa was inserted into the pcDNA3.1-Myc vector by subcloning. The truncations of *UBE3B*, *UBE3B*-FL (amino acids 1–1068), *UBE3B*-N (amino acids 1–244), *UBE3B*-C (amino acids 245–1068), *UBE3B*-L (amino acids 1–682), and *HECT* (amino acids 682–1068) were inserted into the pcDNA3.1-Myc-his vector by standard

subcloning. The control-3XFlag (Empty-Flag) vector (E4151) was purchased from Sigma Aldrich. p-TRIB3-luc was generated as described in the previous study[17]. His-tagged MYC and UBE3B were inserted into the pcDNA3.1-Myc-His vector by standard subcloning. HA- and Flag-tagged ubiquitin were purchased from Sino Biological Inc. (Beijing, China).

**E3 ligase library screening**. The E3 ligase library including 71 E3 ligase expression plasmids (Flag-tagged) was constructed by standard subcloning into the control-3XFlag vector. Most clone plasmids of these E3 ligases were purchased from Sino Biological Inc. and Origene Inc. Detailed information on the E3 ligases is provided in Supplementary Table 3. For E3 ligase screening, HEK 293T cells were seeded into six-well plates. After 12 h, HEK 293T cells were cotransfected with MYC (Myc-tagged) and different E3 ligases separately. After 24 h of transfection, Co-IP experiments were performed to detect the interaction of MYC and different E3 ligases.

**Luciferase reporter assays of MYC transcriptional activity**. Cells were seeded in 12-well plates and transfected with E-box-dependent luciferase reporters (Yeasen Inc.) using VigoFect (Vigorous Biotechnology Beijing, China). pTK-Renilla was used as an internal control. Luciferase activity was measured 20–24 h after transfection using the Dual-luciferase Reporter Assay System (Promega, USA).

**Proximity ligation assay**. A total of $1 \times 10^6$ cells were seeded overnight in a six-well plate. The next day, cells were collected, fixed in 4% paraformaldehyde solution, permeabilized with 0.5% Triton-X100, blocked with Duolink® Blocking Solution, and probed with antibodies directed against MYC (R&D, NB600-302, 1:100), TRIB3 (Novus, H00057761-M03, 1:100), UBE3B (Invitrogen, PA5-59390, 1:100), and MAX (Cell Signaling Technology, #4739, 1:100). The cells were then treated with the Duolink In Situ Red Starter Mouse/Rabbit kit (Sigma Aldrich, DUO92101-1KT) according to the manufacturer's instructions. Images were captured with an OLYMPUS confocal microscope.

**Protein purification**. Purified His-tagged MYC (TP760019) was purchased from Origene Inc. GST-tagged TRIB3 (10731-H09B) and KDC domains, GST-tagged MAX (12885-H20B) and His-tagged MAX (12269-H08H) were purchased from Sinobiological Inc. The recombinant proteins UBE3B (His-tagged), UBE3B-C (His-tagged), UBE3B R346Q, and UBE3B C1036A were produced in Freestyle™ 293-F cells (Life Technology, Carlsbad, CA, USA) following transient transfection with 1 mg/mL DNA at a DNA/PEI ratio of 1:2.5 (PEI, Polyscience). Proteins were purified from culture supernatants using Ni-NTA-Sepharose (GE Healthcare) and dialyzed against PBS. Purified proteins were loaded onto a Hiload 16/600 Superdex 200 gel filtration column (GE Healthcare) to remove aggregates. Purified proteins were concentrated to 4 mg/mL by a 10-kDa Centrifugal Filter Unit (Millipore, Burlington, MA, USA) and filtered by a 0.22-μm Millipore filter.

**Peptide synthesis**. All peptides were manufactured by Chinese Peptide Inc. (Hangzhou, China). Synthetic peptides were purified to >98% purity by high-pressure liquid chromatography for both in vitro and in vivo applications. The CM1, CM2, CM3, and CM4 peptides have the following amino acid sequences in the D-isoform: ERQRRNELKRSFFALR, KVVILKKATAYILSV, QAEEQKLI-SEEDLLR, and KRREQLKHKLEQLR, respectively. The scramble CM4 peptide has the following amino acid sequence in the D-isoform: LRKERQRLEKQLHK. The PCM4 and PCON peptides have the following amino acid sequences in the D-isoform: HLYVSPWGGKRREQLKHKLEQLR and HLYVSPWGGLRKERQR-LEKQLHK, respectively. For in vitro experiments, PCON and PCM4 were dissolved in PBS to generate a 4-mM stock. For in vivo use, PCON and PCM4 were dissolved in PBS and kept on ice until injection. Before injection, the solution was brought to room temperature.

**Virus infection in vivo**. To evaluate the effect of Ube3b overexpression on the lymphomagenesis of $Myc^{E\mu}$ mice, 2-month-old $Myc^{E\mu}$ mice were each infected with $1 \times 10^8$ PFU viral particles of GFP, Ube3b, or Trib3 plus Ube3b adenovirus (i.p., twice a week) for 4 consecutive weeks. These mice were then evaluated for the development of lymphoma and survival. To evaluate the effect of Ube3b deletion on lymphoma inhibition induced by Trib3 knockout, 2-month-old $Myc^{E\mu}$-$Cre^{CD19}Trib3^{F/+}$ mice were each infected with $1 \times 10^7$ PFU viral particles of $Ctrl^{Cas9}$ or $Ube3b^{Cas9}$ lentivirus (the gRNA sequence targeting Ube3b was 5′-CATGGTTCATCGACAGAGCC-3′; i.p., twice a week) for 4 consecutive weeks. These mice were then evaluated to determine the survival rate.

**Lymphoma-bearing PDX mouse models**. Four 5-week-old male NSG mice were implanted subcutaneously with 200 μl of a 1:1 mixture of Matrigel® Basement Membrane Matrix High Concentration (Corning, 354230) and minced fresh tumor fragments resected from lymphoma patients. Approximately 3–4 weeks following implantation, the first-generation mice (G1) were killed, and tumor masses were isolated. The tumor cells were tested for human CD19 (BCL) or CD3 expression (T cell lymphoma), and the remaining tumor mass was cut into 3-mm³ sections and passaged into 4–5 NSG mice as the second generation (G2). After stable passage of

the third generation (G3), the tumor mass of the fourth generation (G4) was equally cut into 3-mm³ pieces and then passaged subcutaneously into 30–50 NSG mice (100 μl of a 1:2 mixture of Matrigel and tumor tissue) for in vivo treatment (6–8 mice per group, dependent on the experimental drug treatment). During and after treatment with the vehicle control or the indicated agent, tumor burden was evaluated by measuring the tumor volume to determine the therapeutic efficacy in the PDX models. Freshly isolated cells from the tumor mass of each generation and treatment group were labeled with a FITC/PE-conjugated anti-human CD19 (Biolegend, 392508,1:100) or PE-conjugated anti-human CD3 (Biolegend, 981004, 1:100) to validate the population of human lymphoma cells by flow cytometry.

**Peptide treatment in vivo**. To evaluate the effect of PCM4 on $Myc^{E\mu}$ mice, 2-month-old $Myc^{E\mu}$ mice were treated with 2 mg/kg PCON, 1 mg/kg PCM4, 2 mg/kg PCM4, or 5 mg/kg PCM4 (i.v., three times a week) for 3 months. For the limiting dilution transplantation of lymphoma cells, $5 \times 10^5$, $5 \times 10^4$, or $5 \times 10^3$ murine $Myc^{E\mu}$ lymphoma cells with or with PCM4 treatment (2 mg/kg) were injected i.p. into immunodeficient NSG mice. To evaluate the survival rate of secondary recipients, these mice were monitored for 70 days. For the evaluation of IHC staining and protein expression, three mice of each group were killed at the end of treatment, and the thymuses, spleens and LNs were collected for pathology detection, Co-IP, and western blot assays.

**Combined treatment with peptide and doxorubicin**. To evaluate the antilymphoma effects of PCM4 and DOX (Sigma Aldrich, D1515), 2-month-old $Myc^{E\mu}$ mice were treated with vehicle, 2 mg/kg DOX (100 μl, i.p., once a week) and 5 mg/kg PCON (100 μl, i.v., three times a week) or 2 mg/kg DOX (100 μl, i.p., once a week) and 5 mg/kg PCM4 (100 μl, i.v., three times a week) for 3 months. To evaluate the survival rate, these mice were monitored for 8 months ($Myc^{E\mu}$ mice). To evaluate the antilymphoma effects of PCM4 and DOX on the PDX model, tumor mass-transplanted NSG mice (0.1 cm³) were treated with vehicle, 2 mg/kg DOX (100 μl, i.p., once a week), 5 mg/kg PCM4 (100 μl, i.v., three times a week) or 2 mg/kg DOX (100 μl, i.p., once a week), and 5 mg/kg PCM4 (100 μl, i.v., three times a week) for 3 weeks. During the indicated treatments, tumor burden was evaluated by measuring the tumor volume to determine the therapeutic efficacy in the PDX models.

**Flow cytometry**. Primary cell suspensions from the thymus, spleen, or LNs or Raji cells with or without TRIB3 deletion were directly labeled. Fluorescently labeled antibodies against the following surface proteins were used for human/mouse cell staining: FITC anti-human CD19 antibody (Biolegend, 392508, 1:100), PE anti-human CD3 antibody (Biolegend, 981004, 1:100), APC anti-mouse CD3 antibody (Biolegend, 100236, 1:100), FITC anti-mouse/human CD45R/B220 antibody (Biolegend, 103205, 1:100), FITC anti-mouse Ki-67 antibody (Biolegend, 652410, 1:100), PE anti-mouse/human CD11b antibody (Biolegend, 101207, 1:100), FITC anti-human CD11b antibody (Biolegend, 301330,1:100). Data were acquired using a FACSCalibur or FACSCanto II flow cytometer (BD). FCS EXPRESS software was used for data analysis.

**Primary cell purification/isolation**. For protein analysis or other assays, B cells from mouse spleens or LNs were isolated by Dynabeads™ Mouse Pan B (B220; Dynabeads; Invitrogen, 11441D). Primary human BCL cells were purified using MACS CD19 MicroBeads (Miltenyi Biotec, 130-050-301) at 4 °C, and the percentage of BCL cells (CD19+) was determined to be >90% by flow cytometry. Primary human TCL cells were isolated with the EasySep™ Human T Cell Isolation Kit with immunomagnetic negative selection according to standard instructions (StemCell Inc, #17951). Thymocytes were isolated from mice as described below: the thymus was removed, trimmed clean of connective tissue and brown fat (if present) and transferred into Hanks' Balanced Salt Solution (Sigma) containing 10% (w/v) FBS. A single cell suspension was prepared by passage through a 70-μm nylon sieve (Falcon).

**Immunoprecipitation, immunoblotting, immunostaining, and lymphoma tissue microarray**. Co-IP experiments were performed as described previously[18]. Briefly, cells were collected and lysed for 30 min on ice. Soluble lysates were incubated with the indicated antibodies at 4 °C overnight, followed by incubation with Protein A/G Plus-Agarose (Santa Cruz Biotechnology, TX, USA) at 4 °C for 2 h. Soluble lysates were incubated with the indicated Anti-Myc magnetic beads (Bimake.com, B26302), Anti-Flag Affinity Gel (Bimake.com, B23102) or Anti-HA Affinity Gel (Bimake.com, B23302) at 4 °C overnight. Immunocomplexes were separated from the beads and then boiled for 10 min. The precipitated proteins were subjected to SDS-PAGE and blotted with specific antibodies. For immunoblotting assays, proteins were extracted from cells, spleens, or lymphoma tissues using RIPA buffer (Cell Signaling Technology, MA, USA). A BCA Protein Assay Kit was used to determine protein concentrations. Protein extracts were separated by SDS-PAGE, transferred onto PVDF membranes, and subjected to immunoblot analysis. Western blot images were captured by a Tanon 5200 chemiluminescent imaging system (Tanon, Shanghai, Beijing). Human lymphoma and benign LN paraffin tissue microarrays were purchased from US Biomax (LM482a).

For immunofluorescence staining, cells seeded on coverslips were briefly washed with PBS and fixed with 4% buffered paraformaldehyde for 15 min, permeabilized with 0.5% Triton X-100 for 15 min, blocked with 3% BSA for 30 min at 37 °C, and stained with specific primary antibodies followed by corresponding secondary antibodies. Nuclei were counterstained with DAPI. Images were captured using a confocal fluorescent microscope (Olympus Microsystems, CA, USA). Quantitative image analysis was performed with the Imaris 9.3.1 software. Pearson's coefficient was use to analyze colocalization between two target proteins.

For immunohistochemistry analysis, the paraffin-embedded tissue sections were deparaffinized with xylene and hydrated through graded alcohols to water. Antigen retrieval was carried out with a citrate buffer (10 mM sodium citrate buffer, pH 6.0) at sub-boiling temperature for 15 min. The sections were permeabilized with 0.5% Triton-100/PBS for 20 min. Endogenous peroxidase activity was blocked with 3% $H_2O_2$ solution for 10 min, followed by washing three times with PBS. Blocking buffer (3% BSA/PBS) was added to the sections and incubated for 30 min. Slides were then incubated with indicated primary antibodies at 4 °C overnight. After washing three times, sections were incubated for 30 min with corresponding secondary antibodies at room temperature. Signals were detected with freshly made DAB substrate solution (ZSGB-BIO Company, Beijing, China). Sections were then counterstained with hematoxylin, dehydrated, and mounted with coverslips. Images were captured using Olympus DP72 microscope (Olympus Microsystems, CA, USA) and analyzed by Image-Pro Plus 5.1.

For immunoblotting, the following antibodies were used: anti-TRIB3 (Abcam, ab75846, 1:1000), anti-TRIB3 (ThermoFisher, PA5-15480, 1:1000), Anti-TRIB3 (Abcam, ab137526, 1:1000), TRB-3 Antibody (D-4) (Santa cruz Biotechnology, sc-365842, 1:500), anti-GAPDH (ZSGB-BIO TA-08, 1:2000) anti-c-Myc (D3N8F) (CST, #13987 S, 1:1 000), anti-c-Myc Antibody (CST, #9402 S, 1:1000), anti-UBE3B (Abcam, ab83834,1:1000), anti-β-Actin (D6A8) (CST,# 8457 S,1:1000), anti-Ubiquitin (CST, #3933 S, 1:1000), anti-Fbxw7 (Abcam,ab109617,1:1000), anti-Max (S20) (CST, #4739,1:1000), anti- HUWE1 (Abnova, PAB12996, 1:1000), anti-Phospho-c-Myc (Thr58) (E4Z2K) (CST, #46650,1:1000), anti-Phospho-c-Myc (Ser62) (E1J4K) (CST, #13748,1:1000), anti-UBE3A [EPR7330](Abcam, ab126765,1:1000), anti-UBE3C (Abcam, ab226173,1:1000), anti-TRIB2 (Abcam, ab117981,1:1000), anti-TRIB1 (Abcam, ab137717,1:1000), anti-TRIM32 (Abcam, ab131223, 1:1000), anti-Myc (MBL, #562, 1:1000), anti-Myc (MBL, M047-3, 1:1000), anti-GFP (MBL, #598, 1:1000), anti-GFP (MBL, M048-3, 1:1000), anti-DDDDK (MBL, PM020,1:1000), anti-DDDDK (MBL, M185-3L,1:1000),anti-HA (MBL, 561, 1:1000), anti-HA (MBL, M180-3,1:1000), anti-His (MBL, PM032,1:1000), anti-His (MBL, D291-3,1:1000). For Immunofluorescence& Immunohistochemistry: anti-TRIB3 (Abcam, ab137526, 1:100), anti-MYC (R&D, AF3696, 1:100), anti-MYC(R&D, NB600-302, 1:100), anti-Ki67 [SP6] (Abcam Ab16667,1:100), anti-UBE3B (Invitrogen, PA5-59390, 1:100), anti-TRIB3 (1H2) (Novus, H00057761-M03,1:100), anti-Max (S20) (CST, #4739,1:100), Alexa Fluor 488 (Thermo Fisher, R37114, 1:200), Alexa Fluor 488 (Thermo Fisher, R37118, 1:200), Alexa Fluor 555 (Thermo Fisher, A-31572,1:200), Alexa Fluor 555 (Thermo Fisher, A-31570,1:200), Alexa Fluor 647 (Thermo Fisher, A-31571, 1:200), Alexa Fluor 647 (Thermo Fisher, A-31573, 1:200). Chip-seq: anti-Rpb1 CTD (4H8) (CST, #2629, 10 μg/ChIP), anti-MYC (D3N8F) (CST, #13987 S, 10 μg/ChIP). For Flow cytometry: FITC anti-human CD19 antibody (Biolegend, 392508, 1:100), PE anti-human CD3 antibody (Biolegend, 981004, 1:100), APC anti-mouse CD3 antibody (Biolegend, 100236, 1:100), APC anti-mouse/human CD45R/B220 antibody (Biolegend, 103212, 1:100), FITC anti-mouse Ki-67 antibody (Biolegend, 652410, 1:100), and FITC anti-human CD11b antibody (Biolegend, 301330,1:100).

**In vivo ubiquitylation assay**. HEK 293T cells were cotransfected with Myc-tagged MYC, HA-tagged or Flag-tagged ubiquitin, and other indicated plasmids. Cells were pretreated for 6 h with MG132 (Sigma Aldrich, C2211) at a concentration of 20 μM, and cell extracts were prepared using lysis buffer containing 1% TX-100 supplemented with 1 mg/ml iodoacetamide. We used a monoclonal anti-HA Affinity Gel (Bimake.com, B23302) or a ubiquitin antibody (CST, #3933S, 5 μg/IP) to purify ubiquitinylated proteins. Precipitates were analyzed by immunoblot with an anti-MYC antibody (CST, #9402 S, 1:1000).

**In vitro ubiquitination assay**. In vitro ubiquitination was performed according to the protocol provided with the Ubiquitination Kit (UW9920, BioMol) with some modifications. Briefly, MYC-His, UBE3B-His, UBE3B-(R346Q)-His, UBE3B-(C1036A)-His, and UBE3B-C-His were expressed in free-style 293 cells and purified by a Ni2+-NTA resin (GE). The assays were carried out at 37 °C in a 50-μl reaction mixture containing 20 U/ml inorganic pyrophosphatase (Sigma Aldrich), 5 mM dithiothreitol, 5 mM Mg-ATP, 100 nM E1, 2.5 mM E2, 0.75–1 mM E3, 1 mM substrate protein, and 2.5 mM biotin-labeled ubiquitin. After incubation for 30–60 min, the reactions were quenched by the addition of 50 μl of 2X nonreducing gel-loading buffer and separated by 12% sodium dodecyl sulfate polyacrylamide gel electrophoresis (SDS-PAGE). To obtain accurate results, the PAGE gel was run for a relatively long time until protein bands smaller than 40 kDa ran off the bottom line of the gel as judged by the protein molecular weight marker. Then, the protein was transferred to a polyvinylidene difluoride membrane (PVDF). To reduce the background noise, an anti-MYC antibody (Cell Signaling Technology, 1:1000) was used to detect MYC polyubiquitination via western blot analysis instead of the

HRP-Streptavidin detection system recommended by the kit for the detection of biotinylated ubiquitin.

**In vitro coimmunoprecipitation**. Purified His-tagged MYC (5 μg) and His-tagged UBE3B (5 μg); purified His-tagged MYC (5 μg) and His-tagged UBE3B (5 μg) with GST-tagged TRIB3 or the KD domain; purified His-tagged UBCH3 (5 μg) and His-tagged UBE3B (5 μg) with GST-tagged TRIB3 or the KD domain; purified His-tagged MYC (5 μg), His-tagged UBE3B (5 μg), and GST-tagged TRIB3 (10 μg) with or without the CM4 peptide; purified His-tagged MYC (5 μg), His-tagged MAX (5 μg) and GST-tagged TRIB3; or purified His-tagged MYC (5 μg), His-tagged UBE3B (5 μg), and GST-tagged MAX were mixed in a reaction buffer (100 μl) consisting of 1% NP-40, 120 mM NaCl, 40 mM tris-HCl (pH 7.4), 1.5 mM sodium orthovanadate, 50 mM sodium fluoride, 10 mM sodium pyrophosphate, and protease inhibitor cocktail (Roche) for 6 h at 4 °C. Co-IP experiments were performed as described in the section of immunoprecipitation.

**Immunoprecipitation with high-throughput sequencing (ChIP-seq)**. ChIP assays were performed according to the manufacturer's protocol using a SimpleChIP® Plus Sonication Chromatin IP Kit (Cell Signaling Technology, Danvers, MA, USA, (#56383). Briefly, cells were fixed with 1% formaldehyde (Sigma Aldrich, F8775) for 8 min, incubated with glycine (50 mM final) for 10 min and washed three times with PBS. After cell lysis and chromatin extraction, chromatin was sonicated to 100–500 bp using a BioRuptor sonicator (Diagenode), followed by centrifugation at 16,000×g for 10 min at 4 °C. To control for experimental variation, our spike-in data were normalized to a spike-in control consisting of a small amount of chromatin from another species. Briefly, Baf3 cell lysates were sonicated exactly as described above and added at a 1:100 ratio of Raji cells to lysates. The spiked lysates were incubated overnight at 4 °C with ChIP-grade antibodies specific for MYC (CST, #13987 S, 10 μg/ChIP) or RNAPII (Cell Signaling Technology, #2629, 10 μg/ChIP), which were coupled to magnetic beads. Precipitated material was eluted (input chromatin was used as a control), the crosslink was reverted, and DNA was purified by chloroform/phenol extraction and resuspended in DNA elution buffer. ChIP-seq libraries were generated by DNA fragmentation, end repair, dA-tailing, adapter ligation and PCR amplification. The Qubit® 3.0 A fluorometer was used for quantitation of the ChIP library. The Agilent 2100 Bioanalyzer was used to test the insert sizes of the library. The StepOnePlus™ Real-Time PCR system was used to assess the molality of the library, which must exceed 10 mM for sequencing.

**ChIP-sequencing analysis**. The ChIP libraries were sequenced on Illumina HiSeq platform in 150 bp paired-end reads. After quality control, the clean reads were aligned to the human reference genome (Homo sapiens. GRCh38/v2) with Bowtie v2.1.0. Unambiguously mapped reads were retained for subsequent generation of binding profiles, heatmaps and calling of peaks. MACS2 (Model-based Analysis of ChIP-seq, version 2) was used to identify regions in ChIPed samples of the signal enriched over the background signal from the corresponding input sample, and P value of $10^{-9}$ was used as the cutoff to identify statistically significant peaks. Mapped reads were visualized using the Integrative Genomics Viewer (IGV). We next used the murine spike-in to quantitatively normalize different samples. Briefly, the total number of reads aligned to the mouse mm9 genome assembly was used as a normalization factor to scale ChIP-seq data sets produced from equal cell numbers. Bamliquidator (https://github.com/BradnerLab/pipeline/wiki/bamliquidator, version 1.0) was used to calculate the ChIP-seq read density over a given genomic coordinate. Heatmaps and genome-wide correlation analyses were generated using DeepTools2. To create density distributions around TSS or ChIP peaks and heatmaps indicating c-Myc or RNA polymerase II occupancies, plotHeatmap was used. Peak annotations were achieved using the 'closestBed' feature from the Bedtools suite v2.20.1.

**MYC K427 and T58 mutations generated by the CRISPR/Cas system**. The sgRNAs for MYC K427R and T58A knock-in mutations were annealed and ligated into the pGK1.1 vector digested with BbsI. The gRNA sequence targeting MYC K427R was 5′-TTTCTGAAGAGGACTTGTTGCGG-3′, and that targeting MYC T58A was 5′-GCAGCCCCCCGGCGCCCAGCGAGG-3′. The repair template designed with a homologous genomic flanking sequence centered around the predicted CRISPR/Cas9 cleavage site was annealed and ligated into a donor vector (pUC57) digested with EcoRI and HindIII. One microgram of each sgRNA plasmid was mixed with 1 μg of donor plasmid for nucleofection into Raji cells with the Cell Line Nucleofector Kit V (LONZA, VCA-1003) according to the manufacturer's instructions. Twelve hours after nucleofection, Raji cells were treated with 1 μg/mL puromycin for 24 h. Then, puromycin was removed from the cell culture medium, and the cultured cells were sorted into 96-well plates as single cells per well. The cells were incubated and expanded for 2–3 weeks, and all of the clones were further subjected to genomic DNA extraction (TIANamp Genomic DNA Kit, DP304), PCR amplification of the MYC sequence and Sanger sequencing. The sequences of the PCR primers were as follows: MYC-K427R forward, 5′-GGGTCAAGTTGGAC AGTGTCAGAGT-3′; MYC-K427R reverse, 5′-TTACGCACAAGAGTTCCGTAG CTGT-3′; MYC-T58A forward, 5′-CTGGATTTTTTTCGGGGTAGT-3′; and

*MYC-T58A* reverse, 5′-CCGCTCCACATACAGTCCTG-3′. The correct K427R and T58A knock-in cell clones were selected for further experiments.

**TRIB3 transcription assay**. To assess TRIB3 transcription, HEK 293T cells were transfected with the *p-TRIB3* luciferase reporter and TK plasmids. After 24 h of transfection, the cells were harvested with lysis buffer, and luciferase activity was then measured using the dual-luciferase assay (Promega, Madison, WI, USA, E1910).

**Proliferation and methylcellulose colony assay**. For analysis of the effects of PCM4 and *TRIB3* depletion on cell viability, cells were seeded in 96-well plates at a concentration of 3000 cells per well in triplicate, and CCK-8 (Dojindo Laboratories, Beijing, China, CK04) was used to assess cell proliferation according to the manufacturer's instructions. Mouse and human lymphoma cells formed colonies upon serum culture (P1). Single colonies of lymphoma cells were dispersed in methylcellulose-based medium MethoCul™ M3630 (StemCell, 03630) or MethoCult™SF H4536 (StemCell, 04536). A representative plate was then washed and cells were resuspended and replated. After an additional 14 days, colonies were counted (P2).

**Real-time PCR and RNA interference**. Total RNA was extracted using TRIzol (Invitrogen, CA, USA) according to the manufacturer's instructions. Reverse transcription of total cellular RNA was carried out using oligo (dT) primers and M-MLV reverse transcriptase (Transgen Biotech, Beijing, China). PCR was performed using a Mycycler thermal cycler and analyzed using LineGene9600. The PCR primer sequences were as follows: *MYC* forward, 5′-TTCTGTGGAAAAGAG GCAGG-3′; *MYC* reverse, 5′-TGCGTAGTTGTGCTGATGTG-3′; *GAPDH* forward, 5′-GTGGACATCCGCAAAGACC-3′; *GAPDH* reverse, 5′-CCTAGAAGCA TTTGCGGTG-3′; *TFAP4* forward, 5′-GCTGAGTCTCGGGGGTTAGT-3′, *TFAP4* reverse, 5′-GTGCCCTCTTTGCAACATTT-3′; *PLD6* forward, 5′-GCTTGGAGCT CTGCCTCTTCGC-3′, *PLD6* reverse, 5′-TACCTGTATCCCTGCCTTGCGC-3′; *CCNA1* forward, 5′-GGTGTTGACTGAAAATGA GCAG-3′, *CCNA1* reverse, 5′-G AAACCTGTCCAGGAAGTTGAC-3′; *CCND1* forward, 5′-CCGTCC ATGGGAG AGATC-3′, *CCND1* reverse, 5′-ATGGCCAGCGGGAAGAC-3′; *CCNE2* forward, 5′-TTC TTGAGCAACACCCTCTTCTGCAGCC-3′; and *CCNE2* reverse, 5′-TCG CCATATACCGGTCAAAGAAATCTTGTGC-3′. The human *FBXW7* siRNA sequence was 5′-GGGACATACAGGTGGAGTA-3′, the *human MYC* siRNA sequence was 5′-GGAAGGTCTAAGAG GCGAA-3′, and the *human HUWE1* siRNA sequence was 5′-GGAGCAGATTATCGTGACA-3′. *TRIB3* siRNAs were produced by RiboBio (Guangzhou, China) and transfected using Lipofectamine RNA interference MAX Transfection Reagent (Life Technologies, CA, USA) according to the manufacturer's instructions.

**Structured illumination microscopy**. Cells (~20,000) were grown on coverslips and fixed with formalin after the indicated treatment. Subsequently, cells were washed in Tris-buffered saline (TBS) and permeabilized for 5 min in 2% Triton X-100 in TBS. The cells were quenched for 10 min with 50 nM glycine in TBS and blocked for 30 min with 5% normal mouse serum or normal goat serum in a 0.2% gelatin-TBS solution depending on the isotype of the secondary antibody. Subsequently, 30-μl droplets containing MYC antibody (R&D, AF3696, 1:100) and TRIB3 antibody (Abcam, ab75846, 1:100) dilutions were placed on Parafilm in a dark humidified chamber. The coverslips were placed facing the droplets and incubated overnight at 4 °C. The next day, the coverslips were lifted by adding a small volume (200 μl) of TBS under the coverslip. After washing for 3 × 20 min with 1 ml of 0.2% gelatin-TBS, Alexa Fluor 488 (Thermo Fisher, R37114, 1:200) and Alexa Fluor 555 (Thermo Fisher, A-31572, 1:200) secondary antibody incubation was performed as described for the primary antibody, and the coverslips were incubated for 1 h at room temperature. Following three 10-min washes with 1 ml of 0.2% TBS-gelatin and one wash with regular TBS, the slides were mounted using soft set mounting medium with DAPI (Vectashield) and sealed with nail polish. Images were acquired using a confocal microscope (Olympus Microsystems, CA, USA) or a GE Healthcare 3D structured illumination microscope. Intensity plots of individual pixels taken from a straight line in the indicated immunofluorescence images were generated by twin slicer analysis using Imaris 3D & 4D image software (Bitplane AG, Switzerland). Images were cropped and processed in Adobe Photoshop. When comparisons were made between images of the same experiment, all levels were adjusted equally, and the ratio between the levels was not altered.

**Analysis of surface plasmon resonance**. Binding kinetics between MYC and UBE3B, TRIB3 and the indicated α helical peptides, MAX and the indicated α helical peptides, or MYC and MAX were analyzed using a BIAcore T200 instrument (GE Healthcare, CA, USA) as described previously[18]. Briefly, binding reactions were performed in HBS-EP buffer (GE Healthcare) at pH 7.4. TRIB3/MYC/MAX protein was coated on the flow cell of a CM5 sensor chip by direct immobilization. Another flow cell, not coated with an immobilized protein, was used to evaluate nonspecific binding. The binding analyses were performed at a flow rate of 25 μl/min at 25 °C. A total of 0–200 nM peptide or 0–500 nM TRIB3 protein was

passed over the surface of the sensor chip. The association rate constant (Ka) and the dissociation rate constant (Kd) were calculated according to the BIA evaluation software. The dissociation constant (KD) was determined by Kd/Ka.

**RNA-Seq library preparation and sequencing**. RNA purity was evaluated using the Kaiao K5500® spectrophotometer (Kaiao, Beijing, China), and the RNA integrity and concentration were assessed using the RNA Nano 6000 Assay Kit and the Bioanalyzer 2100 system (Agilent Technologies, CA, USA). An RNA integrity number [RIN] > 7.5 was required for all samples. Sequencing libraries were generated using the NEBNext® UltraTM RNA Library Prep Kit for Illumina® (#E7530L, NEB, USA) according to the manufacturer's recommendations, and index codes were added to attribute sequences to each sample. In brief, mRNA was purified from 2 μg of each total RNA sample using poly-T oligo-attached magnetic beads. Fragmentation was performed using divalent cations at an elevated temperature in a NEB Next First Strand Synthesis Reaction Buffer (5X). First-strand cDNA was synthesized using random hexamer primers and RNase H. Second-strand cDNA synthesis was subsequently performed using buffer, dNTPs, DNA polymerase I, and RNase H. The library fragments were purified with QiaQuick PCR kits and eluted with EB buffer; then, terminal repair, A-tailing and adapter addition were implemented. The intended products were retrieved, and PCR was performed, at which point the library construction was complete.

**Gene set enrichment analysis**. We ranked the genes by their association with the Si-*Ctrl* (*n* = 3), Si-*TRIB3* (*n* = 3), and Si-MYC groups; BCLs from *Myc*<sup>Eμ</sup> mice (*n* = 3) compared with *Myc*<sup>Eμ</sup>*Cre*<sup>D19</sup>*Trib3*<sup>F/F</sup> mice (*n* = 3); the Ad-*Ctrl* (*n* = 3) and Ad-*UBE3B* (*n* = 3) groups; the PCON (*n* = 3) and PCM4 (*n* = 3) groups; the *MYC*<sup>WT</sup> (*n* = 3) and *MYC*<sup>T58A</sup> (*n* = 3) groups; and the *MYC*<sup>WT</sup> + *shTRIB3* (*n* = 3), *MYC*<sup>T58A</sup> + *shTRIB3* (*n* = 3), the *MYC*<sup>WT</sup> + *shCTRL* (*n* = 3), and *MYC*<sup>T58A</sup> + *shCTRL* (*n* = 3) groups, using the signal-to-noise measurement determined by GSEA according to log2-fold changes. MYC target gene sets were collected from the database (http://software.broadinstitute.org/gsea/msigdb/index.jsp).

**NanoBRET assay**. HEK 293T cells ($8 \times 10^5$ per well) were seeded into six-well culture plate and allowed to attach for 6 h. These cells were then cotransfected with 2 μg MAX-HaloTag Fusion Vector (Promega, N1870) and 0.2 μg MYC-NanoLuc Fusion Vector (Promega, N1870). Six hours after transfection, cells were then harvested and resupended in OptiMEM (Life Technologies) supplemented with 4% FCS at a density of $2 \times 10^5$ cells/ml in the absence or presence of 100 nM HaloTag NanoBRET 618 fluorescent ligand (Promega, G9801). Cells were then seeded into white, 96-well plates, with 100 μl of cell suspension per well. 10058-F4 or PCM4) was added to these cells at various concentrations. Plates were incubated for 3 h at 37 °C, 5% CO2. NanoBRET Nano-Glo Substrate (Promega, N1572) was added to both control and experimental samples at a final concentration of 10 μM. Plates were read within 10 min using a Synergy H1 plate reader (BioTek) equipped with a 450/80-nm bandpass and 610-nm longpass filter module. A corrected BRET ratio was calculated, defined as the ratio of the emission at 610 nm/450 nm for experimental samples (i.e., those treated with HaloTag NanoBRET 618 fluorescent ligand) minus the emission at 610 nm/450 nm for control samples (i.e., those not treated with HaloTag NanoBRET 618 fluorescent ligand). BRET ratios were expressed as milliBRET units (mBU), where 1 mBU corresponds to the corrected BRET ratio multiplied by 1000.

**Apoptosis analysis**. Apoptotic cells were detected by Apoptosis Detection kit (Biogems,62700-80). Cells were collected and washed twice stain buffer, resuspend cells in the Annexin V Binding Buffer and stained with 5 μL Annexin V and 5 μl 7-AAD (BD,559925) for 15 min at RT (25°C) in the dark, followed by washing with 500 μl Annexin V Binding buffer. Data were acquired using a FACSCanto II flow cytometer (BD).

**Cell cycle analysis**. Cells were collected and washed twice with PBS, pelleted, and fixed with cold 70% ethyl alcohol for at least 30 min. After being washed twice with cold PBS, cells were incubated with 200 μg/mL RNase A for 30 min at 37 °C. The cells were stained with 100 μg/mL propidium iodide (PI) for 30 min at room temperature. The samples were immediately analyzed by flow cytometry. Cell cycle phase distribution was determined using FCS EXPRESS software.

**Statistical analysis**. Data are expressed as the mean ± standard error of the mean (SEM). Comparisons between two groups were performed by unpaired Student's *t* test or one-way ANOVA. Correlations between groups were determined by Pearson's correlation test. Survival rates were analyzed by the Kaplan–Meier method. The sample number (*n*) indicates the number of independent biological samples in each experiment. Sample numbers and experimental repeats are indicated in the figures and figure legends. Generally, all experiments were carried out with *n* ≥ 3 biological replicates. *p* < 0.05 was considered statistically significant. Analyses were performed using GraphPad Prism 6.0 software.

**Reporting summary**. Further information on research design is available in the Nature Research Reporting Summary linked to this article.

## Data availability

The RNA-seq and ChIP-seq data have been uploaded to the NCBI Gene Expression Omnibus (GEO) database under accession numbers GSE117128, GSE126258, GSE143862, GSE143863, GSE143864, GSE143865 and GSE143866, respectively. The *TRIB3* mRNA expression was analyzed on the following accession code GSE6338, GSE24881 and the web site http://llmpp.nih.gov/DLBCL/. All other data supporting the findings of this study are available from the corresponding author upon reasonable request. A Reporting Summary for this study is available as a Supplementary Information file. Source data are provided with this paper.

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

## Acknowledgements

This work was supported by grants from the National Key R&D Program of China (2017YFA0205400), the "Ten thousand plan" - National high level talents special support plan, the National Natural Science Foundation of China (81530093, 81773781 to Z.W.H., 8187131307, 82073887 to K.L., 82003798 to F.W.), the CAMS Innovation Found for Medical Sciences (2016-I2M-1-007 to Z.H.W., H.F. and C.X.Z.; 2016-I2M-1-011 to K.L.; 2016-I2M-3-008 to B.C. and F.W.), the CAMS Central Public-interest Scientific Institution Basal Research Fund (2017PT31046), and the Beijing Outstanding Young Scientist Program (BJJWZYJH01201910023028).

## Author contributions

Z.W.H. conceptualized the study and participated in the overall design, supervision and coordination of the study. K.L. and F.W. designed and performed most of the experiments. X.X.L., F.H., B.C., J.J. Y, S.S.L., T.T.Z., Z.Y., Z.N.Y., and C.X.Z., participated in the molecular and cellular biological experiments. S.S., J.M.Y., Y.X., and X.W.Z. performed the animal studies. Y.F.Y. followed up the patients and collected clinical information and materials. Z.W.H. and K.L. wrote the manuscript. All authors read and approved the manuscript.

## Competing interests

The authors declare no competing interests.
