## [Peer Review File · Nature Communications]

Reviewers' comments:

Reviewer #1 (Remarks to the Author):

The authors describe a novel inhibitor/ubiquitin ligase pair and describe significant and interesting biological effects in a MYC-driven lymphoma model. They go on to ascribe these effects to a direct effect of both proteins on MYC ubiquitination and turnover. Finally, they transfect a MYC peptide and claim that the effects of this peptide validate the interaction *in vivo*,

The major and critical weakness of the paper is that the detailed mapping experiment in Extended Data Figure 6 shows that the interaction of MYC broadly maps to the amino terminus and that deletion of the leucine zipper has - relative to input - no effect on the interaction of MYC with either TRIB3 or UBE3B. However, the critical peptide used to validate the relevance of the interaction, CM4, is derived from the leucine zipper of MYC. Hence the experiments with CM4 are completely irrelevant for validating the interaction.

The discussion of this central problem in the text is highly misleading. The authors state: "Given that the TAD, HLH and LZ domains of the c-Myc mediated the c-Myc/TRIB3 interaction-- (line 267) but this is simply not true based on the Extended Data Figure 6.

In the most positive interpretation of the interaction data, the LZ makes a very small contribution to binding to TRIB3 when the TAD is deleted (Extended Data Figure 6g, compare Δ TAD to Δ TAD- Δ LZ). But if anything, this difference is larger for UBE3B, so a leucine zipper peptide, which competes with endogenous MYC, if it has any effect at all, would preferentially displace the UBE3B ligase and hence stabilize MYC

The leucine zipper does, however, interact with MAX, which is critical for MYC function. Therefore, the CM4 experiments in the critical Figures 5 and 6 simply show effects on the MAX/MYC interaction, which is well validated. Extended Figure 7d clearly shows that the CM4 peptide binds MAX, this has to be the case. This data is also mis-portrayed in the text. Disrupting the MYC/MAX interaction is known to affect all aspects of MYC biology and - importantly- to destabilize the MYC protein, so all reported effects of CM4 are expected based on what is known about the MYC/MAX interaction and the effects of 10058-F4, a soluble inhibitor that disrupts the interaction.

This problem and the manner in which the authors deal with it in the text precludes publication of the manuscript.

There are additional issues, several of them major.

The choice of gene sets to show engagement of MYC varies among figures. The authors should show the Hallmark genes sets V1 and V2 in all panels to make things comparable.

The analysis of T58 phosphorylation (lines 202ff) makes little sense. T58 regulates degradation of MYC since it is recognized by Fbxw7, so if T58 phosphorylation is important, as the authors state, the Fbxw7 is almost certainly involved. The authors however claim that it is not. Whether T58A is less ubiquitylated by UBE3B is not shown. Importantly this finding would allow the authors to test whether expression of T58A alleviates the dependence of MYC function and gene expression on TRIB3.

The description of many experiments is cryptic and often panels lack critical controls. Just to give one example: Figure 4B analysis MYC binding in a cell line that expresses high levels of endogenous MYC. So how was endogenous MYC removed? What are the expression levels of MYC and MYC K427A? These are absolutely essential data to interpret Figure 4b and this is simply impossible.

A second example is Figure 4a, which claims that MYC binds to all human genes. This would imply that MYC binds also to inactive genes, which does not agree with the available literature. Again, critical controls are missing to underline this surprising finding.

A large set of CHIP-sequencing data are based on quantitative comparisons between two sample sets, e.g. wtMYC and a mutant (Extended Data Figure 5a, 5e). In some cases, they show different individual genes and this is not acceptable: within 20,000 genes you will see any change possible. Where the authors report global changes, they are in line with the expected effects. However, such data need to be spike in-normalized to control for experimental variations: a small percentage of chromatin from another species needs to be added and used to standardize. The experimental methods show that this has not been done, therefore much of this work needs to be repeated.

Reviewer #3 (Remarks to the Author):

This study is very well executed showing a novel mechanism into Myc driven lymphomas, linked to a TRIB3 interaction, and the possible means of targeting this interaction to abrogate myc driven lymphomas. The experiments are well designed, well executed, and the data fully supports the finding. The paper is fully complete with little room for improvement. I only have minor comments:

1. In the extended data 1 the authors show that in the TRIB3 deletion models there is also deletion of TRIB2. This was not addressed at all in the experimental set up and may constitute an entirely new study. Can the authors comment how they think TRIB2 deletion is occurring in these models?
2. Given the deletion of both TRIB3 and TRIB2, can the authors show the total cell numbers for also the myeloid and B populations as shown for total thymocytes in the extended data 1.
3. It would be better to group the graphs in figure 2 d together and provide a statistical test to show the differences between the groups.
4. Details of the reporter genes in the luciferase assay in figure 2j should be named in the legend and R.L.A abbreviation explained in legend for how the data is graphed.
5. In extended data 2, it appears that while TRIB3 loss affects myc protein in most lymphoma lines, many of the leukaemia lines (eg. U937, malm6, k562, molt4 and jurkat) do not exhibit such effects on myc. The westerns, by eye, do not support the statements on line 158 pg8 and an adjustment on this interpretation may be necessary. The data supporting this mechanism in leukaemia is not as strong as the data and models the authors have for lymphoma.
6. What cells are used in extended data figure 2g and H – it is important to specify given point 5 above.
7. The data is not convincing that K332R ubiquitination is enhanced when UBE3B is overexpressed – a point that does not affect the stated mechanism but is an overinterpretation of the data.
8. What antibody is used in the IP for the tri-molecular complex shown in extended data 6 M. This needs to be in the legend/figure to clarify.
9. The authors discuss treatment with DOX. Page 15 line 297. Please explain and clarify what this is/why it is being used to treat the cells.

Reviewer #4 (Remarks to the Author):

In this paper, Li et al describe a molecular mechanism by which TRIB3 promotes lymphoma genesis via preventing proteasome mediated degradation of c-MYC. First, I'd like to commend the authors for their highly detailed mechanistic study, employing a range of mouse models and

human/murine in vitro systems to demonstrate mechanistic details for the observed oncogenic role for TRIB3. I have no fundamental concerns with the data presented, but have a number of relatively minor comments that would hopefully improve the presentation of the data and make this work more accessible to readers not directly working in this field.

- The work is very elaborate and a hough amount of data is presented in each figure. As a consequence, it is almost impossible for the outside reader to have a high-level view of the message at present. I would highly recommend the development of a schematic capturing the main points of the molecular model the authors propose and incorporate this in the discussion/final figure.

- There are inconsistencies in abbreviating human/mouse genes and proteins throughout the manuscript. for instance, by convention, protein symbols are not italicized, and all letters are in upper-case, both for human and murine proteins. (see <https://www.biosciencewriters.com/Guidelines-for-Formatting-Gene-and-Protein-Names.aspx>). This needs to be addressed both in the text and figures.

- Were MycEμCreCD19Trib3F/+ mice homo or heterozygous for the MycEμ allele? Please clarify this in the manuscript/figures

- Page 11: what is MAX?? Please ensure that you introduce the various genes you study and describe in this work.

- Ext. Fig. 1 G: Please at least comment on reduced TRIB2 levels you see in the Trib3 KO animals. TRIB2 has been shown to be oncogenic and some speculation at least whether there is a genetic/functional hierarchy between TRIB2 and TRIB3 in this context would be interesting. Better still, it would be a further confirmation of TRIB3-specificity of this mechanism if you had evidence to show that restoration of TRIB2 levels (for instance in Raji cells) has no impact on the system you studying.

- please include an alignment to show how specific CM4 is for TRIB3, compared to the other TRIB proteins in the supplementary material.

- Please comment in the manuscript what is the significance of the statement you make in page 5 "Targeted sequencing of TRIB3 showed that the allele frequency of nonsynonymous (Q84R) and synonymous (Y111Y and A323A) SNPs was separately 38.2% and 47.2% in lymphoma patients..."

- There are a number of places where the manuscript will benefit from a careful proof-reading and correction of English. a few examples are copied below:

P4: "none of the inhibitors binding directly to c-Myc has been successfully..."
"We recently report that TRIB3..."

P10" We thus constructed UBE3B R346Q mutant. This mutant showed a reduced abilities of c-Myc ubiquitination..."

Point-by-point response

Reviewer #1 (Remarks to the Author):

The authors describe a novel inhibitor/ubiquitin ligase pair and describe significant and interesting biological effects in a MYC-driven lymphoma model. They go on to ascribe these effects to a direct effect of both proteins on MYC ubiquitination and turnover. Finally, they transfect a MYC peptide and claim that the effects of this peptide validate the interaction in vivo.

1. The major and critical weakness of the paper is that the detailed mapping experiment in Extended Data Figure 6 shows that the interaction of MYC broadly maps to the amino terminus and that deletion of the leucine zipper has - relative to input - no effect on the interaction of MYC with either TRIB3 or UBE3B. However, the critical peptide used to validate the relevance of the interaction, CM4, is derived from the leucine zipper of MYC. Hence the experiments with CM4 are completely irrelevant for validating the interaction.

Re: Thank you for your professional criticism. In the revised MS, we thus redesigned and constructed three main truncations of MYC (N-terminal domain, central region and C-terminal domain) and identified that the C- and N- terminal domains, but not the central region (CT) of MYC mediated the interactions of MYC with either TRIB3 or UBE3B. Among them, the C-terminal domain of MYC showed a major but the N-terminal domain showed a minor contribution to its binding with TRIB3 or UBE3B (Revised Fig. 5g left and 5h left). In addition, we performed the reverse Co-IP assay to examine the interactions of MYC and TRIB3 or UBE3B with anti-MYC truncation antibodies (anti-GFP). Similarly, the C-terminal MYC indeed possessed a strong binding affinity to TRIB3 or UBE3B, while the N-terminal MYC weakly bind to TRIB3 or UBE3B, and the CT domain of MYC did not exhibit any detectable level of binding (Revised Fig. 5g and 5h right). These results suggest that the C-terminal domain of MYC binds to TRIB3 or UBE3B with high affinity and that the N-terminal domain of MYC has a weak binding affinity to TRIB3 or UBE3B. Based on these results, it is reasonable to use CM4 derived from the C-terminal domain of MYC to validate the relevance of the interaction of MYC and TRIB3. Following your and the editor's suggestion, we have corrected the corresponding description (Page 14) and toned down the interpretations of peptide CM4 and the interaction of MYC and TRIB3 throughout the revised MS.

In the original MS, the differential expression levels of several MYC mutants in the input samples and the overloading of IP samples caused this reviewer's criticism regarding the Extended Figure 6 of original MS. In the revised MS, we re-conducted Co-IP assays in a reverse way together with adjusted plasmid transfection process and appropriate exposure time of western blotting. The data indicate that deletion of LZ and HLH diminished the binding of MYC and TRIB3 or UBE3B more apparently than that of the N-terminus (Below panels a and b). Deletion of all three domains of MYC

almost lost its binding with TRIB3 or UBE3B (Below panels a and b). However, these data did not provide more information for the conclusion of our study, and may bring confusion to reviewers. Hence, we present them for your reviewing and will not display them in the revised MS.

(a, b) Mapping MYC regions binding to TRIB3 (a) or UBE3B (b). Top: deletion mutants of MYC. Bottom: HEK293T cells were co-transfected with the indicated constructs of MYC and TRIB3 or UBE3B. Cell extracts were IP with anti-Myc Ab.

2. The discussion of this central problem in the text is highly misleading. The authors state: "Given that the TAD, HLH and LZ domains of the c-Myc mediated the c-Myc/TRIB3 interaction-- (line 267) but this is simply not true based on the Extended Data Figure 6.

Re: Based on the reply to your question 1, we thus deleted this misleading description in the revised MS. New data presented in the revised Fig. 5g suggest that the C-terminal domain of MYC binds to TRIB3 with high affinity. Hence, we modified the description regarding the interacting domains of MYC and TRIB3 in the revised MS (Page 14). Following the editor's suggestion, we have also toned down the interpretations of the MYC-TRIB3 interaction in the revised MS.

3. In the most positive interpretation of the interaction data, the LZ makes a very small contribution to binding to TRIB3 when the TAD is deleted (Extended Data Figure 6g, compare ΔTAD to ΔTAD-ΔLZ). But if anything, this difference is larger for UBE3B, so

a leucine zipper peptide, which competes with endogenous MYC, if it has any effect at all, would preferentially displace the UBE3B ligase and hence stabilize MYC.

Re: Thank you for your professional criticism. As we replied in question 1, new truncations of MYC were used to conduct Co-IP assays, by which cell lysates were immunoprecipitated with anti-GFP (MYC truncations' tag) antibodies or anti-Flag (TRIB3 and UBE3B's tag) antibodies in the revised MS. The data indicate that the C terminus of MYC mainly mediates the interaction of MYC with UBE3B or TRIB3 (Revised Fig 5g and 5h). Hence, we modified the description of these data in the revised MS (Page 14).

Because TRIB3 and UBE3B share the same binding domain (mainly C terminus) of MYC, CM4 peptide derived from the C terminus of MYC can displace the UBE3B ligase and stabilizes MYC. However, there exists the possibility that TRIB3 and UBE3B bind to the distinct regions or residues of C terminal MYC, which deserves further exploration in future study. Moreover, distinct peptides derived from the C terminal MYC may differentially bind with TRIB3 or UBE3B. In this study, we have identified via experiments that CM4 preferentially disturbed the binding of TRIB3 and MYC (Revised Fig 6c and 6d), subsequently restored the interaction of MYC and UBE3B (Revised Fig 6e and 6f), and destabilized MYC protein (Revised Fig 6i and 6j). Furthermore, we found that cell-penetrated CM1 (PCM1) peptide derived from HLH domain of MYC impeded the interaction of MYC and UBE3B (Below panel a) and thus reduced the ubiquitylation of MYC (Below panel b). In this study, we emphasize CM4 peptide but not CM1 to validate the mechanism of TRIB3 in the regulation of MYC function. Hence, we present these data for your reviewing and will not display them in the revised MS.

(a) Effect of PCM1 treatment on MYC/UBE3B interaction. Raji cells were treated with CTRL (PCON) or PCM1 for 4 hr (4 μ M). Cell extracts were IP with rabbit immunoglobulin G (IgG) or anti-MYC Ab and blotted with anti-UBE3B Ab. **(b)** Effect of PCM1 treatment on MYC ubiquitination. Raji cells were treated with CTRL (PCON) or PCM1 for 4 hr (4 μ M). Extracts of cells were IP with anti-MYC Ab. The ubiquitinated MYC was detected by immunoblotting.

4. The leucine zipper does, however, interact with MAX, which is critical for MYC function. Therefore, the CM4 experiments in the critical Figures 5 and 6 simply show effects on the MAX/MYC interaction, which is well validated. Extended Figure 7d

clearly shows that the CM4 peptide binds MAX, this has to be the case. This data is also mis-portrayed in the text. Disrupting the MYC/MAX interaction is known to affect all aspects of MYC biology and - importantly- to destabilize the MYC protein, so all reported effects of CM4 are expected based on what is known about the MYC/MAX interaction and the effects of 10058-F4, a soluble inhibitor that disrupts the interaction. This problem and the manner in which the authors deal with it in the text precludes publication of the manuscript.

Re: Thank you for your professional criticism. In the revised MS, we further examined the effects of PCM4 on the co-localization and the interaction of MYC and MAX through confocal and CO-IP assays. The PLA assay showed the reduced foci of MAX/MYC co-localization in PCM4 treated Raji cells (Supplementary Fig. 7m). Interestingly, in the examination of PCM4's effects on the interaction of MYC and MAX by CO-IP assay, if sufficient MAX antibody (5 µg) was added to the lysates to capture the MAX protein, the quantity of the precipitated MYC protein in control cells was much more than that in PCM4 treated cells (Supplementary Fig. 7n left). However, if less anti-MYC antibody (1 µg) was added to capture the same quantity of MYC protein in the CO-IP assay, and the quantity of the precipitated MAX protein in control cells was identical with that in PCM4 treated cells (Supplementary Fig. 7n right). These data show that reduced MYC abundance induced by PCM4 contributes to the observed reduction of the MYC/MAX interaction.

Following your and the editor's suggestion, we have modified the description of the effects of CM4 on the MAX/MYC interaction in the revised MS (Page 16). Especially, we toned down the clarification of peptide CM4 specifically binding with TRIB3, and pointed out that disrupting the MYC/MAX and MYC/TRIB3 interactions contributed to anti-lymphoma effects of CM4 in the revised MS (Page 16).

There are additional issues, several of them major.

1. The choice of gene sets to show engagement of MYC varies among figures. The authors should show the Hallmark genes sets V1 and V2 in all panels to make things comparable.

Re: Following your suggestion, we have used the Hallmark genes sets V1 and V2 to perform GSEA analysis in the revised MS (Fig 2i, 4c, 6l, S5d, and S5e).

2. The analysis of T58 phosphorylation (lines 202ff) makes little sense. T58 regulates degradation of MYC since it is recognized by Fbxw7, so if T58 phosphorylation is important, as the authors state, the Fbxw7 is almost certainly involved. The authors however claim that it is not. Whether T58A is less ubiquitylated by UBE3B is not shown. Importantly this finding would allow the authors to test whether expression of T58A alleviates the dependence of MYC function and gene expression on TRIB3.

Re: Following your suggestion, we have removed the sentence of the analysis of T58 phosphorylation in the revised MS. Indeed, it is well recognized that MYC T58 phosphorylation is recognized and targeted by FBXW7 for its ubiquitylation and

proteasomal degradation. However, FBXW7 preferentially interacts with doubly phosphorylated MYC (Hao et al. Mol Cell 2007, 26 131-143; Welcker et al. Nat Rev Cancer 2008, 8 83-93), and mutation of T58 to a non-phosphorylatable residue stabilizes MYC more than ablation of FBXW7 does (Chakraborty et al. Exp Cell Res 2009, 315 1772-1778; Popov et al. Cell Cycle 2007, 6 2327-2331). Therefore, it has been suggested that one or more additional E3 ligase(s) also mediate(s) the proteolysis of MYC relying on T58 phosphorylation (Thomas et al. Adv Cancer Res 2011, 110 77-106). Moreover, Huber et al identified that CRY2 and FBXL3 cooperatively degrade MYC by binding MYC Phospho-T58 regardless of the presence or absence of FBXW7 (Huber et al. Molecular Cell 2016, 64 774-789). Similarly, in this study we found that UBE3B promotes MYC ubiquitylation and degradation via recognizing T58 phosphorylation in the absence of FBXW7, indicating that two E3 ligases represent independent pathway for MYC degradation. In the revised MS, we have modified the description of FBXW7 and toned down the interpretation of UBE3B in regulation of MYC ubiquitylation and degradation.

Following your suggestion, we detected the ubiquitylation of MYC T58A mediated by UBE3B. The data show that T58A mutation not only reduced MYC ubiquitination but also diminished the effect of *UBE3B* overexpression on MYC ubiquitination (Supplementary Fig. 4h). The corresponding description was indicated on line 207, 208 and 209 of revised MS (page 11).

Following your suggestion, CRISPR/Cas9 technology was used to generate *MYC*^{T58A} mutant Raji cells with or without *TRIB3* depletion (Supplementary Fig. 5a), the detailed method of which was presented in Page 33 and 34 of revised MS. We further examined the function and gene expression profiling of these cells. Compared with Raji cells with wild type *MYC* (*MYC*^{WT}), the *MYC*^{T58A} mutant Raji cells exhibited enhanced proliferation (Supplementary Fig. 5b) as well more and enlarged colonies (Supplementary Fig. 5c). Importantly, *TRIB3* knockdown in *MYC*^{T58A} mutant Raji cells did not change the abilities of proliferation and colony formation (Supplementary Fig. 5b, c), suggesting that the *MYC*^{T58A} mutant Raji cells are much less sensitive to *TRIB3* depletion compared with *MYC*^{WT} Raji cells. In addition, we analyzed the gene expression profiles of *MYC*^{T58A} mutant Raji cells with or without *TRIB3* deletion through RNA-seq. GSEA data showed the enriched MYC pathway-related genes in *MYC*^{T58A} lymphoma cells compared with *MYC*^{WT} lymphoma cells (Supplementary Fig. 5d). Similar with the observation of functional experiment, GSEA analysis shown no statistical difference between the *MYC*^{T58A} mutant Raji cells with or without *TRIB3* depletion (Supplementary Fig. 5e), suggesting that T58A mutation alleviates the dependence of MYC pathway-related gene expression on *TRIB3*. Taken together, these data indicate that the functions of *TRIB3* in promoting lymphoma cell growth depend on T58 phosphorylation of MYC.

3. The description of many experiments is cryptic and often panels lack critical controls. Just to give one example: Figure 4B analysis MYC binding in a cell line that expresses high levels of endogenous MYC. So how was endogenous MYC removed? What are the expression levels of MYC and MYCK427A? These are

absolutely essential data to interpret Figure 4b and this is simply impossible. A second example is Figure 4a, which claims that MYC binds to all human genes. This would imply that MYC binds also to inactive genes, which does not agree with the available literature. Again, critical controls are missing to underline this surprising finding.

Re: Thank you for your professional criticism. We agree with you that transfection of Raji cells with *MYC* and *MYC*^{K427R} plasmids indeed could not exclude the high levels of endogenous MYC in the original MS. In the revised MS, we first use CRISPR/Cas9 technology to generate *MYC*^{K427R} mutant Raji cell lines (Revised Supplementary Fig. 6b) and these cells have been used to examine *MYC*^{K427R} function and activity without transfection of exogenous plasmids. The detailed method has been illustrated in Page 33 and 34 of revised MS. Next, we performed the experiments of MYC ChIP-seq with spike-in normalization using *MYC*^{K427R} mutant Raji cells. We find that K427R mutation enhances the MYC occupancy at the promoters of its target genes (Revised Fig. 4a, 4b and Supplementary Fig. 6c). The corresponding description has been presented in page 12 of revised MS.

It was a mistake for the inaccurate description of Figure 4a legend. We have corrected it in the revised MS (page 54).

4. A large set of ChIP-sequencing data are based on quantitative comparisons between two sample sets, e.g. wtMYC and a mutant (Extended Data Figure 5a, 5e). In some cases, they show different individual genes and this is not acceptable: within 20,000 genes you will see any change possible. Where the authors report global changes, they are in line with the expected effects. However, such data need to be spike in-normalized to control for experimental variations: a small percentage of chromatin from another species needs to be added and used to standardize. The experimental methods show that this has not been done, therefore much of this work needs to be repeated.

Re: Following your suggestion, we re-performed all the ChIP-sequencing experiments using mouse chromatin spike-in for normalization. The corresponding experimental methods have been described in the revised Methods (Page 32 and 33). We also show change of the same individual gene (*CCNA2*) if two sample-sets have difference. These new data of ChIP-seq have been presented in revised Figure 2k, 2l, 2m, 4a, 4b, 4d, S6c, S6f, 6m and 6n.

Reviewer #3 (Remarks to the Author):

This study is very well executed showing a novel mechanism into Myc driven lymphomas, linked to a TRIB3 interaction, and the possible means of targeting this interaction to abrogate myc driven lymphomas. The experiments are well designed, well executed, and the data fully supports the finding. The paper is fully complete with little room for improvement. I only have minor comments:

1. In the extended data 1 the authors show that in the TRIB3 deletion models there is also deletion of TRIB2. This was not addressed at all in the experimental set up and may constitute an entirely new study. Can the authors comment how they think TRIB2 deletion is occurring in these models?

Re: This is a good point and suggestion. Indeed, we are planning a new study to examine whether and how the level of TRIB2 is regulated by TRIB3 in cancer cells. Actually, TRIB2 has been reported to act as a downstream molecule of Wnt/TCF; and Wnt/ β -catenin activation induces TRIB2 expression in cancer cells (Wang et al. Mol Cell 2013, 51 211-225.). Our recent study indicates that *TRIB3* depletion decreases β -catenin/TCF transcriptional activity by reducing recruitment of TCF and β -catenin to the promoter region of genes regulated by Wnt in colon cancer cells (Hua et al. Gastroenterology 2019, 156 708-721). Hence, we presumed that the reduced protein expression of TRIB2 was caused by *TRIB3* knockout-reduced *TRIB2* transcription by suppressing the activity of Wnt/ β -catenin in lymphoma cells. Following your and reviewer 4's suggestion, we have briefly discussed these in the section of discussion of the revised MS (Page 19 and 20).

2. Given the deletion of both TRIB3 and TRIB2, can the authors show the total cell numbers for also the myeloid and B populations as shown for total thymocytes in the extended data 1.

Re: Following your suggestion, we have presented the total numbers of splenic B cell, T cell and myeloid cell populations in 5-week old mice via flow cytometry analysis. *Trib3* ablation in myeloid/thymocyte/B cells did not affect the total number of thymocytes, splenic B cell, T cell and myeloid cell populations (Revised Supplementary Fig. 1f-1i), indicating that specific ablation of *Trib3* in these cells does not affect lymphocyte and myeloid cell development.

3. It would be better to group the graphs in figure 2d together and provide a statistical test to show the differences between the groups.

Re: Following your suggestion, we have regrouped the graphs together and provided a statistical data to this panel, which has been presented in Revised Figure 2d.

4. Details of the reporter genes in the luciferase assay in figure 2j should be named in the legend and R.L.A abbreviation explained in legend for how the data is graphed.

Re: Following your suggestion, we supplemented the details of the reporter genes and explained the R.L.A abbreviation in the legend of Revised Figure 2j.

5. In extended data 2, it appears that while *TRIB3* loss affects Myc protein in most lymphoma lines, many of the leukaemia lines (eg. U937, malm6, k562, molt4 and jurkat) do not exhibit such effects on Myc. The western, by eye, do not support the statements on line 158 pg8 and an adjustment on this interpretation may be necessary. The data supporting this mechanism in leukaemia is not as strong as the data and models the authors have for lymphoma.

Re: Thank you for your careful observation. Following your suggestion, we have modified the description of these data in the revised MS (Page 8).

6. What cells are used in extended data figure 2g and H – it is important to specify given point 5 above.

Re: Raji cells are used in extended data figure 2g and H. We have added this information in the legend of Supplementary Fig 2f and 2g.

7. The data is not convincing that K332R ubiquitination is enhanced when *UBE3B* is overexpressed – a point that does not affect the stated mechanism but is an overinterpretation of the data.

Re: We agree with your point. We have thus deleted the data of K332R ubiquitination regulated by *UBE3B* from the revised MS.

8. What antibody is used in the IP for the tri-molecular complex shown in extended data 6 M. This needs to be in the legend/figure to clarify.

Re: Cellular extracts were IP with mouse IgG as a negative control or anti-MYC, anti-*TRIB3*, or anti-MAX antibodies. Western blots were performed with anti-MYC, anti-*TRIB3*, and anti-MAX antibodies. We have described these in the legend of revised Figure 5m.

9. The authors discuss treatment with DOX. Page 15 line 297. Please explain and clarify what this is/why it is being used to treat the cells.

Re: Doxorubicin (DOX) is widely used as a chemotherapeutic agent for the treatment of lymphoma, which was illustrated in the Page 17 of the revised MS. In this study, we found that *TRIB3* deletion or PCM4 treatment enhanced the therapeutic efficacy of DOX in lymphoma, suggesting that the combination of DOX with *TRIB3* deletion or PCM4 produces a synergistic anti-lymphoma efficacy.

Reviewer #4 (Remarks to the Author):

In this paper, Li et al describe a molecular mechanism by which TRIB3 promotes lymphomagenesis via preventing proteasome mediated degradation of c-MYC. First, I'd like to commend the authors for their highly detailed mechanistic study, employing a range of mouse models and human/murine in vitro systems to demonstrate mechanistic details for the observed oncogenic role for TRIB3. I have no fundamental concerns with the data presented, but have a number of relatively minor comments that would hopefully improve the presentation of the data and make this work more accessible to readers not directly working in this field.

- The work is very elaborate and a hough amount of data is presented in each figure. As a consequence, it is almost impossible for the outside reader to have a high-level view of the message at present. I would highly recommend the development of a schematic capturing the main points of the molecular model the authors propose and incorporate this in the discussion/final figure.

Re: Following your suggestion, we have created a schematic diagram that illustrates the main points of the working model we proposed. The schematic diagram was shown in the Revised Figure 9 (Page 66).

- There are inconsistencies in abbreviating human/mouse genes and proteins throughout the manuscript. for instance, by convention, protein symbols are not italicized, and all letters are in upper-case, both for human and murine proteins. (see <https://www.biosciencewriters.com/Guidelines-for-Formatting-Gene-and-Protein-Names.aspx>). This needs to be addressed both in the text and figures.

Re: Following your suggestion, we have modified all the abbreviating human/ mouse genes and proteins throughout the text and figures in the revised manuscript.

- Were Myc^{Eμ}Cre^{CD19}Trib3^{F/+} mice homo or heterozygous for the Myc^{Eμ} allele? Please clarify this in the manuscript/figures

Re: Thank you for your reminder. The *Myc^{Eμ}Cre^{Lysm}Trib3^{F/+}*, *Myc^{Eμ}Cre^{Lck}Trib3^{F/+}*, *Myc^{Eμ}Cre^{CD19}Trib3^{F/+}*, *Myc^{Eμ}Cre^{CD19}Trib3^{F/F}* and *Myc^{Eμ}Cre^{ERT2}* mice are heterozygous for the Myc^{Eμ} allele. We have clarified this in the revised manuscript (Page 49, 51, and 55).

- Page 11: what is MAX?? Please ensure that you introduce the various genes you study and describe in this work.

Re: MAX (MYC-associated factor X), a member of transcription regulators, forms a sequence-specific DNA-binding protein complex with MYC and the MYC/MAX complex functions as a transcriptional activator. We have annotated this in the revised manuscript (Page 12).

- Ext. Fig. 1 G: Please at least comment on reduced TRIB2 levels you see in the Trib3 KO animals. TRIB2 has been shown to be oncogenic and some speculation at least

whether there is a genetic/functional hierarchy between TRIB2 and TRIB3 in this context would be interesting. Better still, it would be a further confirmation of TRIB3-specificity of this mechanism if you had evidence to show that restoration of TRIB2 levels (for instance in Raji cells) has no impact on the system you studying.

Re: Actually, TRIB2 was reported to act the downstream of Wnt/TCF and Wnt/ β -catenin activation induced TRIB2 expression in cancer cells (Wang et al. Mol Cell 2013, 51 211-225). Our recent study indicated that *TRIB3* depletion decreased β -catenin/TCF transcriptional activity by reducing recruitment of TCF and β -catenin to the promoter region of genes regulated by Wnt in colon cancer cells (Hua et al. Gastroenterology 2019, 156 708-721). In hence, we presumed that the decreased protein level of TRIB2 was caused by the *TRIB3* knockout-reduced *TRIB2* transcription through suppressing activity of Wnt/ β -catenin in lymphoma cells. Following your and reviewer 3's suggestion, we have briefly discussed in the section of discussion of the revised MS (Page 19 and 20).

Following your suggestion, we further restored TRIB2 expression in *TRIB3*-deleted Raji cells and examined the proliferation and self-renewal of these cells. We found that TRIB2 overexpression partially rescued the decreased cell growth of *TRIB3*-deleted Raji cells (Below panel a), but showed no impact on colony-forming capacity of these cells (Below panel b). These data suggest that reduced TRIB2 partially contributes to the inhibition of proliferation but not that of self-renewal induced by *TRIB3* deletion in lymphoma cells. Indeed, we will perform an entirely new study to examine whether and how the level of TRIB2 is regulated by TRIB3 in cancer cells. These data are shown for your review (Below panels a and b) but not presented in the revised MS.

(a) Relative cell viabilities of Control (*Ctrl*) and *TRIB3* knockout (*TRIB3^{Cas9}*) Raji cells with or without *TRIB2* overexpression for indicated times. *Ctrl* and *TRIB3^{Cas9}* Raji cells were infected with *TRIB2*- or *GFP*-adenovirus and 12 hr later, cell viabilities were detected by CCK-8 for indicated times. The colors represent different time points; the diameter indicates the relative cell viability. (b) Colony-forming capacity of *Ctrl* and *TRIB3^{Cas9}* Raji cells with or without *TRIB2* overexpression. Data are the mean \pm SEM of 3 assays. ** $p < 0.01$; $P > 0.05$ was considered not significant (NS).

- please include an alignment to show how specific CM4 is for TRIB3, compared to the other TRIB proteins in the supplementary material.

Re: Homology modeling based on the TRIB1 crystal structure (Murphy et al. Structure 2015, 23 2111-2121) predicted a hydrogen bond between Glu344 (E344) of the TRIB3 protein and the CM4 peptide (Revised Supplementary Fig. 7d), suggesting that the C terminus of TRIB3 is the binding domain of CM4. We have performed sequence alignment within the C termini of TRIB family members and found that E344 of TRIB3 is not conserved at the same positions on TRIB1 and TRIB2 (Revised Supplementary Fig. 7e), indicating that CM4 is specific for binding to TRIB3.

- Please comment in the manuscript what is the significance of the statement you make in page 5 "Targeted sequencing of TRIB3 showed that the allele frequency of nonsynonymous (Q84R) and synonymous (Y111Y and A323A) SNPs was separately 38.2% and 47.2% in lymphoma patients..."

Re: Following your suggestion, we have added described these data and commented this point in the revised MS (Page 5).

- There are a number of places where the manuscript will benefit from a careful proof-reading and correction of English. a few examples are copied below:

P4: "none of the inhibitors binding directly to c-Myc has been successfully..."

"We recently report that TRIB3..."

P10" We thus constructed UBE3B R346Q mutant. This mutant showed a reduced abilities of c-Myc ubiquitination..."

Re: Thank you for your pointing out these writing and grammar mistakes, we have carefully corrected and modified the manuscript. A native English speaker has proofread our revised manuscript throughout.

Reviewers' comments:

Reviewer #3 (Remarks to the Author):

The authors have addressing almost in full my critiques. I would ask for an amendment to figure 2d. The statistical analysis is not indicated in the legend for each comparison, and please add a comment in the body of text on page 7, line 134, whether the data is statistically significant for the cell doses used in the comparisons.

Reviewer #4 (Remarks to the Author):

The manuscript has been thoroughly revised and addressed all my points of concern.

Reviewer #5 (Remarks to the Author):

This manuscript provides a comprehensive and detailed description of the interaction of MYC with TRIB3 that stabilises MYC protein by disrupting UBE3B-mediated ubiquitination and degradation of MYC. The results are novel and interesting and should be published. The manuscript appears to have been substantially revised and improved following the first round of reviews. The data are well-presented with appropriate controls.

I have several minor points that can largely be answered in a modified text (see below). However, there remain two important issues.

First, the cell lines used express elevated levels of MYC whether or not rearrangements of the Myc gene are present. Given the high starting level of MYC and the relatively modest effects on MYC half-life elicited by changes in TRIB3 or UBE3B function there appears to be surprisingly large effect on MYC transcriptional output. Is there a TRIB3-sensitive MYC population (phospho-MYC?) that exhibits quantitatively different transcriptional activity that can account for the reported reduction of MYC-dependent transcription when TRIB3 is inhibited? After all, steady state MYC protein levels are easily detected in cells lacking TRIB3 presumably since MYC expression is constitutive, albeit with a reduced half-life. If only the stability of phospho-Myc (eg at T58) is affected, as shown in Supplementary Fig. 5, then perhaps phospho-T58 MYC should be shown (as in Supplementary Fig. 4I). What proportion of total MYC is phosphorylated at T58 in these cells?

Second, the data regarding the CM4 peptide is confusing and overly interpreted. For example, in Supplementary Fig. 7n the level of input MYC is lower in the presence of PCM4 in the left panel but this does not seem to be true in the right panel? In addition, I'm not sure that I follow the argument behind using "less anti-MYC antibody" for the IP in this figure. Surely 1µg antibody is saturating and the amount of MYC/MAX complex is the same between PCM4 and control. Are the authors suggesting that this amount of MYC/MAX complex is sufficiently low to account for the reduced MYC transcriptional output (Supplementary Fig 7o)? If so, these data are not convincing. It is also possible the effect of CM4 is mediated by inhibition of the MYC/MAX heterodimer independently of effects on TRIB3 and UBE3B interactions – this has not been adequately addressed in the manuscript.

Minor comments:

1. The efficiency of Cre in deleting Trib3 is clearly variable (compare Supplementary Figs. 1c and 1d) but I do not understand why Trib3 is completely absent from Trib3^{+/F} as well as Trib3^{F/F} thymocytes, especially when there appears to be robust expression in wt cells. Moreover, it is incorrect to imply that Trib3 is lost since there is clearly some expression even in Trib3^{F/F} bone

marrow and B lymphocytes.

2. I may have overlooked this but I can't find a description of the CreERT2 mice (Fig. 1j). Is CreERT2 expressed in all cells?
3. The title "TRIB3 Promotes Lymphoma Through MYC-Driven Malignant Activity" suggests that TRIB3 affects a peculiar neomorphic "malignant" activity of Myc, rather than promoting MYC's normal function which, when persistently activated, can lead to cancer.
4. The Ki-67 assay seems to be based on Trib3F/F cells whereas the CFU assay was on Trib3F/+ B cells (Fig 2b and 2c) – why is this the case? Given the effects seen in the CFU assay and in the mice it is important to show whether proliferation is also affected in Trib3F/+ cells.
5. The legend indicates that Fig. 2g shows "relative cell viabilities". I'm not clear whether the authors are referring simply to proliferative capacity or whether cell death was also observed? This is an important point – if Trib3 deletion inhibits the cell cycle (independently of Myc) then constitutive elevated expression of Myc is likely to induce apoptosis. There is also no mention of the possibility that Trib3 deletion induces differentiation (and cell cycle arrest) in, at least, some of the cell lines used.
6. GO term analysis is often misleading and that shown in Fig. 2h contributes very little and, given the density of data presented, could be removed. Likewise the GSEA in Fig. 2i.
7. The authors state (page 8, line 148) that "Consistently, TRIB3 depletion decreased increased MYC transcriptional activity" I'm not sure what the authors mean by this statement – what is the nature of the "increased MYC transcriptional activity that is "decreased" by Trib3 deletion?
8. The evidence linking RNA pol II occupancy in response to Trib3 deletion and Myc (Fig. 2k-m) is circumstantial. The effect of directly reducing Myc expression (eg RNAi) should be included as a control.
9. siTrib3 seems to have no effect on Myc protein expression in U937 cells (Fig. 3a). Was TRIB3 not repressed in U937 cells or is it not required for Myc stability in these cells?
10. Are the graphs shown in Fig. 3b and 3c densitometry scans of the representative westerns displayed in Supplementary Fig. 2g? If so, do the error bars represent technical or biological replicates and how many? This is a common failing throughout presentation of the figures and this information should be include in figure legends.
11. Fig. 3d – is it not slightly surprising that the steady-state level of Myc is not higher in the Flag-TRIB3 input sample, especially considering the apparent reduction in Myc ubiquitination. The same question applies to Supplementary Figs. 4h and 4i.
12. Why, in Supplementary Fig. 3i is the level of Myc protein so much lower in Raji cells with siCtrl in the siUBE3B panel compared to the siUBE3A and siUBE3C panels?
13. The authors overstate the result shown in Supplementary Fig. 4k. "The T58A mutation prevented UBE3B from binding to MYC". I agree that there is a reduction in MYC T58A binding but it is not "prevented".
14. The authors suggest that UBE3B-mediated K427 ubiquitination of MYC interferes with the MYC/MAX interaction (Supplementary Fig. 6k). Given that interaction of MAX and wt MYC is still observed it is important to determine the biological output of wt and K427R MYC in the absence and presence of UBE3B in Raji cells (eg proliferative assay as shown in Supplementary Fig. 6l). Was Ad-UBE3B present in Supplementary Fig. 6k? It is not mentioned in the figure legend.
15. The authors suggest that UBE3B ablation also induces the ubiquitination and degradation of other tumor suppressor(s) (page 13, line 258). Is the reduction in Raji cell number (Fig. 4f) solely due to reduced proliferation or was increased cell death also noted?
16. The data in Fig. 4 suggests that overexpression of TRIB3 alone does not increase spleen weight, lymph node size or proliferation – is this because endogenous levels of TRIB3 are already sufficient to suppress endogenous UBE3B activity?
17. Page 15, line 298 – I think the authors are referring to Fig. 6a (and not Fig. 5a).
18. In Fig. 6o do the two sets of bars on the graph represent two different isolates of primary DLBCL cells?

Reviewers' comments:

Reviewer #3 (Remarks to the Author):

The authors have addressing almost in full my critiques. I would ask for an amendment to figure 2d. The statistical analysis is not indicated in the legend for each comparison, and please add a comment in the body of text on page 7, line 134, whether the data is statistically significant for the cell doses used in the comparisons.

Re: Following your suggestion, we have added a comment in the body of text on Page 7, Line 135-137. A significant difference could be observed between the survival rates of recipient mice inoculated from *Myc^{Eμ}Cre^{CD19}Trib3^{F/+}* and *Myc^{Eμ}* mice for the transplantation cell dose of 5×10^4 , suggesting that BCL cells from *Myc^{Eμ}Cre^{CD19}Trib3^{F/+}* mice have a lower capacity for killing the recipient mice (Fig. 2d). We have also indicated the statistical analysis in the legend of Fig 2d for the comparison between the groups of the transplantation cell dose of 5×10^4 .

Reviewer #5 (Remarks to the Author):

This manuscript provides a comprehensive and detailed description of the interaction of MYC with TRIB3 that stabilises MYC protein by disrupting UBE3B-mediated ubiquitination and degradation of MYC. The results are novel and interesting and should be published. The manuscript appears to have been substantially revised and improved following the first round of reviews. The data are well-presented with appropriate controls.

I have several minor points that can largely be answered in a modified text (see below). However, there remain two important issues.

1. First, the cell lines used express elevated levels of MYC whether or not rearrangements of the Myc gene are present. Given the high starting level of MYC and the relatively modest effects on MYC half-life elicited by changes in TRIB3 or UBE3B function, there appears to be surprisingly large effect on MYC transcriptional output. Is there a TRIB3-sensitive MYC population (phospho-MYC?) that exhibits quantitatively different transcriptional activity that can account for the reported reduction of MYC-dependent transcription when TRIB3 is inhibited? After all, steady state MYC protein levels are easily detected in cells lacking TRIB3 presumably since MYC expression is constitutive, albeit with a reduced half-life. If only the stability of phospho-Myc (eg at T58) is affected, as shown in Supplementary Fig. 5, then perhaps phospho-T58 MYC should be shown (as in Supplementary Fig. 4l). What proportion of total MYC is phosphorylated at T58 in these cells?

Re: Thank you very much for your constructive suggestions. As you proposed in this point of comment, we examined whether there exists a TRIB3-sensitive MYC population that exhibits quantitatively different transcriptional activity of MYC in this round of revision. Serine 62 (S62) and threonine 58 (T58) phosphorylation of MYC is important for MYC-mediated gene regulation and its protein stability (Lüscher and Vervoorts, 2012; Myant et al., 2015; Sears et al., 1999; Sears et al., 2000; Wang et al., 2011). We used phospho-specific antibodies to accurately quantify the level of Phospho-T58 (P-T58) and Phospho-S62 (P-S62) relative to total MYC in *TRIB3* depleted cells. Upon *TRIB3* knockdown in Jurkat and Raji cells, we observed a decrease in the P-S62 signal, but no change in the P-T58 signal (Supplementary Fig. 2k and 2l). Quantifying the ratio of P-T58 to total MYC revealed a substantial increase in the amount of T58 phosphorylated MYC with *TRIB3* knockdown (Supplementary Fig. 2k and 2l). In addition, the signal for P-S62 relative to total MYC decreased in *TRIB3* depleted cells compared with control group (Supplementary Fig. 2k and 2l). Similarly, *TRIB3* overexpression increased S62 and decreased T58 phosphorylation relative to total MYC level in Raji cells (Supplementary Fig. 2m). Furthermore, the analysis of half-life of endogenous MYC protein revealed that *TRIB3* deletion reduced the stability of MYC protein, especially accelerated P-T58 MYC protein degradation after cycloheximide treatment, as compared to control gRNA (Supplementary Fig. 2n), suggesting mainly the stability of T58 MYC protein is affected in *TRIB3* deleted cells. These data indicate that the altered S62 and T58 phosphorylation of MYC also contributed to the MYC transcriptional output and regulation of MYC protein stability in *TRIB3*-depleted cells.

Supplementary Figure 2

We next explored the mechanism by which TRIB3 regulated MYC phosphorylation at T58 and S62. A number of reports have identified that GSK3B phosphorylates T58, whereas PP2A dephosphorylates S62 (Arnold et al., 2009; Sears et al., 2000). Pseudokinase TRIB3 has been shown to associate with several signal molecules, such as KAT5, β -catenin and AKT (Li et al., 2018; Hua et al., 2019; Yu et al., 2020). We found that endogenous GSK3B and PP2A-C co-immunoprecipitated with TRIB3 in Raji lymphoma cells (Below panel a). TRIB3 overexpression in lymphoma cells decreased the amount of GSK3B and PP2A that co-immunoprecipitated with MYC approximately 60% (Below panel b). Thus, TRIB3 post-translationally regulates MYC T58 and S62 phosphorylation consistent with its effects on the ability of MYC to associate with PP2A-C and GSK3B. Because we focused on investigating the role of TRIB3 in the regulation of MYC ubiquitylation in this study, these data are shown below only for your reviewing but not displaying in the revised MS.

Legend: (a) TRIB3 interacts with GSK3B and PP2A-C in Raji cells. Raji cells were lysed in Co-IP buffer and subject to IP with anti-TRIB3 or IgG as indicated. Input and immunoprecipitated proteins were detected by western blotting as indicated. **(b)** The interaction of MYC with GSK3B and PP2A is decreased by TRIB3 overexpression. Anti-MYC was immunoprecipitated from Raji cells transfected with HA-TRIB3 ($TRIB3^{OE}$) or HA-Control ($CTRL^{OE}$) as indicated. Input and immunoprecipitated proteins were detected by western blot.

Following your suggestion, we also examined the expression of phospho-T58 and total MYC in *TRIB3* depleted and/or MYC^{T58A} Raji cells. Similar with previous observation, *TRIB3* depletion decreased total MYC and increased P-T58 relative to total MYC levels. T58A mutation almost impeded *TRIB3*'s ability of reducing MYC expression (Below panel, see also Supplementary Fig. 5b), suggesting that T58 phosphorylation is essential for the process of MYC expression regulated by *TRIB3* depletion.

Supplementary figure 5b

Furthermore, we detected the levels of P-S62 and P-T58 MYC in UBE3B-overexpressed cells. Although UBE3B overexpression obviously reduced the levels of P-S62, P-T58 MYC and total MYC, UBE3B overexpression didn't affect P-S62 MYC, nor P-T58 MYC level relative to total MYC in Raji cells (Below panel, Supplementary Fig. 3j). Actually, UBE3B overexpression decreased the MYC/MAX interaction (Below panel, Supplementary Fig. 6j). Hence, UBE3B overexpression affected the MYC-dependent transcriptional output by both decreasing steady state MYC protein level and the complex formation of MYC/MAX.

Supplementary Figure 3j

Raji

Supplementary Figure 6j

2. Second, the data regarding the CM4 peptide is confusing and overly interpreted. For example, in Supplementary Fig. 7n the level of input MYC is lower in the presence of PCM4 in the left panel but this does not seem to be true in the right panel? In addition, I'm not sure that I follow the argument behind using "less anti-MYC antibody" for the IP in this figure. Surely 1µg antibody is saturating and the amount of MYC/MAX complex is the same between PCM4 and control. Are the authors suggesting that this amount of MYC/MAX complex is sufficiently low to account for the reduced MYC transcriptional output

(Supplementary Fig 7o)? If so, these data are not convincing. It is also possible the effect of CM4 is mediated by inhibition of the MYC/MAX heterodimer independently of effects on TRIB3 and UBE3B interactions – this has not been adequately addressed in the manuscript.

Re: To address the questions of reviewer 1 about the specificity of CM4 peptide in last round of revision, we performed more experiments to verify it and displayed all of them in the revised MS. However, these data of CM4 peptide, especially Supplementary Fig 7n, without detailed description of experiment procedures further confused this reviewer in the second round of peer review. First, different exposure time of left (4 sec) and right panel (30 sec) of in Supplementary Fig. 7n caused the discrepancy of the levels of input MYC. We indicated the exposure time of both panels and show the western blotting bands with the same exposure time in the revised Supplementary Fig. 7n (see also below panel).

Supplementary figure 7n

Second, as this reviewer suggested, it is true that 1 µg antibody is saturating to capture MYC protein from $0.5 \cdot 10^7$ lymphoma cell lysates during the CO-IP experiment. However, in supplementary Fig. 7n, we collected $4 \cdot 10^7$ Raji cells per samples, and performed the titration of MYC antibody as we did in the previous study (Li et al., Cancer Cell 2017). We found that 4 µg and 5 µg MYC antibody is saturating to immunoprecipitate MYC protein during this CO-IP assay (data shown below). Hence, the data of Supplementary Fig. 7m-7n is convincing and suggests that the low amount of MYC/MAX complex accounts for the reduced MYC transcriptional output. We have already indicated it in the revised legend of supplementary Fig 7n.

Legend: Lysates from $4 \cdot 10^7$ Raji cells were immunoprecipitated (IP) with indicated amount of anti-MYC antibodies, and the levels of immunoprecipitated MYC were analyzed by immunoblotting.

Furthermore, we used the NanoBRET™ MYC/MAX protein interaction assay (#N1870, Promega Biotech Co., Ltd) to identify whether CM4 interferes with the MYC:MAX protein interaction in a dose-dependent manner using the respective proteins tagged with NanoLuc® luciferase (NanoLuc®) and HaloTag® ligands. With 10058-F4 as a PPI inhibitor of MYC-MAX interaction, we found that CM4 showed no effects on the MYC/MAX heterodimer in this assay (Below panel, see also Supplementary Fig. 7o). In addition, quantification of the MYC/MAX heterodimer using Proximity Ligation Assay (PLA) has been performed in the CM4-treated Raji cells

with proteasome inhibitor MG132 reversing MYC degradation. CM4 treatment indeed didn't affect the colocalization of MYC and MAX in MG132-treated Raji cells (Below panel, see Supplementary Fig. 7p). Overall, our results indicate that CM4 showed specificity to MYC and TRIB3 but not to the MYC and MAX.

Supplementary Figure 7o

Supplementary Figure 7p

Minor comments:

1. The efficiency of Cre in deleting Trib3 is clearly variable (compare Supplementary Figs. 1c and 1d) but I do not understand why Trib3 is completely absent from Trib3^{+/F} as well as Trib3^{F/F} thymocytes, especially when there appears to be robust expression in wt cells. Moreover, it is incorrect to imply that Trib3 is lost since there is clearly some expression even in Trib3^{F/F} bone marrow and B lymphocytes.

Re: Thank you for your careful observation. Actually, expression of TRIB3 protein was reduced by approximately 60-80% in thymocytes of Trib3^{F/+} mice, compared with WT mice. In the last version of MS, the image of TRIB3 western blotting in thymocytes were not chosen appropriately to display in Supplementary Fig 1d. We showed the representative blotting result of TRIB3 in this revised MS (Supplementary Fig. 1d right). Furthermore, we noticed that there was some expression of TRIB3 even in bone marrow and B lymphocytes of Cre^{Lysm}Trib3^{F/F} and Cre^{CD19}Trib3^{F/F} mice, which might be caused by the variable efficiency of Cre-mediated recombination and individual difference in Trib3^{F/F} mice. We have corrected the description of TRIB3 expression in heterozygous and homozygous deletion of *Trib3* mice in the revised MS (Page 5, Line 100).

2. I may have overlooked this but I can't find a description of the CreERT2 mice (Fig. 1j). Is CreERT2 expressed in all cells?

Re: We indicated the brief description of the Cre^{ERT2} mice on Page 24 of the original MS. When Cre^{ERT2} (B6. Cg-Tg(CAG-cre/Esr1*)5Amc/J) mice bred with mice containing loxP-flanked sequences, tamoxifen-inducible Cre-mediated recombination results in deletion of the floxed sequences in widespread cells/tissues of the offspring, which has been already supplemented in the Methods of revised MS (Page 26, Line 521-524).

3. The title "TRIB3 Promotes Lymphoma Through MYC-Driven Malignant Activity" suggests that TRIB3 affects a peculiar neomorphic "malignant" activity of Myc, rather than promoting MYC's normal function which, when persistently activated, can lead to cancer.

Re: Following your suggestion, we have already modified the subtitle with "TRIB3 Promotes Lymphoma by Supporting MYC Activity" (Page 7, Line 131).

4. The Ki-67 assay seems to be based on Trib3^{F/F} cells whereas the CFU assay was on Trib3^{F/+} B cells (Fig

2b and 2c) – why is this the case? Given the effects seen in the CFU assay and in the mice it is important to show whether proliferation is also affected in Trib3^{F/+} cells.

Re: Actually, we performed the Ki-67 assay and CFU assay both on the Trib3^{F/F} and Trib3^{F/+} cells. For consistency, we showed CFU assay on Trib3^{F/F} B cells in revised Fig. 2c and the results of Ki-67 assay on Trib3^{F/+} cells below for your reviewing.

Legend: Lymphoma cell proliferation was determined by Ki67 staining in spleens of *Myc^{Eμ}* or *Myc^{Eμ}Cre^{CD19}Trib3^{F/+}* mice (5 months old). Data are representative images of Ki67 staining (left) and statistical analyses of Ki67 positive cells (right). Scale bar, 50 μm.

5. The legend indicates that Fig. 2g shows “relative cell viabilities”. I’m not clear whether the authors are referring simply to proliferative capacity or whether cell death was also observed? This is an important point – if Trib3 deletion inhibits the cell cycle (independently of Myc) then constitutive elevated expression of Myc is likely to induce apoptosis. There is also no mention of the possibility that Trib3 deletion induces differentiation (and cell cycle arrest) in, at least, some of the cell lines used.

Re: In the original Fig. 2g, relative cell viabilities detected by CCK-8 refer to proliferative capacity. Following your suggestion, we examined the cell death and differentiation in Raji cells with or without *TRIB3* deletion. *TRIB3* deletion did not induce a sub-G1 fraction of the cellular DNA content (Below panel a, see also Supplementary Fig. 2a), nor did it increase CD11b positive cells ratio in FACS analysis (Below panel b, see also Supplementary Fig. 2b), demonstrating that *TRIB3* depletion does not induce cell death or differentiation in Raji cells.

Supplementary Figure 2

6. GO term analysis is often misleading and that shown in Fig. 2h contributes very little and, given the density of data presented, could be removed. Likewise the GSEA in Fig. 2i.

Re: Following your suggestion, we have already deleted the data of Fig. 2h from the revised MS. The GSEA in Fig. 2i has been moved to the Supplementary Fig. 2c of revised MS.

7. The authors state (page 8, line 148) that “Consistently, TRIB3 depletion decreased increased MYC transcriptional activity” I’m not sure what the authors mean by this statement – what is the nature of the “increased MYC transcriptional activity that is “decreased” by Trib3 deletion?

Re: It is our mistake. We have already corrected it in the revised MS (page 8, line 152).

8. The evidence linking RNA pol II occupancy in response to Trib3 deletion and Myc (Fig. 2k-m) is circumstantial. The effect of directly reducing Myc expression (eg RNAi) should be included as a control.

Re: During last rounds of revision, one reviewer suggested that MYC’s effects on global RNAPII function are well established and should be documented by a metagene plot in this study. Following his suggestion, we performed ChIP-sequencing with an antibody against total RNAPII in the revision of MS. We also carried out an analysis of gene expression changes and comparison to MYC depletion (RNAi) in the group of *TRIB3*-depleted cells. Cluster analysis of the heatmap revealed that the *TRIB3*-depleted and *MYC*-depleted groups had similar MYC target gene expression tendencies (Revised Supplementary Fig. 2e).

9. siTrib3 seems to have no effect on Myc protein expression in U937 cells (Fig. 3a). Was TRIB3 not repressed in U937 cells or is it not required for Myc stability in these cells?

Re: Thank you for your careful observation. The reason that siTrib3 have no effect on MYC protein expression in U937 cells may be caused by the low expression of UBE3B in U937 cells (data shown below).

10. Are the graphs shown in Fig. 3b and 3c densitometry scans of the representative westerns displayed in Supplementary Fig. 2g? If so, do the error bars represent technical or biological replicates and how many? This is a common failing throughout presentation of the figures and this information should be include in figure legends.

Re: The graphs shown in Fig. 3b and 3c densitometry scans of the representative westerns were indeed displayed in Supplementary Fig. 2i and 2j. The data of Fig. 3b and 3c are presented as the mean +S.E.M from 3 independent assays. Following your suggestion, we have indicated this information throughout the figure legends of revised MS.

11. Fig. 3d – is it not slightly surprising that the steady-state level of Myc is not higher in the Flag-TRIB3 input sample, especially considering the apparent reduction in Myc ubiquitination. The same question applies to Supplementary Figs. 4h and 4i.

Re: In this study, *in vivo* ubiquitylation assays (cellular level) were performed with treatment of proteasome inhibitor MG132. The according description was indicated as the Method section of revised MS (Page 33). Therefore, MYC protein accumulates in both of control group and *TRIB3* overexpression, *T58A*, or *FXBW7* depletion group, which caused no significant difference of the steady-state MYC levels between these groups. If *TRIB3* was overexpressed or *UBE3B* was depleted in Raji cells without MG132 treatment, the steady-state level of MYC increased in these manipulated cells (Supplementary Fig. 2m and 3i).

12. Why, in Supplementary Fig. 3i is the level of Myc protein so much lower in Raji cells with siCtrl in the siUBE3B panel compared to the siUBE3A and siUBE3C panels?

Re: The exposure time of MYC western bands in the siUBE3A and siUBE3C panels were longer (60 sec) than that in the siUBE3B panel (Original Supplementary Fig. 3i). In the revised MS, we re-detected the MYC expressions in Ctrl, UBE3A, UBE3B or UBE3C silenced Raji cells with the same exposure time. Similar with previous observation, knockdown of *UBE3B*, but not the analogous family members *UBE3A* or *UBE3C*, increased MYC expression in Raji cells (Below panel, Revised Supplementary Fig. 3i).

13. The authors overstate the result shown in Supplementary Fig. 4k. “The T58A mutation prevented UBE3B from binding to MYC”. I agree that there is a reduction in MYC T58A binding but it is not “prevented”.

Re: Following your suggestion, we have corrected the description in the revised MS (Page 12, Line 233).

14. The authors suggest that UBE3B-mediated K427 ubiquitination of MYC interferes with the MYC/MAX interaction (Supplementary Fig. 6k). Given that interaction of MAX and wt MYC is still observed it is important to determine the biological output of wt and K427R MYC in the absence and presence of UBE3B in Raji cells (eg proliferative assay as shown in Supplementary Fig. 6l). Was Ad-UBE3B present in Supplementary Fig. 6k? It is not mentioned in the figure legend.

Re: Following your suggestion, we examined *MYC*^{K427R} biological function in the established *MYC*^{K427R} mutant Raji cells with or without *Ad-UBE3B* infection. K427R mutation of MYC increased the growth of Raji cells, while *UBE3B* overexpression lost its ability of reduction in *MYC*^{K427R} Raji cell number compared to that in *MYC*^{WT} Raji cells (Below Panel, see also Supplementary Fig. 6l), suggesting UBE3B-mediated K427 ubiquitination of MYC played an essential role in determining the biological output of UBE3B.

Ad-UBE3B was absent in Supplementary Fig. 6k, which has been indicated in the figure legend of Supplementary Fig. 6k.

15. The authors suggest that UBE3B ablation also induces the ubiquitination and degradation of other tumor suppressor(s) (page 13, line 258). Is the reduction in Raji cell number (Fig. 4f) solely due to reduced proliferation or was increased cell death also noted?

Re: Following your suggestion, we detected the cell death ratio of Raji cells with *UBE3B* deletion or *UBE3B* overexpression via Annexin V-APC/7-AAD staining. *UBE3B* overexpression didn't induce apoptosis of Raji cells (Below panel a, see also Supplementary Fig. 6n). However, *UBE3B* ablation obviously increased the rates of apoptotic Raji cells (Below panel b, see also Supplementary Fig. 6o), which may be caused by constitutive elevated expression of MYC. These data have indicated that the reduction in Raji cell number with *UBE3B* overexpression was due to reduced proliferation while the decreased growth in *UBE3B*-ablated Raji cells was owing to increased cell apoptosis (Page 14 and 15, Line 288-294).

16. The data in Fig. 4 suggests that overexpression of TRIB3 alone does not increase spleen weight, lymph node size or proliferation – is this because endogenous levels of TRIB3 are already sufficient to suppress endogenous UBE3B activity?

Re: We agree with your point that high levels of endogenous TRIB3 are already sufficient to suppress endogenous UBE3B activity in lymphoma cells. In hence, overexpression of TRIB3 alone does not further increase spleen weight, lymph node size or proliferation in this study. We have added the according description to the revised MS (Page 15, Line 301-303).

17. Page 15, line 298 – I think the authors are referring to Fig. 6a (and not Fig. 5a).

Re: We have corrected this mistake in the revised MS (Page 17, Line 335).

18. In Fig. 6o do the two sets of bars on the graph represent two different isolates of primary DLBCL cells?

Re: Two sets of bars represented two times of the serial colony formation of primary human DLBCL cells (T69). Primary lymphoma cells were treated with PCON or PCM4 in methyl-cellulose, and at 14 days, colonies were counted (P1). A representative plate was then washed and cells were resuspended and replated. After an additional 14 days, colonies were counted (P2). We have clarified these in the method section of revised MS (Page 36, Line 810-813) and in the legend of revised Fig. 6o.

REVIEWERS' COMMENTS

Reviewer #5 (Remarks to the Author):

I thank the authors for their clear and detailed responses in their revised manuscript in response to my previous comments. I am satisfied that the authors have addressed all of these comments and have produced an improved and robust manuscript.

Reviewer #5 (Remarks to the Author):

I thank the authors for their clear and detailed responses in their revised manuscript in response to my previous comments. I am satisfied that the authors have addressed all of these comments and have produced an improved and robust manuscript.

Re: Thank you for your positive comments. It is your professional suggestion that has enabled us to make further improvements to our manuscript.